# Heterocellular OSM-OSMR signalling reprograms fibroblasts to promote pancreatic cancer growth and metastasis

Brian Y. Lee[1,6], Elizabeth K. J. Hogg [1,6], Christopher R. Below [1], Alexander Kononov[1], Adrian Blanco-Gomez[1], Felix Heider[1], Jingshu Xu[1], Colin Hutton[1], Xiaohong Zhang[1], Tamara Scheidt[2], Kenneth Beattie[3], Angela Lamarca[4,5], Mairéad McNamara [4,5], Juan W. Valle [4,5] & Claus Jørgensen [1✉]

Pancreatic ductal adenocarcinoma (PDA) is a lethal malignancy with a complex micro-environment. Dichotomous tumour-promoting and -restrictive roles have been ascribed to the tumour microenvironment, however the effects of individual stromal subsets remain incompletely characterised. Here, we describe how heterocellular Oncostatin M (OSM) - Oncostatin M Receptor (OSMR) signalling reprograms fibroblasts, regulates tumour growth and metastasis. Macrophage-secreted OSM stimulates inflammatory gene expression in cancer-associated fibroblasts (CAFs), which in turn induce a pro-tumourigenic environment and engage tumour cell survival and migratory signalling pathways. Tumour cells implanted in *Osm*-deficient (*Osm*$^{-/-}$) mice display an epithelial-dominated morphology, reduced tumour growth and do not metastasise. Moreover, the tumour microenvironment of *Osm*$^{-/-}$ animals exhibit increased abundance of α smooth muscle actin positive myofibroblasts and a shift in myeloid and T cell phenotypes, consistent with a more immunogenic environment. Taken together, these data demonstrate how OSM-OSMR signalling coordinates heterocellular interactions to drive a pro-tumourigenic environment in PDA.

---

[1] Cancer Research UK Manchester Institute, The University of Manchester, Alderley Park, SK10 4TG Manchester, UK. [2] Department of Molecular Biology, University of Salzburg, Salzburg, Austria. [3] FingerPrints Proteomics Facility, College of Life Sciences, University of Dundee, Dundee DD1 5EH, UK. [4] Department of Medical Oncology, The Christie NHS Foundation Trust, Wilmslow Road, M20 4BX Manchester, UK. [5] Institute of Cancer Sciences, University of Manchester, Wilmslow Road, M20 4BX Manchester, UK. [6] These authors contributed equally: Brian Y. Lee, Elizabeth K.J. Hogg. ✉email: claus.jorgensen@cruk.manchester.ac.uk

Pancreatic cancer is characterised by an extensive desmoplastic reaction, which constitutes on average 85% of the tumour volume[1,2]. Here, mutated tumour cells are embedded in a stiff, remodelled extracellular matrix (ECM), which is abundantly co-inhabited by stromal host cells[3]. Dysregulated immune cells and cancer-associated fibroblasts (CAFs) are a perpetual and dynamically regulated feature throughout PDA development[2,4]. CD4[+] regulatory T cells, myeloid derived suppressor cells and macrophages (MØs) infiltrate early during PDA development, and directly promote tumour progression through matrix metalloproteinase (MMP) and IL6 signalling, while simultaneously suppressing anti-tumour immunity[4–8]. Consequently, targeting MØs and granulocytes, such as through CSF1R or CXCR2 inhibitors, increases T cell infiltration and sensitises tumours to immune checkpoint inhibitors (ICIs)[9,10]. In addition, tumour cells conscript resident pancreatic fibroblasts to support tumour cell survival, metabolic adaptation and exclusion of CD8[+] T cells[11–14]. Targeting CAFs and the stiff, remodelled ECM such as through vitamin D receptor agonists, lysyl oxidase and leukaemia inhibitory factor (LIF) blocking antibodies or CXCR4 inhibitors improves response to chemotherapy and sensitises to ICIs[11,15–17]. Tumour-restrictive stromal cell populations have also been described in PDA, where genetic and pharmacologic ablation of CAFs accelerates disease progression and metastatic dissemination[18–21]. Yet, the mechanisms by which discrete stromal populations coordinate to regulate specific tumour functions remain incompletely characterised[2,22].

Single-cell analysis has been instrumental in annotating individual stromal cell populations[21,23–27]. Recently, several distinct CAF populations have been identified, including myofibroblastic CAFs (myCAF), defined by elevated αSMA expression and ECM modulating features; inflammatory CAFs (iCAFs), defined by expression of several inflammatory genes; and antigen-presenting CAFs (apCAFs), defined by expression of MHCII and CD74[21,27–30]. The transcriptional state of these CAF populations is dynamically regulated, and pharmacological intervention can therefore be used to manipulate these CAF subsets in vitro and in vivo[29,30]. Several signals have been demonstrated to induce specific CAF subtypes, where tissue stiffness, TGFβ signalling and spatial proximity to tumour nests induce myCAF features, whilst tumour cell-secreted IL1α drives the iCAF phenotype through the JAK-STAT pathway[29,30]. Importantly, CAF and immune cell regulation of PDA pathobiology is highly interdependent, where targeting individual populations is accompanied by a concomitant adaptation of the tumour microenvironment (TME). For example, ablation of αSMA[pos] or FAP[pos] CAFs sensitises tumours to ICIs, targeting the stiff ECM alters immune infiltration, and pharmacologic and genetic manipulation of individual immune subsets drastically alters CAF functions[10,11,15,19,20,23,31]. Consequently, determining the functional interdependencies between dynamically regulated CAF subsets, immune infiltrates and tumour cell function remains critical to ascertain the benefits of stromal targeting strategies.

In this work, we explore the impact of heterocellular interactions across tumour cells, fibroblasts and MØs on PDA pathobiology. We observe that tumour cell-derived granulocyte–MØ colony-stimulating factor (GM-CSF) induces expression of Oncostatin M (OSM) in MØs, which in turn induces expression and secretion of pro-tumourigenic inflammatory mediators, such as IL6 family members (e.g. IL6 and LIF) and CXC- and CC-chemokines (CXCL1, CCL2 and CCL7), in stromal fibroblasts. This aberrant signalling milieu engages survival, motility and cell migration signalling pathways in tumour cells to accelerate metastasis. Pancreatic cancer cells (PCCs) orthotopically transplanted into immune-competent mice lacking Osm ($Osm^{−/−}$) generate significantly smaller tumours which exhibit an altered microenvironment with a reduction in tumour-promoting cytokines and chemokines (IL6, CXCL1, TNFα and GM-CSF) and increased abundance of αSMA[pos] myofibroblasts. The cognate OSM receptor, OSMR, is expressed in mesenchymal stromal cells, and in particular in fibroblasts and perivascular cells. Notably, OSMR expression level is correlated with inflammatory gene expression, a tumour-permissive immune infiltrate and a poor overall survival in human PDA. Taken together, the data presented establishes a role of OSM-OSMR signalling in driving an inflammatory microenvironment and supporting tumour growth.

## Results

**OSMR expression is associated with tumour-promoting inflammation and poor outcome in PDA.** PDA progression is accelerated by an extensive fibroblast expansion and a concomitant myeloid cell infiltrate[2,4]. Several tumour cell-derived signals have been shown to regulate fibroblast function[29,30,32], however, it is less clear whether other cell populations in the tumour ecosystem influence CAF function. To identify and characterise novel stromal receptor and ligand pair(s) that may regulate CAF function, we re-analysed available RNA-seq data of murine and human PDA. Differential gene expression analysis of iCAF and myCAF subsets[29] identified several membrane receptors, such as Osmr, Antxr1 and Il1r1, which were expressed at a higher level in iCAFs (Fig. 1a). Single-cell RNA-sequencing confirmed expression of Osmr, Antxr1 and Il1r1 in CAFs as well as in endothelial and perivascular cells in both human and murine PDA[27,33]. Moreover, these receptors were expressed at higher levels in iCAFs compared to myCAFs and apCAFs (Fig. 1b, c and Supplementary Fig. 1a, b). To further analyse OSMR expression in human resected PDA we used multiplexed in situ mRNA hybridisation for OSMR with immunofluorescence for Vimentin (to broadly demark stroma) and pan-Cytokeratin (to demark epithelial cells) (Fig. 1d and Supplementary Fig. 1c). Consistent with the scRNA-seq data, OSMR was broadly expressed within the VIM[pos] tumour stroma, however, OSMR-positive cells were also observed within the epithelium (PanCK-[pos]) cells.

To then determine whether expression of OSMR, ANTX1 and IL1R1 was correlated with patient prognosis, we studied their association with patient survival. Analysis of normal tissue samples (GTEx) and human PDA tumours (TCGA PanCancer PAAD) revealed a significantly higher expression level of OSMR and its cognate ligand, OSM, in human PDA compared to normal pancreatic tissue (Fig. 1e). Moreover, of the three iCAF-enriched receptors analysed, OSMR was the only receptor where expression was associated with poor outcome in PDA patients, where PDA patients with higher mRNA expression levels of OSMR exhibited worse overall survival than patients with lower OSMR expression (Fig. 1f and Supplementary Fig. 1d). Further investigating the association between Osmr expression and the iCAF phenotype in vivo, we compared the expression of inflammatory mediators in Osmr[pos] and Osmr[neg] CAFs using publicly available scRNA-seq datasets of KPC[27] and KPP[33] tumours, which revealed elevated expression of Il6, Ccl2, Ccl7 and Cxcl1 in Osmr[pos] CAFs (Fig. 1g).

To investigate how the OSM–OSMR ligand–receptor interaction may be regulated in PDA, we analysed the expression pattern of cognate ligands for Osmr, Il1ra and Antxr1 receptors in both human and murine scRNA-seq datasets. OSM was expressed in MØs and myeloid cells in both murine and human PDA, as well as in murine neutrophils and dendritic cells (Fig. 1h), contrasting a broader expression of IL1R1 and ANTXR1 ligands (Supplementary Fig. 1e). This data indicated that myeloid/MØ-secreted OSM may act in a paracrine manner on OSMR-expressing

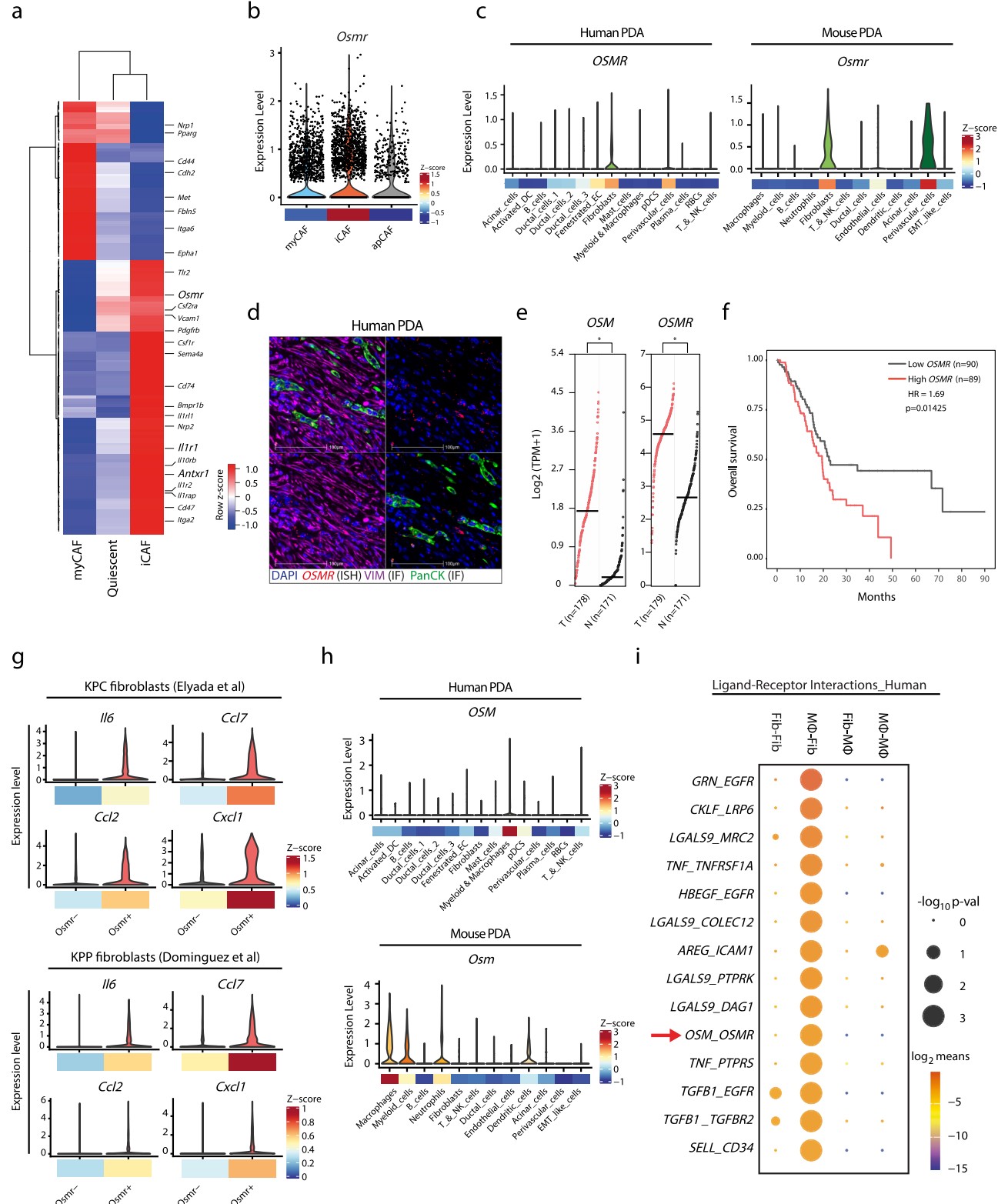

mesenchymal cells, such as fibroblasts. To test this hypothesis, we subjected publicly available scRNA-seq datasets of human and mouse PDA to CellPhoneDB analysis and evaluated MØ–fibroblast ligand–receptor interactions[27,34]. This analysis further highlighted *OSM–OSMR* as a stromal paracrine signal conserved across both human and mouse PDA (Fig. 1i and Supplementary Fig. 1f–h). Given an established role of CAFs as fibroinflammatory regulators, we hypothesised that heterocellular

OSM–OSMR signalling may induce inflammatory fibroblasts in PDA.

**MØ–tumour cell interactions induce inflammatory fibroblasts in vitro.** To further analyse heterocellular OSM–OSMR signalling, we firstly determined whether co-cultures of pancreatic stellate cells (PSCs), PCCs and MØs in vitro increase the

**Fig. 1 OSMR expression is associated with tumour-promoting inflammation and poor outcome in PDA. a** Heatmap displaying relative expression levels of membrane receptors of stellate cells adopting a quiescent, inflammatory cancer-associated fibroblasts (iCAF), or myofibroblastic CAF (myCAF) phenotype (dataset: GSE93313). **b** Violin plot of normalised *Osmr* expression levels in myCAF, iCAF, and apCAFs in murine PDA (dataset: GSE129455) with mean expression *z*-scores shown below. **c** Violin plots of normalised *OSMR/Osmr* expression levels for individual cell types in human (left) and murine (right) PDA (top) and mean expression *z*-scores shown below (datasets: GSE129455, phs001840.v1.p1). **d** Representative in situ mRNA hybridisation (ISH) of *OSMR* multiplexed with immunofluorescence of VIM and PanCK in human PDA (*n* = 3). 20× magnification. Upper left quadrant, full overlay; upper right, *OSMR*; lower left, *OSMR* with VIM; lower right, *OSMR* with PanCK. Scale bar = 100 μm. **e** *OSM* (left) and *OSMR* (right) expression in human PDA (TCGA PanCancer PAAD dataset, *n* = 178) and normal pancreatic tissue (GTEx, *n* = 171). Mantel–Cox test. T tumour, N normal, TPM transcripts per million. **f** Kaplan–Meier survival curves showing overall survival of 179 PDA patients from TCGA PanCancer PAAD dataset. Patients were stratified by high (Top 50%, *n* = 89) and low (bottom 50%, *n* = 90) *OSMR* expression levels. Analysis by log-rank test and cox-proportional hazard regression. HR hazard ratio. **g** Violin plots of inflammatory gene expression in *Osmr*[pos] and *Osmr*[neg] CAFs in murine PDA tumours (datasets: GSE129455, MTAB-8483) with mean expression *z*-scores shown below. **h** Violin plots of normalised *OSM/Osm* expression across cell types of human (top) and murine (bottom) PDA (datasets: GSE129455, phs001840.v1.p1). Mean expression *z*-scores shown. **i** Top 14 predicted interactions between macrophage-derived ligands and fibroblast expressed receptors using CellPhoneDB receptor–ligand interaction statistical analysis (*p*-values < 0.05). Also see Supplementary Fig. 1. Source data are provided as a Source Data file.

production of inflammatory mediators. We, therefore, quantified the abundance of 30 soluble inflammatory signals in the conditioned medium from mono- and co-cultures of PCCs, PSCs and MØs, by LUMINEX (Fig. 2a). Media from 2-way (2 W) co-cultures of PCC–PSC and PCC–MØ displayed moderately elevated levels of several tumour-promoting signals compared to mono-cultured PCCs, PSCs and MØs (Fig. 2a and Supplementary Fig. 2a)[17,35,36]. Furthermore, despite seeding an equal total cell number, 3-way (3 W) co-cultures of PCCs, PSCs and MØs lead to a profound change in the signalling milieu with a further increased abundance of 22 cytokines and chemokines (Fig. 2a and Supplementary Fig. 2a). Notably, several pro-tumoural signals (e.g. IL6, LIF, CXCL1, GM-CSF and CCL2), known to cause stromal activation and immunosuppression in PDA, increased drastically (Supplementary Fig. 2a)[5,17,35–37]. Thus, heterocellular interactions between tumour cells, MØs and stromal fibroblasts accentuate secretion of inflammatory signals in vitro.

To subsequently determine the cellular source(s) of individual signals, we FACS-isolated PCCs, PSCs and MØs from mono- and co-cultures and analysed the expression level of selected signals by RT-qPCR (Supplementary Fig. 2b). Most transcripts were expressed and regulated in a cell-type specific manner (Fig. 2b). For example, MØs expressed *Tnf* and *Osm*, PSCs expressed *Il6*, *Cxcl9* and *Cxcl10*, and PCCs expressed *Csf2* as well as *Cxcl5*. Other signals, such as *Ccl2* and *Ccl7*, were expressed in MØs co-cultured with either PCC or PCC–PSC. However, *Ccl2* and *Ccl7* were also expressed in PSCs isolated from 3 W co-cultures with PCCs and MØs (Fig. 2b), demonstrating context-dependent control of inflammatory gene expression. Unsupervised hierarchical clustering and principle component analysis (PCA) of mRNA levels highlight that PSCs engaged discrete secretory programmes depending on specific heterocellular interactions (Fig. 2b and Supplementary Fig. 2c). In particular, PSCs isolated from PCC–PSC–MØ co-cultures (herein called 3W-PSCs) clustered separately from PSCs isolated from mono- and 2 W co-cultures (Fig. 2b and Supplementary Fig. 2c left), and displayed increased expression of interleukins such as *Il6*, as well as CC-/CXC-ligands such as *Cxcl1*, *Ccl2* and *Ccl7* (Supplementary Fig. 2c right). In contrast, PSCs from PCC–PSC 2 W co-culture (herein called 2W-PSCs) exhibited a distinct secretory pattern with increased expression of *Cxcl12*, *Tgfb2* and *Ccl11* (Fig. 2b and Supplementary Fig. 2c right). The analysed ligands were not differentially expressed in PCCs or MØs isolated from mono- and co-cultures (Fig. 2b), suggesting that PSCs are particularly responsive to changes in the local cellular environment.

To further investigate whether interactions across MØs, PCCs and PSCs induce a transcriptional programme in PSCs that is consistent with iCAFs, we FACS-isolated PSCs from mono-,

PCC–PSC and PCC–PSC–MØ co-cultures, and analysed mRNA expression by RNA-seq (Supplementary Fig. 2d). Gene expression analysis identified 1758 differentially regulated genes in PSCs from 2 and 3 W co-cultures. 2W-PSCs displayed increased expression of genes encoding ECM proteins including collagens *Col6a1*, *Col11a1*, *Col4a5*, *Col4a6* as well as *Acta2*, (α-smooth muscle actin, αSMA) a well-documented marker of myofibroblasts (Fig. 2c)[22]. Similarly, inflammatory signals, many of which were identified in the conditioned medium from 3 W co-cultures (Fig. 2a and Supplementary Fig. 2a), were abundantly expressed in 3W-PSCs, confirming fibroblasts as a major cellular source of inflammatory signals (Fig. 2c and Supplementary Fig. 2e). For example, upregulated genes in 3W-PSCs include CC-chemokines (*Ccl2*, *Ccl7* and *Ccl5*), JAK-STAT (*Osmr*, *Jak2* and *Stat3*) and NFκB (*Chuk*, *Nfkb1* and *Relb*) pathway components, as well as NFκB target genes such as *C1s*, *Vcan*, *Has2*, *Mt2* and *Il4ra* (Fig. 2c and Supplementary Fig. 2e, f). Indeed, gene set enrichment analysis (GSEA) revealed that 3 W co-culture is associated with an enrichment of "Inflammatory Signalling", "TNFα Signalling via NFκB pathway" and "IL6-JAK-STAT pathway" in PSCs, whereas "Epithelial–Mesenchymal Transition" (EMT) and "Oxidative Phosphorylation" pathways are enriched in 2W-PSCs (Fig. 2d and Supplementary Fig. 2f). These results are consistent with the previously described iCAF subsets in PDA[27,29,30,33], and suggest that heterocellular interactions influence fibroblast subtypes. To determine whether the PSC subtype switch is consistent across 2 and 3 W co-cultures with different PCCs, we collected conditioned medium from 2 W PCC–PSC and 3 W PCC–PSC–MØ co-cultures using three different PCCs, and analysed the expression of selected inflammatory (*Il6*, *Il4ra* and *Cxcl1*) and myofibroblastic (*Acta2*) markers in PSCs stimulated with the conditioned medium (Fig. 2e). Indeed, 2 and 3 W PSCs displayed a transcriptional change consistent with subtype switch across all PCC (Fig. 2e, left). Whereas *Acta2* was more highly expressed across all 2 W conditions, 3 W conditioned medium increased inflammatory gene expression and decreased *Acta2* (Fig. 2e, right). Moreover, to assess whether the PSC subtype switch was specific to 3 W co-culture conditions, and not due to direct interactions between PSCs and MØs, we compared gene expression in PSCs treated with conditioned medium from PCCs, PSCs and MØs in mono-2W- and 3W-co-cultures. This data confirmed that only in the presence of all three cell types did PSCs decrease *Acta2* expression and increase *Il6*, *Il4ra* and *Cxcl1* expression (Fig. 2f).

To then confirm the co-culture-dependent switch in fibroblast function using an orthogonal readout, we profiled the expression of 26 mesenchymal markers by mass cytometry (MC) in PSCs grown as mono-culture or in 2W- or 3W- co-cultures

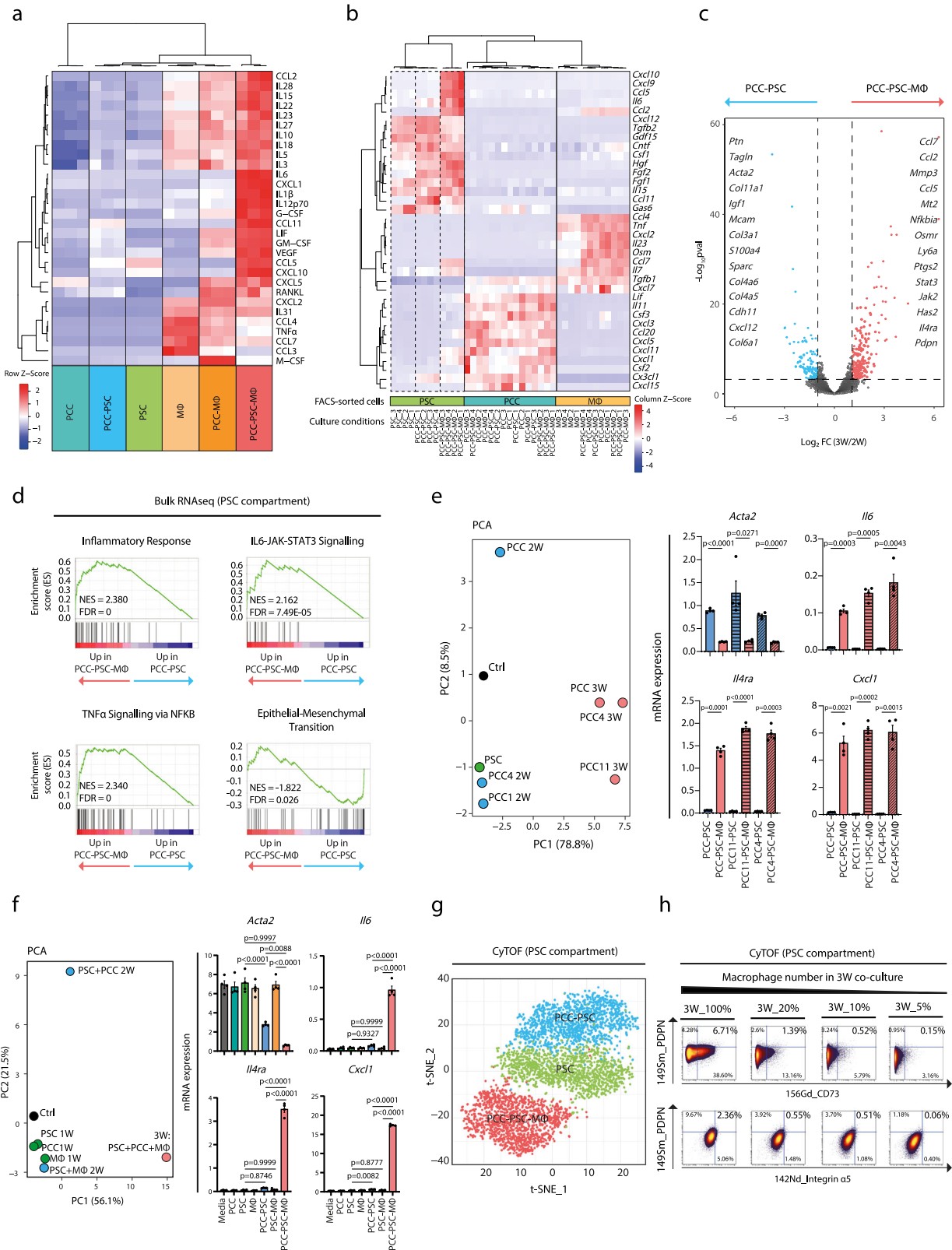

(Supplementary Fig. 2g). This analysis demonstrated a shift in marker levels between the culture conditions (Fig. 2g, Supplementary Fig. 2h), supporting a PSC phenotypic switch between PCC–PSC and PCC–PSC–MØ co-cultures. To confirm the role of MØs, we set up a MØ-limiting titration experiment, where we decreased the number of MØs in 3 W co-culture, and quantified mesenchymal marker expression by MC (Fig. 2h). This

demonstrated that markers preferentially expressed in 3W-PSCs, such as PDPN, CD73 and Integrin α5, decreased as the number of MØs was reduced in the 3 W co-culture (Fig. 2h).

**Heterocellular OSM–OSMR signalling induces inflammatory fibroblasts.** Consistent with the in vivo gene expression analysis (Fig. 1c, h), *Osm* and *OsmR* is differentially expressed in vitro

**Fig. 2 Macrophage–tumour cell interactions induce inflammatory fibroblasts in vitro. a** Heatmap showing relative levels of soluble signals quantified by LUMINEX, clustered with unsupervised hierarchical clustering ($n = 3$ per culture condition). **b** Heatmap of relative expression levels of soluble signals measured by RT-qPCR, clustered with unsupervised hierarchical clustering ($n = 4$ per culture condition). **c** Volcano plot showing differentially regulated genes (DEGs) of PSCs isolated from PCC–PSC ($n = 4$) and PCC-PSC-MØ ($n = 4$) co-cultures. 3 W: PCC-PSC-MØ co-culture. 2 W: PCC-PSC co-culture. FC fold change. **d** Gene set enrichment analysis (GSEA) of RNAseq data from (**c**). NES normalised enrichment score, FDR false discovery rate. **e** RT-qPCR assay of selected myofibroblastic and inflammatory genes of PSCs stimulated with conditioned medium from 2 or 3 W co-cultures using different isolations of PCCs (PCC, PCC4 and PCC11). *Left*: Changes in expression state visualised by principle component analysis (PCA) across experimental conditions ($n = 4$). *Right*: Representative mRNA expression of inflammatory (*Il6, Il4ra* and *Cxcl1*) and myofibroblastic (*Acta2*) genes. Results displayed as mean ± SEM, $n = 4$, one-way ANOVA Tukey test. **f** RT-qPCR of selected myofibroblastic and inflammatory genes in PSCs stimulated with conditioned medium from 1, 2 or 3 W co-cultures of PCC, PSC and bone marrow-derived MØs. *Left*: Changes in expression state visualised by PCA plot across experimental conditions ($n = 4$). *Right*: Representative mRNA expression of inflammatory (*Il6, Il4ra* and *Cxcl1*) and myofibroblastic (*Acta2*) genes. Results displayed as mean ± SEM, $n = 4$, one-way ANOVA Tukey test. **g** Mass cytometry analysis of PSCs in mono- ($n = 3$) and co-culture with PCC ($n = 3$) or PCC and MØs ($n = 3$). Self-organising map clustering (FlowSOM) of PSCs in mono- and co-cultures displayed as *t*-Distributed Stochastic Neighbour Embedding (tSNE). **h** Mass cytometry analysis of PSCs from 3 W PCC-PSC-MØ co-culture. Number of macrophages plated in 3 W co-culture was titrated down and PDPN, CD73 and Integrin α5 expression levels shown as biaxial plots. Also see Supplementary Fig. 2. Source data are provided as a Source Data file.

where MØs express *Osm* and PSCs express *Osmr* (Fig. 3a and Supplementary Fig. 3a). We therefore hypothesised that hetero-cellular OSM–OSMR signalling induces expression of inflammatory mediators in PSCs. To test this hypothesis, we analysed the expression of selected myofibroblastic and inflammatory markers in PSCs treated with recombinant OSM or conditioned medium from mono- and co-cultured PCCs, PSCs and MØs (Fig. 3b). Recombinant OSM and 3 W co-culture conditioned medium induced a transcriptional shift in PSCs distinct from mono- and 2 W PCC–PSC co-cultures, as visualised by PCA (Fig. 3b left). Specifically, recombinant OSM increased expression of inflammatory mediators, including *Il6, Il4ra Cxcl1, Mt2, Nos2* and *Vcan*, (Fig. 3b right). Similarly, markers predominantly expressed in PSCs co-cultured with PCCs, such as *Thbs1, Col4a5, Col4a6, Acta2* and *Ccnd1*, are decreased following OSM stimulation (Fig. 3b right). Several of these genes have previously been associated with inflammatory (iCAF) and myofibroblastic (myCAF) phenotypes, confirming an inverse relationship between iCAF and myCAF transcriptional programmes [27,29,30].

To further elucidate the molecular dependencies of MØ-induced inflammatory gene expression in PSCs, we added OSM blocking or control antibodies to conditioned medium from PCC–PSC–MØ 3 W co-culture and assessed the impact on PSC gene expression (Fig. 3c and Supplementary Fig. 3b). As inflammatory PSCs isolated from 3 W co-cultures were associated with increased JAK-STAT and NFκB pathway activation (Fig. 2d), we also included respective pharmacological inhibitors (Fig. 3c and Supplementary Fig. 3b). As anticipated, and in agreement with previous observations[30], inhibition of JAK2 and IKKβ reduced the expression of inflammatory mediators while simultaneously increasing expression of myofibroblastic genes. This resulted in a transcriptional state more similar to 2W-PSCs (Fig. 3c and Supplementary Fig. 3b). Similarly, inclusion of an OSM blocking antibody in 3 W conditioned medium also induced a transcriptional shift in the PSCs, marked by decreased expression of inflammatory mediators and increased expression of myofibroblastic genes (Fig. 3c and Supplementary Fig. 3b). The effect of the OSM blocking antibody as well as the JAK-STAT and NFκB pathway inhibitors was replicated using CRISPR–Cas9 mediated knockout of *Osmr* in PSCs (Fig. 3c and Supplementary Fig. 3b), confirming a role for OSM–OSMR signalling in inducing inflammatory gene expression in PSCs. Notably, inclusion of neutralising antibodies against previously described regulators of inflammatory fibroblasts, such as TNFα, IL1α and IL1β[30], differentially influences fibroinflammatory gene expression when compared to OSM–OSMR-targeting modalities (Supplementary Fig. 3c, d). For example, the OSM blocking antibody decreased expression of inflammatory genes such as *Il6, Il4ra, Jak2, Has2,*

*Vcan* and *Nos2*, and increased myofibroblastic genes such as *Thbs1* and *Ccnd1* (Supplementary Fig. 3c). However, neutralising antibodies for OSM, TNFα and IL1β were needed in combination ("OTI") to efficiently block *Mmp3, Pdpn* and *Nt5e* (Supplementary Fig. 3c). In addition, an IL1α neutralising antibody did not alter 3W-PSC gene expression differentially to an IL1β neutralising antibody (Supplementary Fig. 3d). This suggests that MØ-induced inflammatory genes in fibroblasts relies on multiple combinatorial signals. Together, this demonstrates a requirement of heterocellular OSM–OSMR signalling as a potent inducer of inflammatory fibroblasts.

**Tumour cell-secreted GM-CSF increases MØ secretion of OSM.** To identify putative *Osm* inducing signals, we treated MØs with PCC conditioned medium and determined OSM levels in conditioned medium by ELISA (Fig. 4a). None of the tested PCC cell lines produced OSM in mono-culture, however, conditioned medium from all 13 PCC lines increased OSM levels in MØs (Fig. 4a). To subsequently determine which PCC-derived signals induce OSM, we treated MØs with recombinant signals identified in the LUMINEX analysis of PCC conditioned medium (Fig. 2a), and measured OSM in the conditioned medium by ELISA (Supplementary Fig. 4). Strikingly, GM-CSF was the only tested soluble factor that induced OSM secretion (Supplementary Fig. 4). Moreover, increased levels of GM-CSF were generally observed in the conditioned medium of PCCs that strongly induced OSM (Fig. 4b). To functionally confirm a role of PCC-secreted GM-CSF, we collected PCC conditioned medium and included control or GM-CSF neutralising antibodies before stimulating MØs and measuring OSM by ELISA. GM-CSF neutralising antibodies blocked OSM secretion from MØs when added to PCC conditioned medium and recombinant GM-CSF (Fig. 4c), confirming that PCC-produced GM-CSF is a key regulator of OSM secretion. We then sought to determine whether GM-CSF could substitute PCCs in 3 W co-cultures to induce expression of inflammatory signals in PSC and added recombinant GM-CSF to PSC–MØ co-cultures and stimulated PSCs with the conditioned medium. Recombinant GM-CSF–PSC–MØ cultures induced both *Osmr* and inflammatory gene expression in PSCs (Fig. 4d). Together this data supports a role of GM-CSF in inducing *Osm* expression in MØs, thereby enabling MØ-induced inflammatory gene expression in fibroblasts. Finally, analysis of TCGA PanCancer PAAD dataset identified a positive correlation of *OSM* expression levels and *CSF2* (encoding GM-CSF) expression in PDA patients (Fig. 4e), highlighting a clinical association between GM-CSF and OSM expression in PDA. Collectively, these findings imply GM-CSF as a positive regulator of MØ OSM secretion.

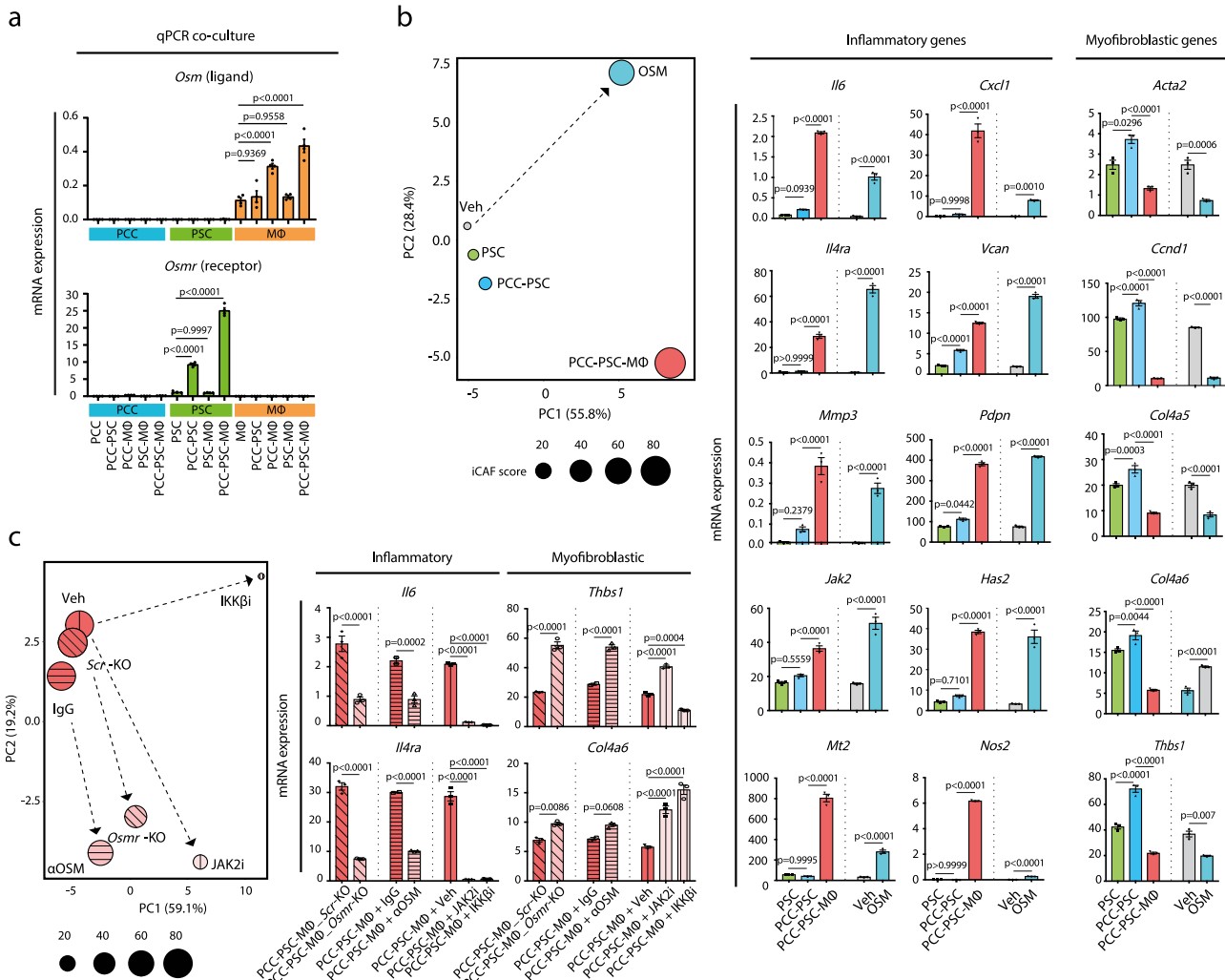

**Fig. 3 Heterocellular OSM–OSMR signalling induces inflammatory fibroblasts. a** Cell-type-specific mRNA expression of *Osm* and *Osmr* across different culture conditions in vitro. Results displayed as mean ± SEM, $n = 4$, one-way ANOVA Tukey test. **b** RT-qPCR assay of myofibroblastic and inflammatory genes. PSCs were treated with recombinant OSM, conditioned medium or vehicle control, and gene expression analysis was performed. *Left*: Changes in expression state visualised by PCA. iCAF score is a sum of mean $z$-values of 15 selected inflammatory genes. *Right*: Representative mRNA expression of inflammatory and myofibroblastic genes of PSCs treated as indicated. Results displayed as mean ± SEM, $n = 3$, one-way ANOVA Tukey test. **c** RT-qPCR analysis of myofibroblastic and inflammatory genes in PSCs following pharmacological inhibitors and CRISPR-Cas9-mediated *Osmr* knockout (*Osmr*-KO) in PSCs. For inhibitor perturbation, PSCs were stimulated with conditioned medium supplemented with vehicle control, IKKβ inhibitor (TPCA-1, 1 μM), JAK2 inhibitor (AZD1480, 2.5 μM), control IgG antibody (2.5 ng/mL) or neutralising OSM antibody (αOSM, 2.5 ng/mL). For *Osmr* knockout, scrambled (*Scr*) control and *Osmr*-KO fibroblasts were stimulated with PCC–PSC–MØ conditioned medium and gene expression was analysed by RT-qPCR. *Left*: Changes in expression state visualised by a PCA plot. iCAF score is a sum of mean $z$-values of 15 selected inflammatory genes. *Right*: Representative mRNA expression of inflammatory (*Il6* and *Il4ra*) and myofibroblastic (*Thbs1* and *Col4a6*) genes of PSCs treated as indicated. Results displayed as mean ± SEM, $n = 3$, one-way ANOVA Tukey test. Also see Supplementary Fig. 3. Source data are provided as a Source Data file.

**OSM supports the formation of an immunosuppressive microenvironment.** We next sought to determine the role of OSM in vivo and analysed the microenvironment of tumours from PCCs[38] orthotopically implanted into syngeneic wildtype or *Osm*-deficient C57BL/6 mice (*Osm*$^{-/-}$) (Fig. 5a). First, we quantified a number of inflammatory cytokines with known tumour-promoting effects in PDA. Tumours from *Osm*$^{-/-}$ animals exhibited markedly reduced levels of several cytokines, including IL6, CXCL1, GM-CSF and TNFα, highlighting a profound change in the tumour environment (Fig. 5b). Immunohistochemical (IHC) analysis of the prototypical myCAF marker, αSMA (*Acta2*), revealed an increase in the αSMA$^{pos}$ area in *Osm*$^{-/-}$ animals compared to wildtype animals (Fig. 5c).

However, fibrillar collagens, as determined by picrosirius red staining, was not significantly different between tumours growing in wildtype or *Osm*$^{-/-}$ animals (Supplementary Fig. 5a). FACS analysis demonstrated a similar levels of CAF infiltration in wildtype and *Osm*$^{-/-}$ tumours (Fig. 5d right, Supplementary Fig. 5b), suggesting that the altered αSMA staining is due to a phenotypic switch from iCAFs to myCAFs. This is further supported by gene expression analysis, where CAFs isolated from *Osm*$^{-/-}$ tumours exhibited decreased inflammatory gene expression (e.g. *Il6*, *Mmp3*, *Has2* and *Osmr*) but increased myofibroblast gene expression (e.g. *Acta2*, *Itgav* and *Fn1*) (Fig. 5d left). Together, these results demonstrate a role of OSM in establishing the characteristic inflammatory microenvironment of PDA.

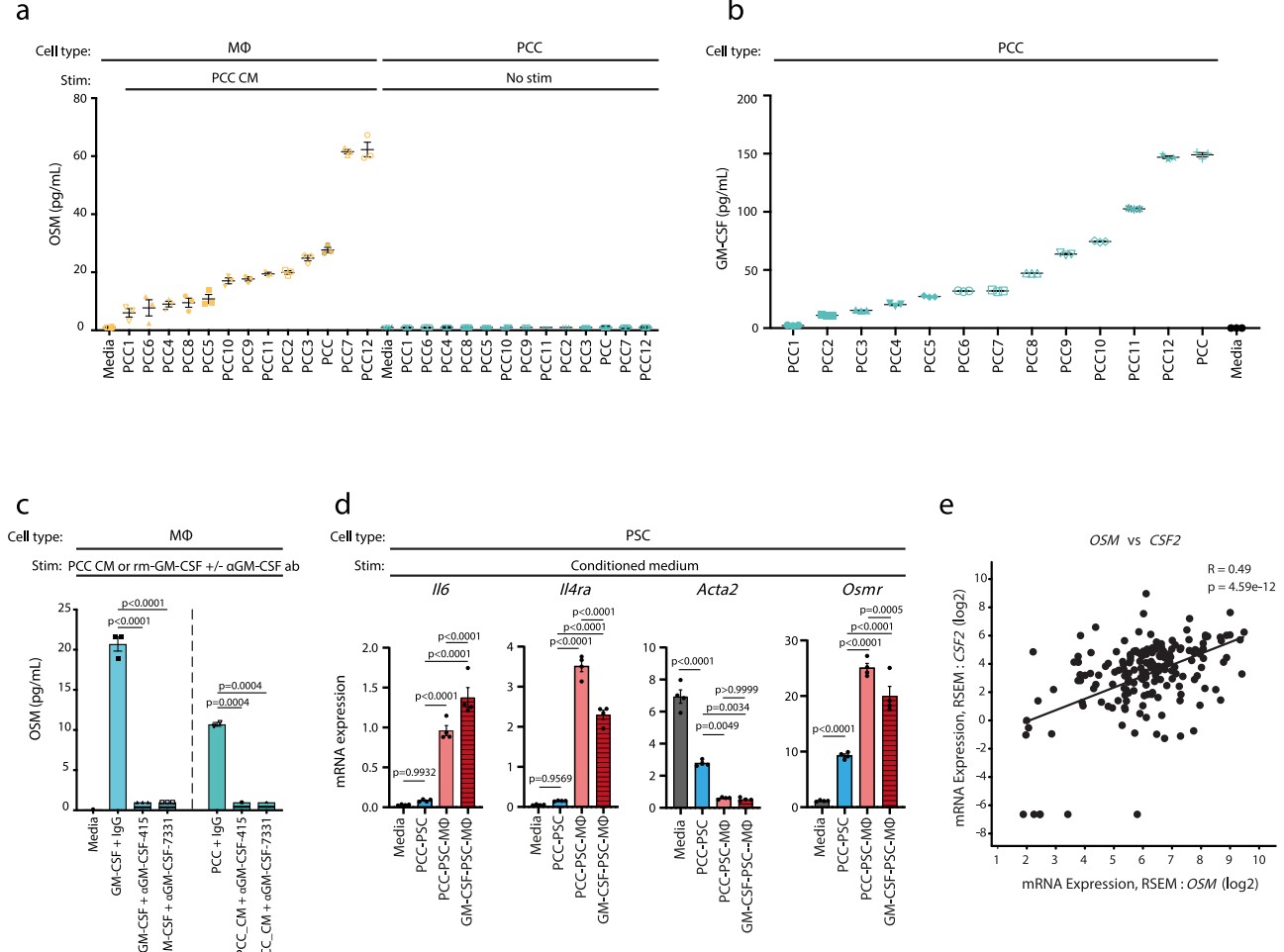

**Fig. 4 Tumour cell-secreted GM-CSF increases macrophage secretion of OSM. a** Quantification of OSM levels in conditioned medium from bone-marrow-derived MØs stimulated with conditioned medium from PCC lines. Results displayed as mean ± SEM, $n = 3$. CM conditioned medium.
**b** Quantification of GM-CSF levels in conditioned medium of PCC cell lines. Results displayed as mean ± SEM, $n = 3$. **c** Quantification of OSM levels in conditioned medium of bone-marrow-derived MØs stimulated with either recombinant murine GM-CSF (rm-GM-CSF) or PCC conditioned medium in the presence of neutralising GM-CSF antibodies (αGM-CSF-415, 1 μg/mL and αGM-CSF-7331, 1 μg/mL) or control. Results displayed as mean ± SEM ($n = 3$), one-way ANOVA Tukey test. **d** RT-qPCR assay of selected genes in PSCs stimulated with conditioned medium from 2 W (PCC–PSC) or 3 W co-cultures (PCC–PSCs–bone marrow-derived MØs) where recombinant murine GM-CSF substitutes for PCCs in the 3 W culture. Results displayed as mean ± SEM ($n = 4$), one-way ANOVA Tukey test. **e** Scatter plot of *OSM* (x-axis) and *CSF2* (y-axis) expression levels in PDA patients (TCGA PAAD dataset, $n = 179$, two-tailed Spearman correlation test). RSEM RNA-seq by expectation-maximisation. Also see Supplementary Fig. 4. Source data are provided as a Source Data file.

We subsequently stained tumours from wildtype and $Osm^{-/-}$ animals with the T cell marker CD8. Although the total CD8 staining was similar between wildtype and $Osm^{-/-}$ tumours, CD8 T cells infiltrated more efficiently into $Osm^{-/-}$ tumours (Fig. 5e and Supplementary Fig. 5c). We then immunopheno-typed wildtype and $Osm^{-/-}$ tumours[21]. Single-cell dissociated tumours from wildtype ($n = 5$) and $Osm^{-/-}$ (n = 5) animals were barcoded, pooled and stained with 40 selected markers to annotate Myeloid, NK and B cells (M/N/B), after which samples were subjected to MC. CD45$^{pos}$/CD3$^{neg}$ immune cells were manually gated, subjected to self-organising map clustering by flow cytometry (FlowSOM), using 25,000 cells from each wildtype ($n = 5$) and $Osm^{-/-}$ ($n = 5$) tumours (Supplementary Fig. 6a), and visualised by uniform manifold approximation and projection (UMAP) (Supplementary Fig. 6b)[39,40]. This analysis identified 25 populations, including natural killer (NK) cells, B cells and several myeloid populations, such as MØs, monocytes, dendritic cells, neutrophils and eosinophils (Supplementary Fig. 6a, b). Intriguingly, we observed an inverse MØ–monocyte

ratio between wildtype and $Osm^{-/-}$ tumours (Supplementary Fig. 6c), where $Osm^{-/-}$ tumours contain decreased number of MØs but an increased number of monocytes (Supplementary Fig. 6d). Specifically, the MØ subset (M/N/B_C10), which express markers consistent with an immunosuppressive tumour-associated MØ (TAM) phenotype (CD11b$^{pos}$; F4/80$^{pos}$; MHCII$^{high}$)[9], was significantly decreased in $Osm^{-/-}$ tumours (Supplementary Fig. 6a and e). In contrast, the classical monocytes and TAM precursors[41,42], M/N/B_C19 (CD11b$^{pos}$; Ly6C$^{pos}$) was more abundant in $Osm^{-/-}$ tumours (Supplementary Fig. 6a, e). This implies that the differentiation of classical monocytes into immunosuppressive TAMs is impaired in $Osm^{-/-}$ tumours. Moreover, antigen presenting cells (APCs) from $Osm^{-/-}$ tumours expressed higher levels of co-stimulatory molecules such as CD40 and CD86 (Fig. 5f). This shows that immune cells in $Osm^{-/-}$ tumours exhibit characteristics associated with improved antigen presentation, such as elevated expression of co-stimulatory markers, which may lead to more effective T cell activation.

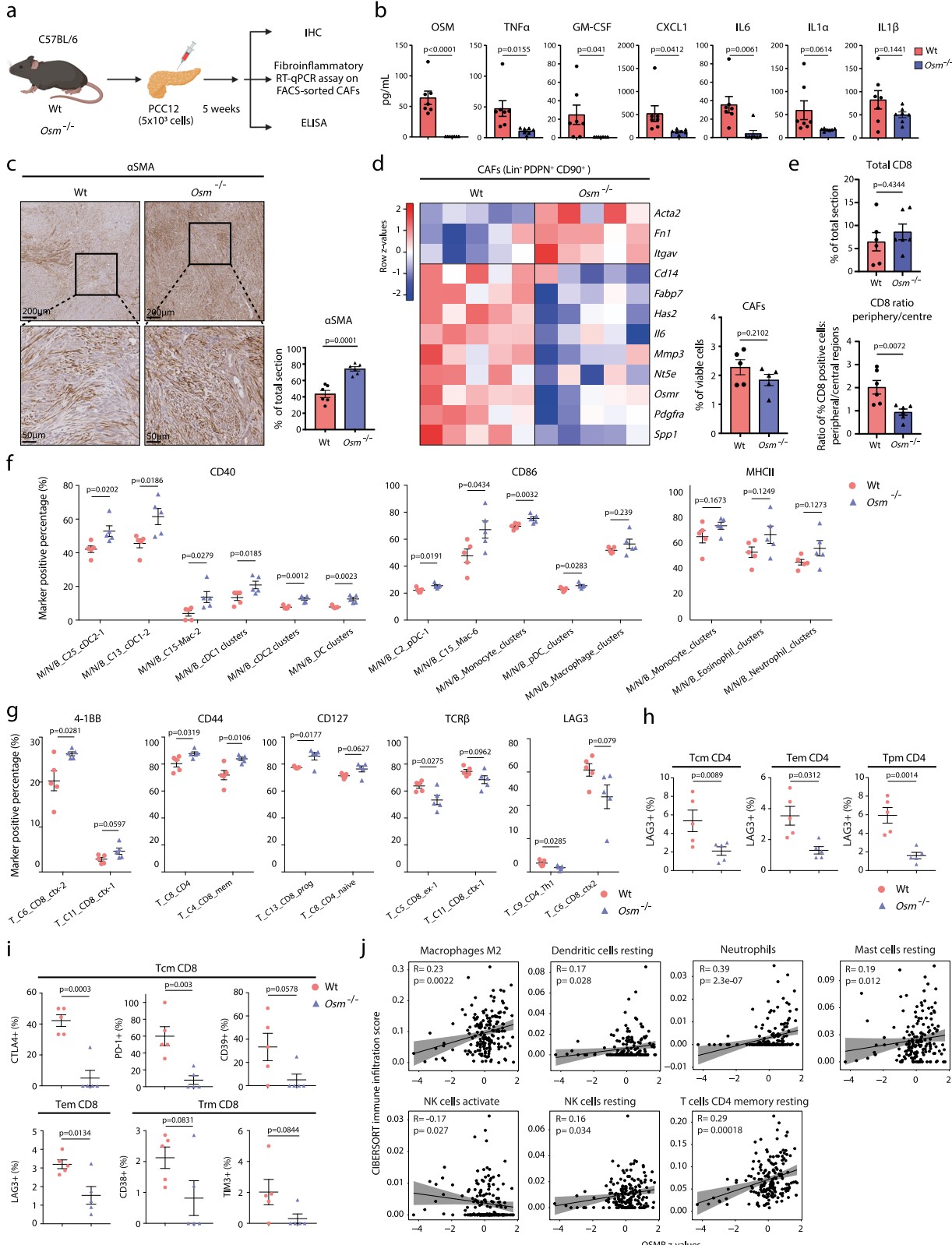

To determine whether $Osm^{-/-}$ tumours also exhibit altered composition and function of CD45$^{pos}$/CD3$^{pos}$ T cells, we immunophenotyped the T cell infiltrate of the same tumours by MC[21]. FlowSOM clustering of 182,800 T cells from wildtype ($n = 5$) and $Osm^{-/-}$ tumours ($n = 5$) identified 20 populations including CD8$^{pos}$ (progenitor, cytotoxic, exhausted and memory), CD4$^{pos}$ (naïve, Th1, Th2, memory and Treg) and γδ T cells

(Supplementary Fig. 6f, g). Importantly, although the abundance of major T cell populations remained unchanged between wildtype and $Osm^{-/-}$ tumours (Supplementary Fig. 6h), we observed a notable shift in the expression of markers consistent with increased T cell activation in $Osm^{-/-}$ tumours (Fig. 5g). Specifically, T cell populations from $Osm^{-/-}$ tumours expressed higher levels of the co-stimulatory molecule 4-1BB (T_C6),

**Fig. 5 OSM supports the formation of an immunosuppressive microenvironment. a** Experimental workflow of in vivo experiments. **b** Protein levels of indicated soluble signals in wildtype ($n = 7$) and $Osm^{-/-}$ ($n = 7$) tumours, measured by ELISA. Results displayed as mean ± SEM, two-tailed student $t$ test. **c** Immunohistochemical analysis of αSMA in wildtype and $Osm^{-/-}$ tumour sections. *Left*: Representative αSMA staining. *Right*: Quantification of αSMA[pos] staining of tumour sections of wildtype ($n = 6$) and $Osm^{-/-}$ ($n = 6$) animals. Results displayed as mean ± SEM, two-tailed student $t$ test. **d** RT-qPCR gene expression analysis of FACS-isolated CAFs from wildtype ($n = 5$) and $Osm^{-/-}$ ($n = 5$) tumours displayed as a heatmap (left). Number of isolated CAFs shown as bar plot (right). Results displayed as mean ± SEM, two-tailed student $t$ test. **e** Immunohistochemical analysis of CD8 in wildtype ($n = 6$) and $Osm^{-/-}$ ($n = 6$) animals. *Upper*: Total CD8[pos] staining. *Lower*: Ratio of peripheral/central CD8[pos] staining using the mean CD8[pos] stain from four representative peripheral regions and four representative central regions per tumour. Results displayed as mean ± SEM, two-tailed student $t$ test. **f** Mass cytometry analysis quantifying abundance of antigen presenting cells (APCs) expressing co-stimulatory markers in wildtype ($n = 5$) and $Osm^{-/-}$ ($n = 5$) tumours. Results displayed as mean ± SEM. Unpaired two-tail student $t$ test. **g** Mass cytometry analysis of CD45$^+$/CD3$^+$ T cells expressing indicated markers in wildtype ($n = 5$) and $Osm^{-/-}$ ($n = 5$) tumours. Results displayed as mean ± SEM, two-tailed student $t$ test. **h, i** Mass cytometry analysis of memory T cells expressing indicated exhaustion/dysfunction markers in wildtype ($n = 5$) and $Osm^{-/-}$ ($n = 5$) tumours. Results displayed as mean ± SEM, two-tailed student $t$ test. Tcm central memory, Tem effector memory, Trm resident memory. **j** Spearman correlation analysis between infiltration of indicated immune cells (determined by CIBERSORT) and *OSMR* expression levels in PDA patients from TCGA PanCancer PAAD dataset ($n = 179$). 95% confidence interval as shaded area. Also see Supplementary Figs. S5 and S6. Source data are provided as a Source Data file.

activation markers such as CD44 (T_C8 and T_C4) and CD127 (T_C13), alongside decreased levels of TCRβ (T_C5) (Fig. 5g), consistent with improved T cell activation[43,44]. Further, the CD4[pos] Th1 population (T_C9) displayed reduced expression of exhaustion/dysfunctional marker, LAG3 (Fig. 5g) suggesting that T cell populations in $Osm^{-/-}$ tumours remain more functional and less exhausted.

CD44 and CD127, which demark T memory cells[45], were increased across several populations of CD8[pos] and CD4[pos] T cells in $Osm^{-/-}$ tumours (Fig. 5g). We therefore further characterised memory T cell populations in these animals. Central memory (Tcm) and effector memory (Tem) CD4[pos] and CD8[pos] T cells displayed reduced levels of T cell exhaustion and inhibitory markers such as LAG3 (Fig. 5h, i), CTLA4 and PD-1 (Fig. 5i), consistent with improved function of memory CD4[pos] and CD8[pos] T cell populations in $Osm^{-/-}$ tumours (Fig. 5h, i and Supplementary Fig. 6i). Analysis of immune infiltrates in human PDA patients from TCGA PanCancer[46] by CIBERSORT[47] was consistent with the MC analysis (Fig. 5j). Specifically, we observed that *OSMR* expression levels, as a marker of iCAFs, correlated with infiltration of M2 MØs and neutrophils and resting or inactivated dendritic cells, mast cells, NK cells and CD4[pos] memory T cells in patients with PDA (Fig. 5j). In contrast, infiltration of activated NK cells was negatively correlated with patient *OSMR* expression (Fig. 5j). Together, this emphasises the relationship between *OSMR* expression and the establishment of an immunosuppressive microenvironment in patients with PDA, and underscores a functional role of heterocellular OSM–OSMR signalling in mediating an immunosuppressive, tumour-promoting environment.

**PCC–PSC–MØ interactions reshape tumour cell signalling.** Given the extensive shift in the cellular signalling milieu between PCC–PSC and PCC–PSC–MØ co-cultures (Fig. 2a), we hypothesised that tumour cell signalling and function is differentially regulated under these two conditions. We, therefore, undertook a quantitative phosphoproteomics analysis of stroma-regulated tumour cell signalling, treating tumour cells with conditioned medium from mono-cultured PCC and 2 W PCC–PSC or 3 W PCC–PSC–MØ co-cultures (Fig. 6a). In agreement with previous observations, tumour–stroma interactions rewired tumour cell signalling to engage additional signalling pathways beyond the tumour cell-autonomous signalling state (Supplementary Fig. 7a)[14,48] with additional differences between 2 and 3 W co-culture conditions. Specifically, we quantified 2523 phosphorylation sites that were differentially engaged in tumour cells treated with co-culture conditioned medium, of which, 3 W signals specifically regulated 508 unique tumour cell phosphosites, and 2 W signals specifically regulated 433 tumour cell

phosphosites (Supplementary Fig. 7a). Kinases and pathways with known function in PDA development, such as MEK-MAPK, JAK-STAT and PI3K-AKT signalling, were more engaged in tumour cells treated with 3 W conditioned medium compared to 2 W, including the phosphorylation of the activation sites of MAPK1 [Y185], MAPK3 [Y205] and STAT3 [Y705] as well as AKT substrate phosphorylation, such as GSK-3β [S9] and EIF4EBP1 (4E-BP1) [S64] (Fig. 6b and Supplementary Fig. 7b). Indeed, enrichment analysis of phosphorylation motifs revealed that 3W-regulated tumour cell signalling was predominantly regulated by the MAPK- and PKA/AKT-signalling pathways (Supplementary Fig. 7c). Concordantly, a kinase-substrate directed phospho-network analysis further highlighted MAPK1/3 and AKT as central kinases regulating PCC–PSC–MØ-induced tumour cell signalling (Fig. 6b). For example, PCC–PSC–MØ-activated MAPK1/3 can phosphorylate a number of downstream phosphoproteins, such as RPS6KA1 (RSK1), in turn phosphorylating GSK-3β, and 4E-BP1, which are also regulated by AKT[49,50]. Notably, both SHC1-engaged SRC and gp130-activated JAK2 signals increased the phosphorylation of STAT3 [Y705] (Fig. 6b), a critical step in pancreatic cancer progression[36,51]. Furthermore, PCC-PSC-MØ signalling uniquely results in phosphorylation of p38 and JNK pathway components including TAOK1 and MAP2K7 (MKK7) (Fig. 6b, Supplementary Fig. 7b), implicated in proliferation and invasion in multiple cancers[52,53]. To corroborate the global phosphoproteomics analysis we undertook a targeted phosphorylation analysis, which confirmed that PCC–PSC–MØ signals accentuate STAT3, MAPK1/3, AKT, p38 and JNK activation beyond mono- and 2 W co-culture conditions (Supplementary Fig. 7d). In addition, immunoblotting for phosphorylated STAT3 [Y705] in PCCs treated with conditioned medium across all mono- and co-culture conditions revealed a specific increase only with PCC–PSC–MØ conditioned medium (Fig. 6c). This effect is unlikely to be a direct effect of OSM on PCCs as recombinant OSM only engaged STAT3 phosphorylation in PSCs (Supplementary Fig. 7e). Furthermore, 3 W co-culture induced STAT3 phosphorylation in PCCs was dependent on OSMR expression in PSCs where CRISPR–Cas9-mediated knockout of *Osmr* in PSCs ablate 3 W co-culture induced STAT3 phosphorylation in PCCs (Fig. 6d and Supplementary Fig. 7f). These data underscore the importance of interactions between tumour cells, fibroblasts and macrophages in shaping tumour cell signalling.

Single sample GSEA (ssGSEA) of human PDA tumours adjusted for tumour purity (Fig. 6e and Supplementary Fig. 7g), confirmed differential signalling pathway engagement where *OSMR*[lhigh] tumours display increased KRAS pathway activity, PI3K-AKT-mTOR and IL6-JAK-STAT3 signalling. Moreover,

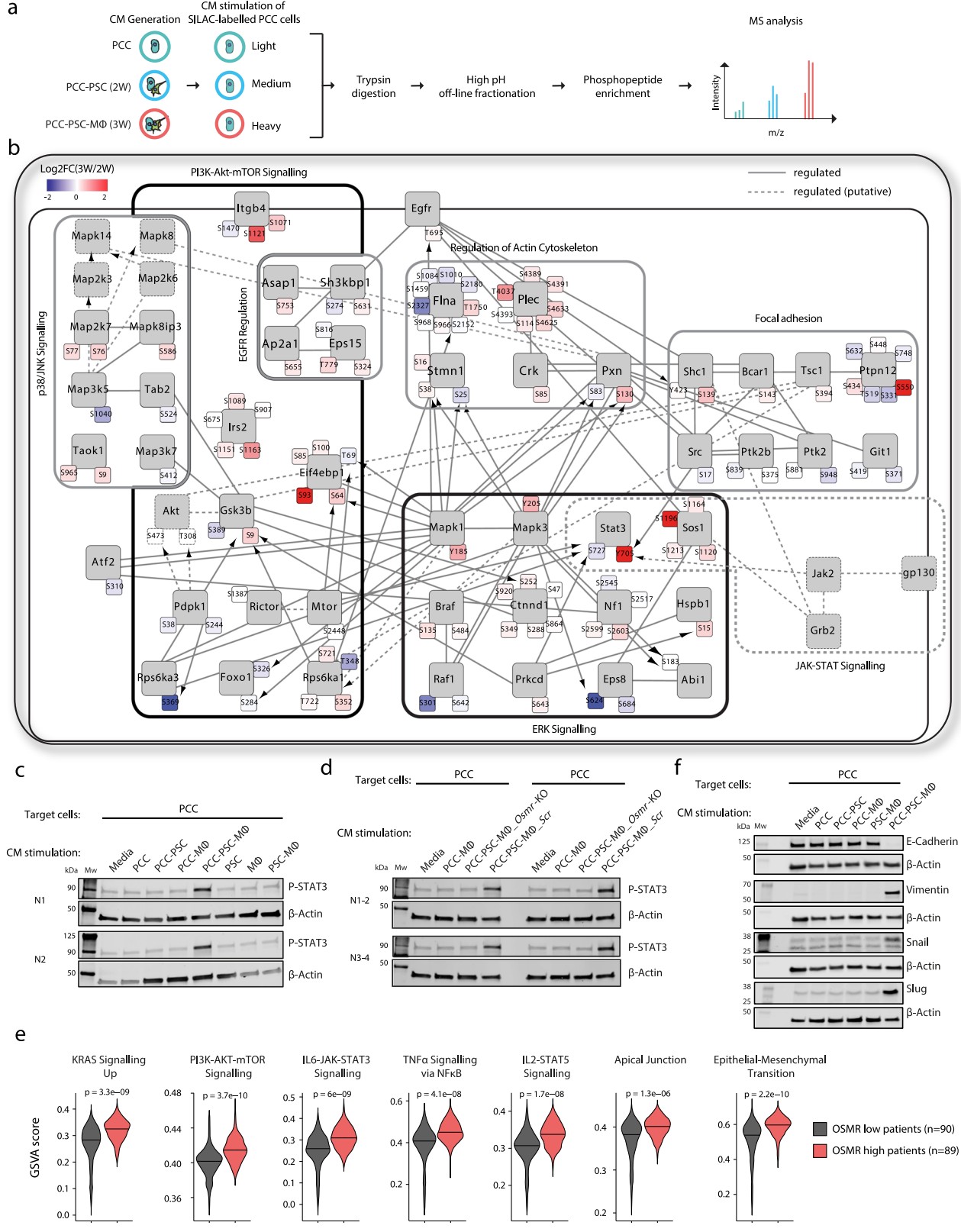

consistent with the observation that PCCs in 3 W co-culture regulated cell–cell adhesion, motility and actin cytoskeleton (Supplementary Fig. 7h), ssGSEA also identified an enrichment of EMT signature genes in *OSMR*[high] tumours (Fig. 6e and Supplementary Fig. 7i). To further validate this observation, we analysed PCCs treated with conditioned medium from 2 and 3 W

co-cultures, which revealed a decrease in E-Cadherin levels concomitant with an increase in VIM and SNAIL/SLUG specifically in PCCs treated with conditioned medium from 3W-cultures (Fig. 6f, and Supplementary 7j). Thus, interactions between tumour cells, fibroblasts and MØs, through OSM–OSMR signalling, induce an inflammatory fibroblast phenotype, which in

**Fig. 6 PCC–PSC–MØ interactions reshape tumour cell signalling. a** Experimental workflow outlining SILAC-based phosphoproteomics analysis of stroma-regulated tumour cell signalling. "Light" (L), "Medium" (M) and "Heavy" (H) labelled PCCs were stimulated (5 min) with PCC or PCC–PSC or PCC–PSC–MØ conditioned medium, respectively ($n = 5$). Pooled L/M/H peptides (in a 1:1:1 ratio) were fractionated, phospho-peptides enriched and analysed by LC-MS/MS. **b** PhosphoPath phospho-network analysis summarising PCC–PSC–MØ-regulated tumour cell signalling, relative to PCC–PSC-regulated signals ($p < 0.05$, Fisher exact test with Benjamini–Hochberg multiple testing correction). Arrows indicate kinase-substrate relationships and blunt-ended edges indicate protein–protein interactions. **c** Western blots of phospho-STAT3 levels in PCCs treated with conditioned medium from 1, 2 or 3 W co-cultures ($n = 2$). **d** Western blots of phospho-STAT3 levels in PCCs treated with conditioned medium from 2 W or 3 W co-cultures containing either *Osmr*-KO or *Scr*-PSCs ($n = 4$). **e** Single-sample gene set enrichment analysis (ssGSEA) of PDA patients from TCGA PanCancer PAAD dataset ($n = 179$). Patients were stratified into OSMR$^{high}$ ($n = 89$) and OSMR$^{low}$ ($n = 90$) groups. **f** Representative western blot of EMT markers in PCCs treated with conditioned medium from 1, 2, or 3 W co-cultures; E-cadherin, Vimentin, SNAIL, and SLUG ($n = 3$). Also see Supplementary Fig. 7. Source data are provided as a Source Data file.

turn engage migratory and mesenchymal tumour cell signalling pathways.

**OSM drives tumour growth and metastasis in vivo.** Lastly, we sought to determine the pathobiological role of OSM–OSMR signalling on tumour cell function in vivo. We analysed tumour growth and metastasis in the wildtype and $Osm^{-/-}$ orthotopic mouse model of PDA used in Fig. 5. Due to the desmoplastic reaction of pancreatic tumours, we transduced PCCs with infrared RFP (iRFP) before implantation, thereby enabling quantitative analysis of tumour cell-specific growth[54]. As expected, tumour cells injected into the pancreas of wildtype animals efficiently established primary tumours, as quantified by iRFP signal intensity, primary tumour volume and tumour weight (Fig. 7a). In contrast, injection of the same cells into $Osm^{-/-}$ mice results in a significantly reduced tumour burden, shown by iRFP intensity, tumour volume and tumour weight (Fig. 7a), demonstrating a systemic role for OSM in primary tumour growth. IHC analysis confirms a reduced PanCK$^{pos}$ tissue area in $Osm^{-/-}$ tumours, corresponding to restrained tumour growth (Fig. 7b). No significant differences were observed in staining for the proliferation marker Ki67, however cleaved caspase 3 levels were increased in tumours from wild-type animals (Fig. 7c and Supplementary Fig. 8a, e). Quantitative morphology analysis of PanCK$^{pos}$ tumour cells highlights a morphological shift from a mixed mesenchymal and epithelial phenotype in wildtype animals towards an epithelial-dominated phenotype in the $Osm^{-/-}$ animals (Fig. 7b). Consistently, IHC staining for the EMT marker SLUG revealed decreased levels in tumours from Osm$^{-/-}$ animals but a non-significant difference in SNAIL levels (Fig. 7c and Supplementary Fig. 8c, d), supportive of an $Osm$-dependent regulation of tumour cell EMT.

Liver is the most common metastatic site for PDA, and metastasis is the major contributor of cancer mortality[55]. Due to the observed difference in EMT signature between 2 and 3 W co-culture conditions in vitro and $OSMR^{high}$ and $OSMR^{low}$ human PDA tumours (Fig. 6e, f), the morphological shift of tumour cells from a more migratory and mesenchymal to an epithelial-dominated phenotype (Fig. 7b), together with reduced levels of pro-metastatic inflammatory signals, such as IL6 (Fig. 5b), we hypothesised that loss of $Osm$ would also be associated with reduced metastasis. Strikingly, whereas more than 50% of wild-type animals develops iRFP$^+$ liver metastasis, none of the $Osm^{-/-}$ animals shows any signs of liver metastasis (Fig. 7d) demonstrating a role of OSM in regulating metastatic spread. Notably, staining for CD31 did not reveal any overt differences in the blood vessel abundance between tumours in wildtype or $Osm^{-/-}$ animals (Fig. 7c and Supplementary Fig. 8b). Taken together, these findings highlight that OSM is a key MØ-derived signal in establishing an inflammatory microenvironment promoting tumour growth and metastasis.

## Discussion

Tumours are evolving multicellular ecosystems. Pancreatic ductal adenocarcinoma (PDA) is characterised by an abundant desmoplastic reaction, where CAFs have emerged as coordinators of both tumour-promoting and -restricting functions of the microenvironment[2,22]. Single cell RNA expression and de novo inference analyses have been instrumental in annotating transcriptionally distinct CAF subsets[21,24,26,27,33], however, further characterisation of individual populations and their heterocellular interdependencies is essential in elucidating pathological functions of CAFs. CAFs are highly plastic and readily adopt two interchangeable transcriptional subtypes, where myofibroblastic CAFs (myCAFs) are characterised by high contractility and expression of αSMA (*ACTA2*), whereas inflammatory CAFs (iCAFs) produce pro-tumourigenic cytokines such as IL6, CXCL1 and CCL2[27,29,30]. Whereas genetic and pharmacologic ablation of CAFs, results in rapid progression and metastatic spread[19,20], functional CAF reprogramming induce a microenvironment favourable to chemotherapy and ICI[11,16], highlighting functional opposing roles of CAFs in the microenvironment. Tumour cells have been described to directly modulate CAF function though secretion of SHH, LIF, TGFβ and IL1[14,17,29,30], however, it is less clear whether other cell populations in the TME, beyond the tumour cells, can regulate CAF function.

Here, we describe a role for OSM and its cognate receptor (OSMR) in regulating pancreatic fibroblasts and the inflammatory TME (Fig. 7e). In PDA, *Osmr* is expressed in mesenchymal cells such as CAFs, endothelial cells and pericytes, and *Osm* is expressed in immune cells such as MØs, myeloid cells, dendritic cells and neutrophils. Notably, *Osmr*$^{high}$ CAFs express higher levels of inflammatory signals such as *Il6*, *Ccl7*, *Ccl2* and *Cxcl1* in animal models of PDA, OSMR is expressed at increased levels in human PDA and is associated with a tumour-promoting immune infiltrate and worse patient outcome. In vitro, co-cultures of cancer cells, MØs and fibroblasts as well as recombinant OSM increase *Osmr* expression and production of inflammatory signals in fibroblasts. However, while the presented data indicate a role of heterocellular OSM–OSMR signalling as a regulator of inflammatory fibroblasts, future studies will be critical to establish whether OSM–OSMR signalling directly involving tumour cells and other stromal cell populations play additional roles in PDA.

Reconstituted in vitro co-cultures of tumour cells, stellate cells and MØs activate tumour cell PI3K-AKT-mTOR, MEK-MAPK and EMT pathways. This is consistent with patient data, where high levels of *OSMR* expression is associated with enrichment of pro-tumourigenic pathways including KRAS signalling, IL-JAK-STAT signalling, PI3K-mTOR signalling and EMT. Tumours from orthotopically implanted PCCs are smaller and appear more epithelial in *Osm* deficient ($Osm^{-/-}$) animals. Moreover, no overt metastasis is observed in Osm$^{-/-}$ animals, but further studies are needed to establish a mechanistic link between OSM signalling and metastasis—in particular given an established relationship between primary tumour size and metastatic dissemination. In

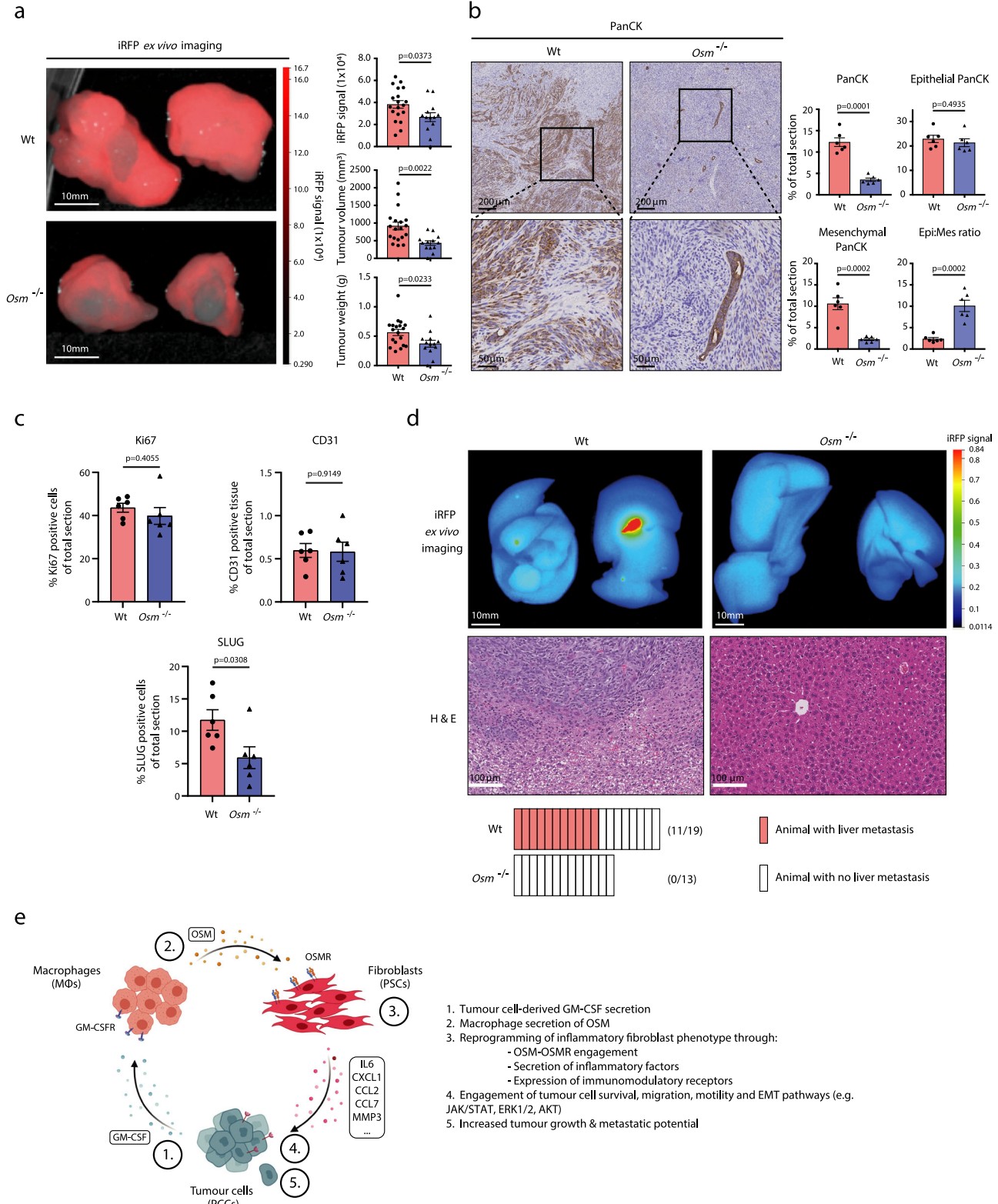

other solid cancers such as breast cancer and squamous cell carcinoma, OSMR is directly expressed by the tumour cells and regulates cancer stem cell properties, EMT and metastasis[56–58]. Importantly, while the analysed scRNAseq data did not identify OSMR expression in epithelial cells, we were unable to apply a consistent cell-type annotation across other available data sets to confirm whether OSMR was expressed in epithelial cells, thus we cannot conclusively exclude that distinct epithelial populations may express OSMR. In situ hybridisation of human PDA demonstrated that OSMR was expressed both in stromal and epithelial cells, however response to recombinant OSM was restricted to pancreatic stellate in vitro. Moreover, OSMR expression in stellate cells is required for co-culture dependent activation of phospho-STAT3 signalling in tumour cells. Thus, although future studies are required to fully establish whether OSM may directly regulate tumour cell function in human PDA,

**Fig. 7 OSM drives tumour growth and metastasis in vivo. a** *Left*: Representative iRFP imaging of resected pancreatic tumours from orthotopically transplanted immune-competent wildtype (Wt) ($n = 19$) and $Osm^{-/-}$ ($n = 13$) mice. *Right*: Quantification of iRFP signal intensity, tumour volume and tumour weight. Results displayed as mean ± SEM (Wt, $n = 19$ and $Osm^{-/-}$ $n = 13$), two-tailed student $t$ test. **b** *Left*: Representative Pan-Cytokeratin (PanCK) staining of wildtype and $Osm^{-/-}$ tumour sections. *Right*: Quantification of total PanCK[pos] staining, mesenchymal PanCK[pos] staining, and epithelial PanCK[pos] staining following morphology analysis, and the epithelial–mesenchymal PanCK[pos] ratio in tumour sections from wildtype ($n = 6$) and $Osm^{-/-}$ ($n = 6$) mice. A shift of PanCK[pos] cells from a mixed mesenchymal and epithelial morphology in wildtype tumours to a more epithelial-dominant morphology in $Osm^{-/-}$ animals. Results displayed as mean ± SEM, two-tailed student $t$ test. **c** Quantification of total Ki67,[pos] CD31[pos], and SLUG[pos] immunohistochemical staining of tumour sections from wildtype ($n = 6$) and $Osm^{-/-}$ ($n = 6$) mice. Results displayed as mean ± SEM, two-tailed student $t$ test. **d** *Upper*: Representative iRFP in vivo imaging (top) and haematoxylin and eosin staining (bottom) of resected livers from wildtype and $Osm^{-/-}$ mice. *Lower*: Quantification of iRFP[pos] liver metastasis incidents in wildtype ($n = 11/19$) and $Osm^{-/-}$ ($n = 0/13$) animals. **e** Model outlining heterocellular OSM–OSMR signalling between pancreatic cancer cells, fibroblasts and macrophages. Pancreatic cancer cell secretion of GM-CSF induces macrophage secretion of OSM, which reprograms fibroblasts to an inflammatory phenotype, in turn accentuating tumour growth and metastasis. Also see Supplementary Fig. 8. Source data are provided as a Source Data file.

the data presented here demonstrates that fibroblast reprogramming and rewiring of tumour cell signalling can regulate tumour cell signalling in a heterocellular manner.

OSM signalling has been associated with pro-inflammatory and anti-fibrotic effects in cardiac fibroblasts[59] and in inflammatory bowel disease where OSM–OSMR signalling and stromal reprogramming drives colitis and mediates resistance to anti-TNFα treatment[60]. Moreover, a correlation between inflammatory fibroblasts and T cell exhaustion and dysfunction has been found in a de novo study of triple negative breast cancer[61]. Loss of OSM function in vivo is associated with a decrease in the level of several inflammatory signals with known pro-tumorigenic function, a shift in the myeloid to MØ infiltration of orthotopic tumours, as well as improved function of APCs and T cells. This suggests that OSM signalling may act to coordinate the inflammatory microenvironment through functional fibroblast reprogramming, and that the role may be conserved across multiple tissues and inflammatory diseases.

Together, the data presented here demonstrates a role for heterocellular OSM–OSMR signalling in establishing a pro-tumourigenic environment in PDA and highlights OSM–OSMR signalling as a putative therapeutic target in PDA.

## Methods

**Mice and in vivo experiments.** To establish an immune-competent mouse line with whole-body depletion of $Osm$ ($Osm^{-/-}$), the cryopreserved sperm from a male C57BL/6 $Osm^{-/-}$ mouse (MMRRC_048921-UCD) was purchased from The KOMP Repository (University of California, Davis, US) and fertilised in vitro with eggs from a female wildtype donor C57BL/6 mouse (purchased from Envigo). The fertilised embryos were implanted into pseudo-pregnant surrogate females. The litters were crossed with age-matched wildtype C57BL/6 mice to produce heterozygotes, which were then crossed further to produce wildtype and $Osm^{-/-}$ breeders. All animals were genotyped and maintained under pathogen-free and ventilated cages with environment enrichment in the Biological Resources Unit at the CRUK Manchester Institute (CRUK MI), and allowed free access to irradiated food and autoclaved water ad libitum in a 12 h light/dark cycle, with room temperature at $21 \pm 2\,°C$ and humidity between 45 and 65%. All animal experiments were reviewed and approved by the Animal Welfare and Ethical Review Body (AWERB) of the CRUK MI and undertaken under Home Office regulations under the Animals (Scientific Procedures) Act 1986 and under Project Licence PPL 70/8745. Experiments are reported in accordance with the Animal Research: Reporting of In Vivo Experiments (ARRIVE) guidelines. A 8- to 12-week-old male and female mice were enroled in approximately equal proportions for all animal experiments.

**Patient samples.** Research samples were obtained from the Manchester Cancer Research Centre (MCRC) Biobank with informed patient consent obtained prior to sample collection. The MCRC Biobank (ethics code: 18/NW/0092) is licensed by the Human Tissue Authority (license number: 30004) and is ethically approved as a research tissue bank by the South Manchester Research Ethics Committee (Ref: 07/H1003/161 + 5). The role of the MCRC Biobank is to distribute samples. For more information see [www.mcrc.manchester.ac.uk/Biobank/Ethics-and-Licensing].

**Cell culture and cell lines.** Mouse PDA cell lines (PCC, PCC1–PCC12) and PSCs used in this study have been previously described[14,38,62]. Mouse MØ cell line is from ATCC (ATCC TIB-71) and was used for all experiments except where bone

marrow-derived MØs are specifically stated. All cells were cultured at 37 °C, 5% $CO_2$ in normal culture media (NCM), consisting of high glucose and pyruvate Dulbecco's Modified Eagle's Medium (DMEM) (ThermoFisher, 41966029) supplemented with 10% v/v heat inactivated foetal bovine serum (FBS) (ThermoFisher, 10500064) and 1% v/v Hyclone Antibiotic/Antimycotic (Cytiva, SV30079.01). PCC (iKras mouse PDA cancer cell line) were cultured in the presence of 1 μg/mL doxycycline (Sigma, D9891). All cell lines, except PSC, were cultured for no more than ten passages to minimise culture-induced phenotypic drift. PSC cells were cultured for no more than five passages and CRISPR edited PSCs no more than nine passages. All cells used in experiments were routinely tested free of *Mycoplasma*. In addition, cells used in vivo were further tested for Mouse Hepatitis Virus (MHV) and *Mycoplasma* 5 days prior to the experiment. To ensure cell surface proteins engaged in signalling are preserved during in vitro experiments, cells were detached using Accutase Solution (Sigma, A6964). All in vitro experiments were performed in Minimal Media (DMEM with 0.5% v/v dialysed FBS (dFBS) (ThermoFisher, 26400044)) to minimise serum-induced signalling. All cell lines used and origin are listed in Supplementary Table 8.

**Conditioned medium generation.** To generate conditioned medium of mono- and co-cultures of PCC, PSC and MØ, a total of $2.5 \times 10^6$ or $7.5 \times 10^6$ cells was plated in either a T25 or T75 flask, respectively, and cultured in Minimal Media for 72 h. Conditioned medium from these mono- and co-cultures were passed through 0.22 μm syringe filters (VWR), and were either used fresh for stimulation experiments or stored at −80 °C until used for ELISA/LUMINEX experiments.

**Stimulation of target cells with conditioned medium or recombinant protein or inhibitors.** Target cells were plated and serum starved in Minimal Media 18 h prior to stimulation. On the day of stimulation, media was aspirated and the target cells were stimulated with fresh Minimal Media containing either conditioned media in a 1:1 ratio (v/v) or indicated concentrations of recombinant protein or inhibitors, and cultured for 72 h. A list of recombinant proteins, inhibitors, and neutralising antibodies can be found in Supplementary Table 9.

**Bone marrow-derived MØ generation.** Bone marrow-derived MØs (BMDMs) were generated as follows. Femurs and tibias were harvested from 8- to 10-week-old C57BL/6 mice (Envigo). Bones were washed in 70% ethanol, followed by two subsequent washes in ice-cold PBS. Epiphyses of the femurs and tibias were cut off with a disposable scalpel, and bone marrow was flushed out with 10 mL NCM using a 25 gauge needle, and was filtered through a 70 μm cell strainer. Bone marrow cells were washed and cultured in BMDM media (NCM supplemented with 20 ng/mL M-CSF) for 3 days. Media was replaced after 3 days with fresh BMDM media and cells were further cultured for 3 additional days. The resulting cells are bone-marrow-derived MØs and used for experiments.

**LUMINEX assay.** Custom magnetic beads (ThermoFisher) were used to measure the concentrations of a panel of murine CCL2, CCL3, CCL4, CCL5, CCL7, CCL11, CXCL1, CXCL2, CXCL5, CXCL10, IL-1β, IL-3, IL-5, IL-6, IL-10, IL-12p70, IL-18, IL-22, IL-23, IL-27, IL-28, IL-31, M-CSF, GM-CSF, G-CSF, LIF, VEGF, RANKL and TNFα in conditioned media collected from mono- and co-cultures of PCC, PSC and MØ on MAGPIX (ThermoFisher) as per manufacturer's protocol.

**ELISA.** Concentrations of OSM (DY495-05), GM-CSF (DY415-05), IL6 (DY406-05), IL1α (DY400-05), IL1β (DY401-05), TNFα (DY410-05), CXCL1 (DY453-05) (all from R & D Systems) in samples were measured using ELISA DuoSet kits as per manufacturer's instructions. For tumour samples, ELISA was performed as previously described[63]. In brief, tumours were homogenised in PBS containing Protease Inhibitor Cocktail (Sigma, P8340) and PMSF (1 mM). Tumour homogenates were then centrifuged at $14,000 \times g$ for 30 min at 4 °C and supernatants were stored at −80 °C until required for ELISA assay. Following quantification of

total protein concentration in each tumour sample using Pierce BCA Assay Kit (ThermoFisher, 23225), tumour sample containing 100 mg of total protein was then used for ELISA assay.

**Lentivirus transduction and EGFP-, mCherry-, TagBFP-, iRFP-labelling of cell lines.** To generate EGFP-, mCherry- and TagBFP-expressing cell lines, a second-generation lentiviral system was used. 293FT cells (ThermoFisher) were plated in a 10 cm culture dish and cultured overnight in NCM. Next day, 4 µg of SFFV-EGFP or SFFV-mCherry or SFFV-TagBFP transfer plasmid, 2 µg of pCMV delta R8.2 packaging plasmid (Addgene, 12263), 1 µg pMD2.G envelope plasmid (Addgene, 12259) and 21 µL Polyethylenimine (PEI) (Sigma, 764647) were combined in 3:1 (PEI:plasmids) v/v ratio and mixed in 1 mL of serum-free DMEM and incubated at RT for 30 min. Plasmid/PEI mixture was added directly to 293FT cells, and incubated overnight. Virus-containing supernatant was collected and stored at 4 °C and replaced with fresh NCM and cells were cultured for another 24 h. Virus aliquots from both 24 and 48 h collections were combined, passed through a 0.22 µm syringe filter, and stored at −80 °C until required. Target cells were plated at 50% confluence and cultured overnight in NCM. Target cells were incubated in neat virus-containing conditioned media, supplemented with Polybrene (8 µg/mL final concentration) for 24 h. Media was replaced with fresh NCM after several washes with PBS. Cells were cultured until confluence and FACS-isolated for positivity for fluorescent protein expression. Transduced cells were further isolated on FACSAria III Cell Sorter (BD Biosciences, BD FACSDiva Software) to ensure high expression of fluorescent proteins. For iRFP-labelling of PCC12 cell line, lentiviral particles expressing iRFP were generated in 293FT cells using the PGK_iRFP720-WPRE transfer, pMD2.G envelope and psPAX2 packaging plasmids by following the steps shown above. PCC12 cells were labelled with iRFP-expressing viral aliquots as shown above. High iRFP-expressing PCC12 cells were isolated by FACSAria III Cell Sorter. PCC12-iRFP cells were passaged once more and multiple vials were cryopreserved until required for in vivo experiment.

**Biomark HD multiplex qPCR.** Assay primers and probes were designed using the Roche Universal Probe Library Assay Design Centre Tool (https://lifescience.roche.com/en_gb/brands/universal-probe-library.html). Refer to Supplementary Table 7 for a full list of primer sequences and respective TaqMan probe numbers of genes used in this study. Where possible, primer pairs spanning two exons were selected to avoid amplification of genomic DNA. cDNA was synthesised from 500 ng of RNA of each sample using Multiscribe RT enzyme (ThermoFisher) according to manufacturer's instructions. Reverse transcription was performed in a cycle of 25 °C for 10 min, 37 °C for 60 min, 95 °C for 5 min and 4 °C indefinitely until stored at −20 °C. A pre-amplification of the genes of interest was performed on cDNA of each sample in a reaction mixture of 1× TaqMan Pre-Amp Master Mix and a pool of assay-specific primers (5 nM) by temperature cycling at 95 °C for 10 min for 1 cycle, 95 °C for 15 s and 60 °C for 4 min for 14 cycles and resting at 4 °C and stored at −20 °C. Assay Mix was made using 8 µM of each primer pair and 1 µM of the appropriate hydrolysis probe in 1× Assay Loading Reagent (Fluidigm, 85000736). Sample Mix was made by diluting the pre-amplified cDNA in TaqMan Universal PCR Master Mix (Applied Biosystems, 4304437) and GE Sample Loading Reagent (Fluidigm, 85000746) in a 1:1 v/v ratio. Samples combined with Sample Mix and Assay Mix were then carefully loaded onto a 96 × 96 Dynamic Array Chip (Fluidigm, BMK-M-96.96) and analysed according to manufacturer's instructions using standard settings with auto-exposure and ROX as passive reference dye. Raw RT-qPCR data was analysed using the BioMark Real-Time PCR Analysis Software (Fluidigm). Assay dependent thresholds were used to calculate cycle threshold (Ct) values and relative expression calculated as: relative expression = $2^{-\Delta Ct}$ where $\Delta Ct$ = (Ct value of Gene A) − (Geometric mean of Ct values for housekeeping genes: *Gapdh*, *Tubb4a*, *Ppia*, *Tbp*).

**Immunoblotting.** Cells were washed in ice-cold PBS and lysed for 1 min in Protein Lysis Cocktail (PLC) buffer [50 mM Trizma base (Sigma), 150 mM NaCl (Sigma 71376), 1.5 mM MgCl2 (Sigma), 1 mM EDTA (Sigma), 10 mM Sodium pyrophosphate tetrabasic (Sigma), 10% v/v Glycerol (Sigma), 1% v/v Triton X-100 (Sigma)], supplemented with 1 mM sodium orthovanadate (Sigma), Protease Inhibitor Cocktail (1:500, Sigma) and Phosphatase Inhibitor Cocktail 3 (1:100, Sigma). Lysates were cleared by centrifugation (16,000g, 10 min, 4 °C), quantified by BCA Protein Assay (Pierce, Thermo Scientific), and boiled in 6× Laemmli Loading Buffer [415 mM SDS (Sigma), 895 µM Bromophenol blue (Sigma), 50% v/v Glycerol (Sigma), 60 mM Trizma base (Sigma) 600 mM Dithiothreitol (Sigma), dH₂O to 10 mL], for 5 min at 95 °C. Lysates were loaded at 10 µg into pre-cast gels, 4–15%, 15-well, 15 µL (mini-PROTEAN, BIO-RAD), in 1× TGS buffer, and transferred onto 0.2 µm nitrocellulose membranes (Amersham) for 1 h 30, in 1× TGS, 20% MeOH, dH₂O transfer buffer. Membranes were blocked for 1 h in ROTI-Block (Carl-Roth), incubated with primary antibodies (overnight, 4 °C) washed in PBST, and incubated with secondary antibodies (1 h, RT). Membranes were imaged using Odyssey CLx Imaging System (LI-COR Biosciences) for infra-red duplexing of fluorescent signals and quantified using Image Studio Lite software (LI-COR, version 5.2.5). A list of antibodies used and uncropped western blots can be found in the Supplementary Information file (Supplementary Table 5, and Supplementary Figs. 9–12) and the Source Data file.

**CRISPR–Cas9-guided *Osmr* knockout of fibroblasts.** *Osmr* knockout of fibroblasts was achieved by using a multi-gRNA approach (Synthego), where PSC cells were transfected with recombinant Cas9 and three individual gRNAs against *Osmr*, which were designed on https://www.synthego.com/. Specifically, prior to transfection, PSC cells were washed in PBS, detached using Accutase Solution, and washed twice in PBS. Subsequently, $2.5 \times 10^5$ cells were resuspended in 5 µl of P3 Primary Cell Nucleofector™ Solution (Lonza). In parallel, Ribonucleoprotein (RNP) complexes were prepared by combining 4 µL of 30 µM of multi-gRNA mix (Synthego) and 1 µL of 20 µM Alt-R® S.p. HiFi Cas9 Nuclease V3 (Integrated DNA Technologies) in 10 µL of Primary Cell P3 Nucleofector solution (Lonza) and incubated at RT for 20 min. Finally, the cell suspension, RNP mixture and 0.8 µL of 100 µM Electroporation Enhancer Solution (Integrated DNA Technologies) were combined together and mixed gently by pipetting. The RNP–cell mixture was transferred into the Nucleocuvette Strip (Lonza) and transfected using the 4D-Nucleofector Core Unit (Lonza) with the pulse "CM-137". Cells were then transferred into a T75 flask filled with pre-warmed NCM and cultured at 37 °C, 5% CO₂. Double-strand cleavage of *Osmr* gene was confirmed by gel electrophoresis of PCR product using primers flanking the gene-edited region. *Osmr* knockout was confirmed by Sanger sequencing. All the reagents used for nucleofection were from P3 Primary Cell 4D-Nucleofector X Kit S (Lonza, V4XP-3032). Target sequences of gRNAs and Sanger sequencing confirmation are included in the Supplementary Information file (Supplementary Table 4, Supplementary Fig. 13).

**Bulk RNA sequencing of fibroblasts from mono- and co-cultures**

*Cell Culture.* mCherry-labelled fibroblasts (PSC-mCherry) were mono- ($n = 4$) and co-cultured with PCC-EGFP ($n = 4$) or PCC-EGFP and MØ-TagBFP ($n = 4$) in T75 flasks containing Minimal Medium with a total cell number of $7.5 \times 10^6$ cells per condition. After 72 h of culture, cells were gently washed in PBS, detached using Accutase Solution and resuspended in FACS Buffer (1% FBS in PBS) after filtered through 70 µm cell strainer. PSC-mCherry cells from each culture condition were sorted based on mCherry positivity using FACSAria III Cell Sorter (BD Biosciences), and RNA was immediately extracted using RNeasy Plus Mini Kit (Qiagen, 74134).

*RNA sequencing library preparation and analysis.* Indexed PolyA libraries were prepared using the QuantSeq 3′ mRNA-Seq Library Prep Kit FWD for Illumina (Lexogen, 015.96). An input of 100 ng of total RNA was used with 14 cycles of amplification. Libraries were then quantified by qPCR using a Kapa Library Quantification Kit for Illumina (Roche, 07960336001). Single read 75 base-pair (bp) sequencing was performed by clustering 1.9 pM of the equimolar pooled libraries on a NextSeq 500 sequencer (Illumina Inc.). RNA-Seq reads were quality checked using FastQC (http://www.bioinformatics.bbsrc.ac.uk/projects/fastqc/) software and trimming of bases was performed. Raw sequencing reads were then aligned to the mouse genome assembly (GRCm38.75) in single-end mode using STAR 2.7 RNAseq aligner[64] with default parameters. Mapped data were converted to normalised read counts (expression) using *featureCounts*[65] and the Ensembl GTF annotation (Mus_musculus.GRCm38.75.gtf). Differential expression (DE) was evaluated using the DESeq2 Bioconductor package[65]. Genes with relative expression of Benjamini–Hochberg-adjusted *p*-value < 0.05, and Log₂Fold-Change > 1 were considered to be differentially expressed.

**Gene set enrichment analysis (GSEA).** GSEA was performed using the GSEA software (V4.1.0) from the Broad Institute (https://www.gsea-msigdb.org/gsea/index.jsp) and run using default settings, permutation type and gene set[66].

**Orthotopic transplantation.** KPC cells (PCC12, 8513) were prepared for orthotopic injection as follows. Twenty-four hour prior to the transplantation, cells were plated at 70% confluence in normal culture medium (NCM). On the day of injection, cells were washed with ice-cold PBS and dislodged using Accutase Solution. After spinning down at 300 × *g*, 4 °C for 5 min, cells were washed two more times with ice-cold PBS before cell counting. Cell suspension was prepared at 5000 cells in 20 µL ice-cold Matrigel, and was kept on ice until injection. Mice were anaesthetised using isoflurane and subcutaneous injection of Buprenophine (Royal Veterinary College, RVC) (20 µL/10 g body weight, Buprecare) and Carprofen (RVC) (60 µL/30 g body weight, Rimadyl). Prior to surgery, fur was clipped, the exposed skin was cleaned using Chlorhexidine surgical scrub, and the animal was draped for surgery to maintain aseptic conditions. A median laparotomy was performed with a 5 mm incision made to the left abdominal flank. Mouse pancreas was carefully externalised using surgical forceps, and 20 µL of Matrigel cell suspension was injected into the pancreatic tail using an insulin syringe and needle (Braun Omnican 50, VWR). To prevent any leakage at the injection site, the needle was slowly withdrawn and the pancreas was left for 30 s. A successful injection without any leakage was confirmed by a formation of a bubble at the injection site. Muscle and skin layers were sutured with absorbable Vicryl suture (Ethicon). Prior to suturing the skin, Bupivacaine Hydrochloride (1:20 in sterile saline, Marcain Polyamp 0.5%) were administered locally to aid local anaesthesia, and animals were injected subcutaneously with 0.2 mL sterile saline. Mice were weighed twice a week, and were closely monitored for exhibiting any moderate signs of PDA such as swollen abdomen, impaired body conditioning (e.g. reduced mobility),

hyperventilation and loss of body weight by more than 10%. Five weeks after the transplantation, mice were euthanised by cervical dislocation and tissues were collected accordingly for further analyses. Resected tumours were weighed, and tumour volumes were measured by external calliper using the formula, tumour volume = ½(length × width$^2$).

**iRFP in vivo imaging on LI-COR Pearl imager.** Prior to imaging in vivo, mice were sacrificed via cervical dislocation and pancreatic tumours and livers were resected. Tissues were imaged for iRFP excitation (emission wavelength of 685 nm, 700 channel) on Pearl imager using Image Studio V5 software with the default setting for animal imaging as per manufacturer's instructions. All quantifications were performed on Image Studio V5 as per manufacturer's instructions.

**Tumour disaggregation for flow and MC.** Tumours were freshly isolated from wildtype or $Osm^{-/-}$ animals and immediately placed into ice-cold PBS (Fisher Scientific, 10091403), and kept on ice. All non-tumour tissues attached to tumours were thoroughly removed before proceeding further. Tumours were washed once with ice-cold RPMI media and minced into ~1 mm pieces using disposable scalpels in 2 mL of Disaggregation Buffer (DB; 2 mg/mL Collagenase Type IV (Thermo Fisher, 17104019), 1 mg/mL DNase1 (Sigma Aldrich, 10104159001) and 0.5 mg/ mL Hyaluronidase (Sigma Aldrich, H3757) in RPMI). Tumour fragments were transferred into C-Tubes and individually disaggregated using GentleMACS Octo Dissociator (Miltenyi Biotech) on automated "37C_m_TDK1" programme. Once complete, the C-tubes were centrifuged at 100 g for 2 min, and the cell pellets were resuspended with fresh DB, and filtered through a 70 μm strainer. Cells were then washed with 10 mL ice-cold RPMI to quench the digestion. The single-cell sus-pension was pelleted at 300g for 6 min and used for flow and MC. Antibodies used are can be found in Supplementary Tables 1–3 and 6.

**CAF isolation from mouse tumours.** Single-cell dissociation of freshly isolated tumours was performed as described above. Cells were counted for each sample, and CD45$^+$ cells were depleted using mouse CD45 MicroBeads (Miltenyi Biotec, 130-052-301) and LS columns (130-042-401) following manufacturer's recom-mendation. To discern viable cells, CD45$^+$-depleted samples were stained with LIVE/DEAD Near-IR Dead Cell Stain Kit (LD) (ThermoFisher, L10119) following manufacturer's recommendation. Cells were then washed and pelleted and stained with Fc block (BD Biosciences, 558636) for 5 min. A master mix of lineage and fibroblast marker antibodies was then added to each sample and incubated on ice for 30 min. Stained cells were washed in PBS, pelleted and resuspended in FACS Buffer (1% FBS in PBS) and kept on ice until FACS-isolation. Viable CAFs were sorted on BD FACSAria III Cell Sorter based on LD$^-$ CD45$^-$ EpCAM$^-$ CD31$^-$ PDPN$^+$ CD90$^+$ staining profile, and data analysed using Flowjo (BD Biosciences). Sorted CAFs were rapidly lysed and their RNA was extracted using RNeasy Plus Micro Kit (Qiagen, 74034) for RT-qPCR assay.

**MC antibodies and antibody conjugation.** Fibroblast and immune cell (myeloid/ NK/B cell (M/N/B) and T cell) antibody panels used for MC analyses in this study were generated as follows[21]. The dedicated panel builder (https://www.dvssciences.com) was used to estimate isotope and oxide spill-over and guide channel selection. Indicated antibodies were purchased pre-conjugated from Fluidigm, whereas all other antibodies were purchased in a carrier protein-free form and labelled with the indicated heavy-metal tag using Maxpar X8 Antibody Conjugation Kit (Fluidigm) as per manufacturer's recommendations.

**MC barcoding and antibody staining**

*Fibroblast MC analysis.* Prior to setting up in vitro cultures, PCCs and MØs were labelled with EGFP using a lentivirus approach, as described above. PSC mono-culture (n = 3), PCC–PSC (n = 3) and PCC–PSC–MØ (n = 3) co-cultures were prepared as described above. After 72 h of in vitro cultures, 10 μL of 10 mM 5-iodo-2′-deoxyuridine (IdU) (Sigma Aldrich, 17125) solution in 0.2 M NaOH/water was added directly to the media, gently swirled and the cells incubated at 37 °C for 20 min. Cells were then washed in PBS, dislodged using Accutase Solution and quenched with CSM (Cell Staining Buffer), consisting of 5 mg/ml bovine serum albumin (BSA) (Sigma Aldrich, A3294) in PBS. Cells were filtered through a 70 μm cell strainer (BD Biosciences, 340633). Subsequently, cells were resuspended and counted.

*Barcoding.* Totally, 3 × 10$^6$ cells from each culture condition were aliquoted into individual 5 mL polypropylene FACS tubes, pelleted and gentle mixed. A unique pair of lanthanide-conjugated anti-ITGB1 antibody pairs in 50 μL of CSM was then added to each sample. This live-cell barcoding method leveraged the consistently high expression of ITGB1 in fibroblasts, and in this instance a 5-choose-2 scheme (ITGB1-115In, -163Dy, -166Er, -169Tm, -172Yb) was used. Each sample was gently mixed and incubated on ice in the dark for 30 min. Cells from each sample were washed twice with CSM and combined into a 5 mL polypropylene FACS tube.

*Live/dead cell staining.* The pooled cells were resuspended in 300 μL of ice-cold PBS and vortexed well. Totally, 300 μL of 1 μM cisplatin (Fluidigm, 201064) in PBS was

added into the cells, and incubated for 1 min. The staining was quenched with CSM, and cells were pelleted and aspirated. Subsequently, 60 μL of 100 U/mL heparin sodium salt (Sigma Aldrich, H3393) solution in PBS and 3 μL Fc block (BD Biosciences, 558636) was added to the cells. The cell mixture was gently mixed and incubated on ice for 5 min.

*Fluorophore antibody staining.* A master mix of fluorophore-conjugated antibodies in 150 μL CSM was added, gently mixed and incubated on ice in the dark for 45 min. Cells were washed with CSM and pelleted. Extracellular antibody staining. A master mix of 6 equivalents of extracellular targeting, metal-conjugated anti-bodies in 150 μL of CSM was added, gently mixed and incubated on ice in the dark for 45 min. Cells were washed twice with CSM.

*Fixation and permeabilisation.* The cell pellet was resuspended in 100 μL of PBS, vortexed and 2 mL of 1× FOXP3 Fixation Buffer from FOXP3 Fixation/Permea-bilization Kit (Thermo Fisher, 00-5523-00) was added, followed by thorough vortexing. After 30 min incubation at RT, 2 mL of 1× FOXP3 Permeabilization Buffer was added and the cells were pelleted. Additional 2 mL of 1× FOXP3 Per-meabilization Buffer was added and the cells were pelleted. Sixty microlitre of 100 U/mL heparin sodium salt in PBS and 3 μL of Fc block was added and the sample was mixed by gently rocking.

*Intracellular antibody staining.* Intracellular antibody staining. After incubating for 5 min at RT in the dark, a master mix of 6 intracellular targeting, metal-conjugated antibodies in 150 μL CSM was added. The sample was gently mixed and incubated on ice in the dark for 45 min. Cells were washed twice with CSM. The cell pellet was resuspended in 1 mL of PBS and vortexed well. Two mililtre of 4% paraf-ormaldehyde (PFA) (Thermo Fisher, 28908) in PBS was added. The sample was and stored overnight at 4 °C in the dark until the MC analysis following day.

**MC analysis of immune cells in mouse tumours.** Single-cell dissociation of freshly isolated tumours was performed as described above and in ref. [21].

*Live/dead cell staining.* Each of the disaggregated tumour cell pellet was resus-pended in 300 μL of ice-cold PBS, vortexed well and 300 μL of 1 μM 198Pt monoisotopic cisplatin (Fluidigm, 201198) in PBS was added, and incubated for 1 min at RT. The staining was quenched with 20 mL of CSM-E (Cell Staining Buffer—Extracellular) consisting of 5 mg/ml BSA (Sigma Aldrich, A3294), 0.5% v/v FBS (Thermo Fisher, 10270106) and 0.2 mg/mL DNase1 in PBS. The cells were resuspended and counted. Totally, 3 × 10$^6$ cells from each tumour sample were aliquoted into individual 5 mL polypropylene FACS tubes, washed with CSM-E and pelleted. Twenty microlitre of 100 U/mL heparin sodium salt (Sigma Aldrich, H3393) solution in PBS and 1 μL Fc block (BD Biosciences, 558636) was added. The cell mixtures were gently mixed and incubated on ice for 5 min.

*Extracellular staining.* A master mix of extracellular targeting, metal-conjugated antibodies in 50 μL of CSM-E was added, gently mixed, incubated on ice in the dark for 45 min. The cells were washed twice with CSM-E.

*Fixation and permeabilisation.* The cell pellet was resuspended in 100 μL of PBS and vortexed and 1 mL of 1× FOXP3 fixation buffer from FOXP3 Fixation/Per-meabilization Kit (Thermo Fisher, 00-5523-00) was added, followed by thorough vortexing. After 30 min incubation at RT, 2 mL of 1× FOXP3 Permeabilization Buffer was added and the cells were pelleted. The cell pellets were resuspended in 1 mL of 10% v/v DMSO (Sigma Aldrich, D2650) in CSM-I (Cell Staining Buffer—Intracellular), consisting of 5 mg/ml BSA and 0.2 mg/ml sodium azide in PBS, vortexed and stored at −20 °C until required.

*Barcoding and intracellular staining.* Cells were thawed, 2 ml of CSM-I was added into each sample, washed once with 4 mL PBS and pelleted. Each cell pellet was resuspended in a unique barcoding aliquot from the Cell-ID 20-plex Pd Barcoding Kit (Fluidigm, 201060) in 1 mL of cold PBS, vortexed and incubated at RT for 15 min. The mixtures were diluted in 3 mL of CSM-I, pelleted and washed with CSM-I. Each cell pellet was resuspended in 200 μL of 1× FOXP3 permeabilization buffer each, pooled into a 5 mL polypropylene FACS tube and pelleted. For each sample included in the pooled sample, 10 μL of 100 U/mL heparin sodium salt in PBS and 0.5 μL of Fc block was added and the sample gently mixed.

*Intracellular antibody staining.* After incubating for 5 min at RT in the dark, a master mix of intracellular targeting, metal-conjugated antibodies in CSM-I was added. For each sample included in the pooled sample, one equivalent of antibody and 25 μL of CSM-I was used. The sample was gently mixed and incubated on ice in the dark for 45 min. Cells were washed twice CSM-I. The cell pellet was resuspended in 1 mL of PBS and vortexed well. For every individual sample included in the pooled sample, a minimum of 500 μL of 4% Paraformaldehyde (PFA) (Thermo Fisher, 28908) in PBS was added to fix the cells. The fixed sample was vortexed and stored overnight at 4 °C in the dark until the MC analysis following day.

*MC DNA staining and acquisition.* On the day of acquisition, 5 µL of 125 µM of Cell-ID Iridium Intercalator (Fluidigm, 201192A) was added to the cells and vortexed well. After 1 h of incubation at RT, the cells were washed once with PBS, twice in water and resuspended at a concentration of $1 \times 10^6$ cells/mL in 15% EQ Four Element Calibration Beads (Fluidigm, 201078) in water. This was filtered twice through a 70 µm cell strainer and acquired on a Helios Mass Cytometer (Fluidigm), using a Super Sampler (Victorian Airship & Scientific Apparatus LLC).

**MC data processing.** FCS files were normalised for signal-drift during the acquisition run using the in-built Helios normalisation tool (Fluidigm). For *in vitro fibroblast analysis*, individual sample events were deconvoluted using manual gating based on the pre-determined ITGB1 barcode after removing non-fibroblast cell types based on expression of canonical markers and GFP. For *immune cell analysis of mouse tumours*, individual sample events were deconvoluted using the a stand-alone debarcoder[67], with a Mahalanobis distance of 15 and a minimum barcode separation of 0.3. Individual sample FCS files were uploaded to the cloud-based cytometry platform Cytobank (https://www.cytobank.org, Beckmann Coulter) and checked for consistent signal across the entire acquisition period and correct deconvolution. As per standard methods, live cell events were selected based on DNA-191Ir positivity and cisplatin-198Pt negativity. 191Ir+ debris, remaining non-fibroblast cell types, cell doublets and aggregates were removed. M/N/B and T cells were selected by biaxial gating as CD45+ CD3ε- and CD45+ CD3ε+, respectively, and exported from Cytobank as FCS files, which were subsequently loaded into Cytofkit2 (version 2.0.1). To analyse the different cell populations in an unbiased way, cells were clustered using FlowSOM[40] and visualised using UMAP projections[39], exporting cell data with annotated clusters for further downstream analysis. Clusters were assigned to specific immune cell populations based on the expression of respective markers, and clusters belonging to the same cell population were merged. For memory T cells, which were not allocated to a specific cell cluster by FlowSOM, were gated manually on biaxial plots, as shown in Supplementary Fig. 6i.

**Bioinformatic processing of publicly available murine and human PDA scRNAseq datasets.** Two publicly available single-cell RNAseq datasets of murine PDA and one dataset of human PDA were re-analysed in this study. The murine PDA data from KPC GEMM[27] was acquired from Gene Expression Omnibus (GEO) under the accession number GSE129455, whereas the human PDA data[27] was obtained from NCBI dbGaP under the accession number phs001840.v1.p1. Additionally, the mouse PDA data from KPP GEMM[33] was obtained from the ArrayExpress database under accession number E-MTAB-8483. All the subsequent processing and analyses of scRNAseq data described herein were performed using Seurat package (v3.0.2) in R 4.0.0. From the KPC mouse PDA data 11,260 cells expressing 13,813 genes, 14,915 cells expressing 24,392 genes from the KPP mouse PDA, and 21,200 cells expressing 16,267 genes from the human PDA data were selected for downstream analyses. Subsequently, the built-in ScaleData function was used to normalise and scale the data. Using the variance stabilising transformation (vst regime) on the built-in FindVariableFeatures, we identified 2000 most variable genes and applied PCA by RunPCA function. For dimensionality reduction, 50 principal components were subjected to elbow test (ElbowPlot) and "JackStraw" estimation (JackStraw, ScoreJackStraw and JackStrawPlot) for each cell. Subsequently, Seurat's graph-based kNN approach (k-Nearest Neighbours) was applied for cell clustering of top 20 principal components (Dimension 20 and resolution 0.5). Annotation of cell types for each cell was made based on the expression of cell type markers indicated in the original articles. UMAP projection was used for visualisation of the data (RunUMAP). Scatter plot visualisations were done by Seurat's functions DimPlot and FeaturePlot. For violin plot visualisation, Seurat's function VlnPlot was used.

**Ligand–receptor interaction analysis.** Publicly available single-cell RNAseq datasets of murine and human PDA[27] were subjected to ligand–receptor interaction analysis using CellPhoneDB in Python 3.8.5[34]. Only MØ-fibroblast ligand–receptor interactions were tested in this study. To this end, 700 MØs and 700 fibroblasts were extracted from respective datasets and used as input for CellPhoneDB analysis using default settings. For mouse interaction analysis, gene names were replaced with human orthologues using biomaRt package in R 4.0.0. Only the MØ–fibroblast ligand–receptor interactions with $p$-value < 0.05, which simultaneously exhibit MØ–macrophage and fibroblast–fibroblast interactions with $p$-value > 0.05, were selected for further analysis. In addition, only the ligands with "secreted" function were selected for further analyses. Putative interaction pairs were visualised using ggplot2 package in R 4.0.0.

**TCGA analysis.** Comparisons of normalised *OSM* and *OSMR* levels between tumours from TCGA (PanCancer)[68] and their host tissues from GTEx[69] were retrieved from the GEPIA2 web platform (http://gepia2.cancer-pku.cn/#index) and filtered for tumour types with significant differences between normal and tumour tissue (Mantel–Cox test). Clinical information and gene-expression data for human pancreatic cancer (TCGA-PAAD) were downloaded from cBioPortal (accessed: 22/04/2020) (https://www.cbioportal.org/)[70]. *OSMR*, *OSM*, *CSF2* expressions were Z-score normalised, and the cox-proportional hazard regression $p$-value was

determined in median-categorised patients (top 50% vs. bottom 50%) for the TCGA PAAD dataset ($n = 179$). The survival analysis was performed using the survival package (v3.2.3) and Kaplan–Meier survival plots were computed using the *survminer* R-package in R 4.0.0.

**Single-sample ssGSEA.** Single-sample gene-set enrichment (ssGSEA)[71], a modification of the standard GSEA method[66], was performed on RNA expression profiles from the PAAD-TCGA cohort ($n = 179$) for each sample using the GSVA package (v1.36.0)[72] in R 4.0.0 using protein-coding RNAs only. Hallmark gene sets were obtained from MSigDB (accessed: 27/04/2020)[73]. Protein-coding information was retrieved from the PAAD-TCGA cohort using the biomaRt R package (v3.11)[74]. To test for the association between *OSMR* expression levels and Hallmark ssGSEA enrichment independent of tumour purity we created two linear models (LM1: Hallmark[ssGSEA]~OSMR + Tumour_Purity, LM2: Hallmark[ssGSEA]~Tumour_Purity) and tested their likelihood ratio using the R package lmtest (v0.9-37) with a significance level of $p = 0.05$. For tumour, purity estimation, ESTIMATE[75] was utilised and ESTIMATE scores, immune-scores and stromal-scores were downloaded from the ESTIMATE database (https://bioinformatics.mdanderson.org/estimate/, accessed: 07/05/2020).

**In-solution protein digestion for mass spectrometry analysis**

*Cell culture.* PCCs were labelled with either 0/0 K/R ("Light"), +4 Da/+6 Da K/R ("Medium") or +8 Da/+10 Da K/R ("Heavy") SILAC amino acids as previously described[14,76]. "Light"-PCCs were stimulated with conditioned medium of PCC, "Medium"-PCCs were stimulated with PCC–PSC conditioned medium, whereas "Heavy"-PCCs were stimulated with PCC–PSC–MØ conditioned medium for 5 min in a staggered manner to ensure accurate timing of stimulations.

*In-solution protein digestion.* The conditioned medium-stimulated PCC SILAC-labelled cells were immediately washed twice with ice-cold PBS, lysed in TEAB Lysis Buffer (0.5 M TEAB, 0.05% SDS, phosphatase inhibitors) on ice and sonicated for 3 cycles (30 s sonication and 30 s incubation) on ice. Samples were centrifuged at 16,100$g$ for 15 min at 4 °C. Supernatants were collected and protein concentration was determined using Pierce BCA Assay Kit. 766.66 µg of "Light"-, "Medium"- and "Heavy"-labelled PCC lysates (a total of 2.3 mg) were combined in a 1:1:1 ratio. Subsequently, the pooled proteins were reduced with 10 mM TCEP (at RT for 30 min), alkylated with 50 mM iodoacetamide (ThermoFisher) (at RT for 15 min in the dark), and digested with 46 µg of Trypsin (Promega, V5111) at 37 °C for 16 h. Next day, proteins were further digested with 46 µg of Trypsin at 37 °C for 4 h. Peptides were acidified in 5% trifluoroacetic acid (TFA), and dried down on Speedvac Concentrator (ThermoFisher) at 60 °C for 2 h prior to reverse-phase fractionation.

**Reverse phase off-line peptide fractionation.** Peptides were fractionated as previously described[77]. Peptides were separated on a Zorbax Extend-C18 column (4.6 × 150 mm, 3.5 µm, Agilent Technologies) at 250 µL/min using the following gradient profile (minutes: %B); 5:0.5, 20:30, 24:40, 26:75, 29:75, 30:0.5 and 55:0.5. The buffers used were Buffer A (LC-grade water supplemented with 0.1× v/v NH$_4$OH (pH 10.5)) and Buffer B (100% Acetonitrile). The eluent was collected into 96 round-bottom plates and fractions were collected into each well every 15 s. Only fractions in the elution window with traces of high amounts of peptide material were used and all of the fractions were concatenated into 10 final fractions with each containing 200 µg peptides on average. Fractions were dried using Speedvac Concentrator (Thermo Fisher) at 60 °C for 2 h.

**Automated phosphopeptide enrichment.** Phosphopeptides were enriched using the automated phosphopeptide enrichment method as previously described[78]. TiO$_2$ magnetic beads (ReSyn Biosciences, MR-TID005) were prepared as per manufacturer's recommendation. Fractions were resuspended in Wash Buffer 1 (80% ACN, 5% TFA, 1 M glycolic acid) at 1 µg/µL, centrifuged at 16,100 × $g$ for 10 min, and transferred into KingFisher Flex 96 deep well plate (Thermo Fisher, 733-3004). Phosphopeptides were enriched using KingFisher Flex (ThermoFisher) as previously described. Subsequently, phosphopeptides were resuspended in 0.1% formic acid (FA) and desalted using HLB Desalting cartridges (Waters, WAT094225) on SPE vacuum manifold according to manufacturer's instructions. Desalted phosphopeptides were dried down on Speedvac Concentrator at 60 °C for 2 h and stored at −80 °C until required for LC–MS/MS acquisition.

**Data-dependent acquisition (DDA) LC-MS/MS.** Phosphopeptides were analysed using an Ultimate 3000 RSLCnano system (Thermo Scientific) coupled to a LTQ OrbiTrap Velos Pro (Thermo Scientific). Peptides were initially trapped on an Acclaim PepMap (C18, 100 µm × 2 cm) and then separated on Easy-Spray PepMap RSLC C18 colum (75 µM × 50 cm) (Thermo Scientific) over a 156 min gradient from 0 to 98% Buffer B (80% ACN in 0.08% FA) against Buffer A (0.1% FA) at a flow rate of 300 nl/mL. Samples were transferred to mass spectrometer via an Easy-Spray source with temperature set at 50 °C and a source voltage of 2.0 kV. The mass spectrometer was operated in data-dependent acquisition mode with Multi Stage Activation for phosphorylation neutral loss detection. Survey full scan of MS

spectra (335–1800 m/z) were acquired in the Orbitrap with a resolution of 60,000 and FTMS Full AGF Target of 1,000,000. Top 15 method was selected for fragmentation using collision-induced dissociation. The resulting raw files were searched against the SwissProt *Mus musculus* database on Mascot server (Matrix Science; 2016) in Proteome Discoverer 2.1 (ThermoFisher Scientific). Search parameters included a precursor mass tolerance of 10 ppm, fragment mass tolerance of 0.8 Da, peptide modifications of carbamidomethylation (C) as static modification and oxidation (M, P and K) as well as deamination (N, Q) and phosphorylation (S, T, Y) as dynamic modification. A decoy database search was performed to determine the peptide FDR with the Percolator module. Phosphosite localisation probabilities were calculated with PhosphoRS node. A 1% peptide FDR threshold was applied, and peptides were filtered for medium peptide confidence, minimum peptide length of 6, and finally peptides without protein reference were removed. All peptides that exhibited a confidence of less than "high" and with less than two uniquely identified peptides were excluded from further analysis.

**Phosphoproteomics data analysis**. Quantified phosphopeptides used for downstream analysis were filtered by applying the following criteria: (1) peptide quantification in all three SILAC channels (light, medium and heavy), (2) presence of only one unique phosphorylation site per peptide, (3) phosphorylation-site localisation probability of >75%, (4) detection and quantification of each unique phospho-peptide in at least two out of five biological replicates. Phosphopeptide abundances were normalised by the median abundance of the SILAC-channel before a SILAC-ratio was computed. Phosphorylation-motifs were obtained for each phospho-peptide by retrieving the flanking ±6 amino acids around the detected phosphorylation locus by aligning the peptide-sequence to the UniProtKB/Swiss-Prot database (accessed: 10/08/2019). Kinase-recognition motifs were identified by using the PhosphoSitePlus (v6.5.9.3)[79]. MotifAll algorithm by applying a significance threshold of 1e−06 and a support threshold of 0.05 with against background set to automatically selected phosphorylation sites. Kinases were then identified using the MEME-suite CentriMO[80,81] local motif-enrichment algorithm in pre-identified motifs by applying default options against the Eucaryotic Linear Motif (ELM 2018) motif database. For phospho-network analysis and kinase-substrate network visualisation, the Cytoscape software package[82] was utilised by incorporating the PhosphoPath app[75] with default settings by selecting the following databases:(1) PhosphoSitePlus for kinase-substrate (KS) interactions set to human KS interactions, (2) BIOGRID for protein–protein interactions and (3) WikiPathways for pathway information. Pathway-enrichment was conducted by selecting the whole proteome as background and only pathways with a significant enrichment of p-value < 0.05 (Fisher exact test with Benjamini–Hochberg multiple testing correction) were incorporated into the analysis.

**Phospho-antibody array analysis**. For phospho antibody array analysis of tumour cell signalling, PCCs were plated in 6-well dish and cultured in Minimal Media overnight. CM was freshly generated from mono- and co-cultures of PCC, PSC and MØ, as shown above. PCCs were stimulated with respective conditioned medium for 5 min at 37 °C 5% $CO_2$, and the lysates were immediately collected and analysed using the PathScan Intracellular Signalling Array Kit (CST, 7744) as per manufacturer's recommendation. Fluorescent intensities were analysed on Image Studio (LI-COR).

**CIBERSORT analysis**. The CIBERSORT cell estimation method was utilised to estimate immune cell levels in the TME of the PAAD-TCGA cohort as previously described[83,84] using the CIBERSORT R implementation (v1.04). Only absolute CIBERSORT values for immune-cell populations were used. Spearman-based rank correlation of the z-score normalised *OSMR* expression levels and the CIBERSORT immune abundances was chosen.

**Immunohistochemistry, staining quantification and cell morphology analysis**. Immunohistochemistry staining was performed using the BOND RX automated platform (Leica Microsystems). Four micrometre sections of formalin-fixed paraffin-embedded (FFPE) sections were cut and mounted on charged slides. Dewaxing and heat induced epitope retrieval (HIER) of slides was automated on the Bond RX using epitope solution 1 (AR9961) for 20 min at 100 °C. Using the Bond Refine Kit (DS98007) as per manufacturer's instructions, briefly endogenous peroxidase was blocked (hydrogen peroxide, 10 min) followed by a non-specific protein block (casein, 20 min) prior to primary antibody application (15 min). Following buffer washes, the post primary linker was omitted and the labelled polymer was applied (8 min) followed by visualisation with diaminobenzidine (10 min) and nuclei haematoxylin counterstain (5 min). The following antibodies concentrations were used: Pan Cytokeratin (Abcam, 9377) 2 μg/ml. For αSMA (Sigma Aldrich, A5228) 0.4 μg/ml, the Agilent animal research kit (K3954) was used as per manufacturer's instructions. Both endogenous peroxidase (3% hydrogen peroxide, 10 min) and non-specific protein (casein 20 min) (Vector, SP5020) blocks were applied. Following this biotin-labelled primary antibody was applied (30 min), followed by streptavidin-peroxidase (30 min) and visualisation with diaminobenzidine (5 min). Nuclei were then counterstained with Gills haematoxylin (30 s). Following counterstaining, all sections were dehydrated through graded alcohol, cleared in xylene and coverslipped. Heamatoxylin and eosin (H & E)

staining was performed on 4 μm FFPE sections. Briefly, following dewaxing and rehydration, sections were stained with Gills Haematoxylin (ThermoFisher 6765007) for 3 min and following an extended water wash, counterstained with Eosin (ThermoFisher 6766007) for 1 min. Sections were then coverslipped following dehydration in graded alcohol and clearing in xylene. These stained tumour sections were imaged using Leica SCN400 Slide Scanner (Leica). Quantification of positive marker staining of slides was performed using HALO (Indica Labs). Morphology analysis of PanCK$^{pos}$ cells was performed using the Tissue Classifier machine learning algorithm on HALO. A list of antibodies used can be found in Supplementary Table 5.

**In situ hybridisation**. Human *OSMR* LS 2.5 probe (ACD-biotech) was visualised using the RNAscope LS Multiplex Fluorescent kit on the BOND RX automated platform (Leica Microsystems). Totally, 4 μm sections of FFPE samples were cut and mounted on charged slides. Dewaxing and heat induced epitope retrieval of slides was automated on the Bond RX, using Epitope Retrieval Solution 2 (ER2) (Leica Microsystems, AR9640) for 15 min at 98 °C, and Protease 15 min, the probe was visualised using the standard RNAscope IF protocol, TSA amplification was conducted by a specific premixed TSA 570 reagent (Perkin Elmer, FP1488001KT) at 1/200 for 10 min. The slides were then stained on the bench using 10% Casein (Vector) 20 min, followed by the mouse anti-human Vimentin (Agilent, M0725) at 1/100 for 30 mins, and the donkey anti-mouse 647 (Invitrogen, A-31571) at 1/400 for 30 min. After a further 10% Casein (20 min), the second antibody was added, rabbit anti-human PanCK antibody (Abcam, ab9377) at 1/200 for 30 min, followed by the donkey anti-rabbit 488 (Invitrogen, A-11070) at 1/400 for 30 min. Finally, nuclei were counterstained with 0.33 μg/ml 4′,6-diamidino-2-phenylindole (DAPI) (Thermo Fisher, 62248) for 15 min and coverslipped with ProLong Gold Antifade Mountant (Thermo Fisher, P36930). Slides were scanned using a VS120 microscope (Olympus Lifescience) at 20× and analysed using HALO (Indica Labs). A list of antibodies and probe used can be found in Supplementary Table 5.

**Statistics analysis and data presentation**. General graphical visualisation of data and statistical tests were performed using Prism 7 (GraphPad Software Inc) and various packages on R (4.0.0). Heatmaps and unsupervised hierarchical clustering (Ward's linkage, Euclidean) were generated using R package, Heatmap.2. PCA plots were made using prcomp function on R package, ggplot2.

**Reporting summary**. Further information on research design is available in the Nature Research Reporting Summary linked to this article.

## Data availability

Source data are provided with this paper. RNAseq data have been deposited into the Gene Expression Omnibus (GEO) database under accession number GSE161359. Publicly available scRNAseq datasets of human, KPC and KPP PDA were acquired from NCBI dbGaP (phs001840.v1.p1), GEO (GSE129455) and ArrayExpress (E-MTAB-8483), respectively. Publicly available RNAseq dataset of my/i/apCAFs were acquired from GEO (GSE93313). Patient data from TCGA were downloaded from cBioPortal (https://www.cbioportal.org/). Phosphoproteomics mass spectrometry datasets have been deposited to the ProteomeXchange Consortium via the PRIDE partner repository with the dataset identifier PXD022484. The remaining data are available within the Article, Supplementary Information or Source Data file. Source data are provided with this paper.

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

## Acknowledgements

This work was supported by Cancer Research UK Institute Awards A19258, Experimental Medicine Programme Award (A25236) and European Research Council Consolidator Award (ERC-2017-COG 772577). Dr. Angela Lamarca was part-funded by The Christie Charity. The authors would like to acknowledge colleagues at CRUK Manchester Institute Systems Oncology Team, CRUK Manchester Institute core facilities: Flow Cytometry, Molecular Biology Core Facility and Visualisation, Irradiation and Analysis, the Biological Research Unit and Transgenic Breeding Facility. The authors would also like to acknowledge Dr. Dieter Saur, Dr. Ronald DePinho, Dr. Kris Freese, Dr. Raul Urrutia, Dr. Didier Trono and Dr. Tim Somervaille for kindly sharing cell lines and plasmids and Dr. Santiago Zelenay for fruitful discussion and help with figures. Figure panels 5a and 7e were generated with help from Biorender (www.biorender.com). The results shown here are in part based upon data generated by the TCGA Research Network: https://www.cancer.gov/tcga.

## Author contributions

B.L.: Planned and conducted experiments, analysed data and wrote the paper. E.K.J.H.: Planned and conducted experiments, analysed data and wrote the paper. C.R.B.: Conducted experiments and analysed the data. A.K.: Analysed the data. A.B.G., F.H., C.H., J.X., X.Z., T.S. and K.B.: Planned and conducted experiments, provided technical support, analysed the data. A.L., M.M. and J.V. provided the technical support. C.J.: Conceived project, analysed data, provided oversight and wrote the paper.

## Competing interests

The authors declare no competing interests. J.X. is currently an employee of GlaxoSmithKline and B.L. is currently an employee of AstraZeneca, but all the work carried out by J.X. and B.L. were performed during their time at CRUK Manchester Institute, the University of Manchester.
