## [Peer Review File · Nature Communications]

Heterocellular OSM-OSMR signalling reprograms fibroblasts to promote pancreatic cancer growth and metastasisREVIEWER COMMENTS

Reviewer #1 (Remarks to the Author): with expertise in cancer and transcriptomic

The manuscript reveals a novel signaling in pancreatic cancer development that dictates the pro-tumor growth/metastasis and shapes immunosuppression environment in Pancreatic cancer. Overall, the finding is new in PDA. The role of OSM-OSMR signaling in tumor growth and metastasis is mostly convincing except some conflicts of the published results and some confusing result on CAF states that need to be addressed.

1. First, it is advised the authors should carefully examine and demonstrate the OSM and OSMR signal in the published single-cell RNAseq datasets for real PDAC. (Peng et.al. Cell Research 2019 and Lin et.al. Genome Medicine, 2020). How do these datasets support the authors' conclusions? Does OSMR specifically express in CAF only or any other cell populations? These datasets reflect the real situation of PDA than the ex vivo and animal models.

2. Fig.1A is unclear. The rows for the specific 3 gene expressions should be marked. The column labels should be marked.

3. Fig.1B doesn't show Osmr is predominantly expressed to iCAF.

4. As a possible control, why the assays for the 2W co-culture of PSC-MØ were not mentioned at all? I will be still interested to learn what could happen if tumor cell is not available. If GM-CSF from tumor cells drives the secretion of OSM, can we test PSC for inflammatory response in the 2W co-culture of PSC-MØ without PCC but using recombinant GM-CSF medium?

5. What drives the segregation of PSC in the three ex vivo conditions, technical reasons (batch effects) or there are actually three variable phenotypes of CAF? If the authors try to make the same points using Fig. 2B (PSC columns) and Fig. 2D, shouldn't the points of PSC and PSC+PCC be clustered together in the mixed colors, when PCC alone doesn't change PSC? If it does, what has changed?

6. In Fig. 2G and 2H, using one of the PC axes to reflect the inflammatory level of CAF is unusual. The variance other than inflammatory genes could contribute to PC1. Z-score is a statistic defined within the assayed samples. It is also not a great measurement of inflammatory state because there is no clear threshold. The authors should have a more straightforward way to illustrate the inflammatory state of CAFs.

7. The color legend of Fig. 4B for WT vs.Osm^{-/-} is different from other Fig. 4 panels. I believe it was mislabeled.

8. Other than the comparison of metastasis in wild-type vs. Osm^{-/-} mice, the data to correlate the OSM-OSMR signaling and metastasis through PSC-MØ signaling is inadequate. It would be better if the authors at least check the correlation between the OSM-OSMR signal and the EMT signal (the expression of EMT gene set, including SNAI2, CDH2, ZEB1, TWIST1, VIM etc.) in the published datasets. If the co-upregulation is found in the same patient, it would be more convincing. Or let's find out if iCAF signal is high in metastatic tumors. Maybe the recent scRNAseq datasets could help.

Overall, I am convinced for the significant role of the OSM-OSMR signaling in pancreatic cancer. However, I am not very sure it works through the regulation of iCAF-myCAF transformation as OSMR is not specific in iCAF subpopulations or even in CAF. Actually, I am neither quite sure if the inflammatory state of CAF can be convincingly defined by the authors in this paper. They used co-culture and FACS sorting and the follow-up qPCR assay, which takes too long after the CAF leaves the original biological context. It may not reflect the real CAF state in the PDA tissue after all. We need a method to measure the inflammatory state in the tissue on-the-fly.

As I have checked in previous scRNAseq dataset for PDAC tissues, other than CAF, both EMT tumor cells and Endothelial cells show high expression of OSMR, whose role shouldn't be overlooked in the OSM-OSMR signaling.

Reviewer #2 (Remarks to the Author): with expertise in PDAC and macrophages

In this paper, Lee et al studies the formation of different subtypes of cancer associated fibroblast (CAF) in pancreatic cancer. They claim that macrophage derived factors, particularly Oncostatin M (OSM) induces the formation of an highly inflammatory CAF phenotype, which promotes tumour formation and metastatic progression. In a pre-clinical mouse model, genetic ablation of OSM expression drives CAF subtypes towards a more myofibroblastic phenotype, reduces tumour burden and metastasis, and promotes the presence of rather immune-stimulatory immune infiltrates. The paper contains a substantial amount of work. The findings that macrophage-derived OSM promotes iCAF polarisation are novel and interesting and complements previous work from the Tuveson group. The reported experimental data are of high quality and the manuscript is well written.

Following additional analysis would improve the quality of this manuscript:

Main points:

Figure 1: The identification of OSM expression by macrophages and the expression of OSMR by fibroblasts are based on RNA Seq data analysis, but and their expression hasn't been validated in tissue. Thus, the validation of these findings in human/mouse tissue section of PDAC tumours would strengthen the reported cell type-specific expression of these two factors.

Figure 2: Comprehensive analysis of 2- and 3-way co-culture systems using macrophages, pancreatic stellate cells and cancer cells. 3W culture systems shows an marked upregulation of inflammatory factors in stellate cells. But what is the effect of macrophages alone on pancreatic stellate cells in the absence of tumour cells? Or in other words, is the presence of tumour cells necessary to induce the highly inflammatory CAF phenotype in vitro?

The authors should also comment on the use of podoplanin (PDPN) in their studies: Is PDPN used a pan-CAF marker as the cell sorting strategy in Figure 4d would suggest and others have previously published (Elyada PMID: 31197017) or is PDPN based on these new studies rather an iCAF marker?

Figure 3: Tumour derived GM-CSF induces OSM secretion in macrophages. Does the tumour cell derived conditioned media also affect macrophage activation/polarisation?

Figure 4: this figure would benefit from additional data to strengthen the statement that OSM depletion induces a type of fibrosis that is tumour growth restrictive.

- i) Tissue section analysis to assess fibrosis/desmoplasia (i.e. Picrosirius Red)
- ii) Assess spatial localisation of selected reported immune cells, particularly CD8+ T cells: It has previously been suggested by other groups that high aSMA+ CAFs impair T cell infiltration leading to immune cold tumours. Based on the CyTOF data, the authors report that OSM depletion results in an increase in anti-tumorigenic immune cells. However, are these immune cells indeed able to infiltrate aSMARich tumours or are the immune cells rather accumulating in the periphery?
- iii) is there a reduction of pSTAT3 in CAFs in the OSM deficient mice which would confirm the in vitro data from Fig 2h?
- iv) Currently, the authors use OSM knockout animals which lack OSM expression in the entire body. Thus, while the RNASeq data analysis suggest that OSM is mainly expressed by macrophages, the reported in vivo mouse study is only in part supporting this statement since all host cells will lack OSM expression in this model. To further strengthen the point that OSM is mainly expressed by macrophages and to show its biological relevance in vivo, the authors could perform a bone marrow transplant experiment by generating chimeric mice lacking OSM expression in their bone marrow

followed by tumour implantation.

Figure 5: Reported data and conclusions are based on a large screening experiment. To strengthen their statements, the authors could further validate selected targets using kinase specific antibodies to confirm observed changes in phosphorylation in their screen.

Figure 6: The reported increase in epithelial like tumour cells in response to OSM depletion would benefit from additional data. As shown, in the OSM^{-/-} tumours, there are very few CK positive ducts/cells and the ductal structures could also represent in part remaining normal pancreatic ducts, surrounded by tumour cells which lost their CK expression. Could the author strain for another cancer cell marker, or anti-iRFP to rule this out?

Along these lines, some functional analysis of reduced migration/invasion of cancer cells in response to 3W conditioned media would further strengthen the authors claims. Or change in EMT marker expression?

Methods:

Incorrect methods: mouse macrophage cell line. Provided ID is a culture media, not a cell line.

Mouse in vivo: the authors should please refer to the generation of a knockout mouse line rather than the generation of a "genetically engineered mouse model (GEMM)".

Minor comments:

Figure 1A: Author should provide full list of genes identified for both CAF subtype in their analysis reported in figure 1A.

Reviewer #3 (Remarks to the Author): with expertise in PDAC and cancer associated fibroblasts

Lee et al investigate a tumour-promoting cross-talk between epithelial cells, fibroblasts and macrophages in pancreatic cancer. Among their findings, they show that in vitro cancer-secreted GM-CSF induces oncostatin M (OSM) production from macrophages, which in turn induces a more inflammatory phenotype in fibroblasts. By using a whole-body knock out of OSM, the authors also show OSM role in pancreatic tumour growth and immune representation. These are important new findings further highlighting the complexity and heterogeneity of the tumour microenvironment in pancreatic cancer. Due to the known roles of OSM in other malignancies and diseases, these findings could also be relevant in other contexts.

Overall, this is a good study with most conclusions supported by the data and by nice complementary experiments. The text and methods are well presented, though the figure legends would benefit from more details in some instances. A few points should be addressed experimentally and/or some conclusions should be re-phrased prior to publication in Nature Communications.

Main points

- 1) The authors should clarify for each figure panel how many different lines, and which ones, they have used for the experiments reported. This is not always clear and it is necessary to assess the robustness of their findings. For example, it is unclear whether some transplantation experiments have been only performed with one pancreatic cell line, or whether only one PCC/fibroblast/macrophage line has been used for some in vitro experiments. Using only one cell line would not be, in my opinion, sufficient for publication.
- 2) The authors highlight the observation of less metastases in the Osm KO mice as striking. However, this is not particularly surprising as it is in line with the fact that primary tumours are smaller in the Osm KO mice, and so it is not evidence of the major role of OSM in metastasis formation, but rather of the role of OSM depletion in impairing pancreatic cancer progression. Therefore, this should be rephrased (lines 431-434) or more data should be presented in support of the authors' conclusion. For

example, if the metastasis burden was found to be different in the 2 cohorts when mice reached endpoint, that would be a more supportive evidence than the one provided so far as it appears that both cohorts were culled at the same time.

3) Related to the point above, the authors should discuss other possibilities of their current cancer cell/macrophage/fibroblast model. In particular, OSM is also expressed by dendritic cells in PDA. Moreover, the authors mention that OSMR is only expressed in fibroblasts. This is not the case. Analysis of the same single-cell RNA-sequencing datasets that the authors look at (from Elyada et al) show that OSMR is also expressed in endothelial cells and pericytes in both murine and human PDA. The violin plots that the authors use to illustrate the data do not show low levels of gene expression. A different type of representation (e.g. normalised/scaled heatmap/dot blot) is particularly important since single-cell RNA-sequencing is already biased towards highly expressed genes due to its poor gene coverage. Can the authors confirm whether OSMR levels were indeed undetectable by qPCR in their pancreatic cancer cell lines? Can the authors confidently state that there is no possible direct OSM-OSMR signalling from macrophages to cancer cells? E.g. can the authors evaluate in published RNA-seq datasets (PDA cancer lines and/or organoids) whether indeed OSMR is not expressed by PDA cells?

4) In relation to the point above, the fact that endothelial cells express OSMR should be discussed in light of the authors' in vivo results and in particular in respect to the authors' claims that OSM plays a role in metastasis formation. Analysis of the vasculature (e.g. CD31 staining) should be performed to evaluate whether any difference is observed between the 2 cohorts, as it could have an effect on the epithelial differentiation status and potential metastatic differences that the authors discuss as mediated by changes in fibroblast composition.

5) Also related to point 3), in Figure 5, it appears that the 2-way co-culture of cancer cells and macrophages was not included. However, this is a key control to conclude that the macrophage-activated fibroblasts are mediating the changes in signalling observed in the cancer cells, and that this is not a direct effect of macrophages on cancer cells. Moreover, to conclude that these changes are indeed mediated by the OSM-OSMR signalling (lines 405-408), further evidence is needed (e.g. performing the same MS experiments with OSMR KO PSCs). If no additional experiment can be performed, the authors should re-phrase this. To support the authors' conclusions in respect to their in vivo findings, more evidence should be provided to show that these changes in signalling pathways have a functional read-out (e.g. is the proliferation of cancer cells affected? Is the migration of cancer cells affected?). Similarly, whereas in extended figure 2B, it appears that 2-way cancer cell/macrophage cultures have been analysed, the results are not reported in extended figure 2c.

6) Similar to the point above, whereas tumour growth is clearly impaired in OSM KO mice, it is less clear whether this is indeed dependent on a change in differentiation status of the cancer cells, as suggested by the authors. More evidence should be provided in support of a more differentiated histology of tumours in OSM KO mice (e.g. MUC5AC IHC ECAD IHC). Moreover, are Ki67 or pH3 levels different in the 2 cohorts? And CC3 stain? Additionally, the authors should clarify how they performed the epithelial/mesenchymal PanCK quantifications in figure 6.

7) To make conclusions in Figure 3 (panels a, b, c, d), all ELISA results should be complemented by qPCR validation (and ideally also proliferation assays), as cell proliferation differences may lead to misleading results (i.e. more cells could result in more ligand in the media regardless of whether the ligand is actively induced). Similarly, the authors should discuss/show whether the Luminex data in figure 2a may be dependent on differences in cell proliferation. Similarly, the authors mention "despite maintaining a constant total cell number" on lines 145-146. However, it is unclear how this was evaluated, and these data should be shown.

8) Clear validation of the OSMR KO in PSCs should be included (e.g. western blot or DNA sequencing analysis – qPCR analysis is not sufficient). Can the authors also please clarify whether these are pooled lines or derived from single clones?

Minor points

1) Some panel figures require more information or need to be fixed, for example:

a. figure 1a requires labels (quiescent PSCs, myCAFs, iCAFs).

b. I believe 1c (right panel, mouse PDA) is not correct. It appears *Osmr* is high in ductal cells and

RBCs but the main text states otherwise.

c. As mentioned above, I suggest to include a dot blot or normalised heatmap plot instead of/in addition to all violin plots, as it will show more clearly the differences in gene expression between clusters.

d. In the methods, the authors talk about 8513 cells for the generation of transplantation models, however in figure 3a/c they are not shown and it is unclear for which in vivo experiments they have been used.

2) I believe figure 6D is not cited in the text.

3) The authors perform flow cytometry/FACS on fibroblasts by gating on double positive PDPN and CD90 cells. However, CD90 (=Thy1) has been shown to be more highly expressed in myCAFs in single-cell RNA-sequencing datasets of PDA and therefore it seems confusing to look at inflammatory markers in this population. Although I do not think it is needed for publication to repeat the sorting + qPCR analysis on all PDPN+ cells, the authors should comment on this caveat of their experimental design and clarify their choice. Related to this, as a suggestion, the authors could look at the datasets they already have and calculate the % of CD45-EPCAM-PDPN+CD90+ (myCAFs-enriched) and CD45-EPCAM-PDPN+CD90- (iCAFs-enriched) in control WT mice and Osm KO mice as proxy for detecting changes in iCAF/myCAF populations. Do PDPN+CD90+ increase and PDPN+CD90- decrease as suggested by the other data reported?

4) In Figure 4b, it appears some OSM is detected. Can the authors discuss why this is the case? Is this OSM from the transplanted cancer cells since this is a whole-body KO mouse? How does this affect the model that the authors propose?

5) In Figure 4, do the changes in SMA levels correlate with changes in matrix deposition (e.g. Masson's trichrome stain) since myCAFs have been associated with that and this may have an effect on the epithelial differentiation status?

6) The authors should briefly discuss the limitations of the monolayer co-culture system used. For example, Figure 2 shows no induction of inflammatory gene signatures in PSCs upon co-culturing with cancer cells, whereas SMA is induced (Extended figure 3a), which suggests that cancer cells do not induce an inflammatory phenotype in PSCs, which is not correct as shown by other studies and is likely due to the fact that here the PSCs are cultured in monolayer. Whether macrophages would have such a profound effect in inducing the iCAF phenotype in the presence of cancer cells/organoids in less stiff/bi-dimensional conditions (which are rather different from the in vivo situation) should also be briefly discussed.

7) As IL1 has been shown to induce the iCAF phenotype, the authors should look at IL-1 (alpha and beta) expression in cancer cells, macrophages and fibroblasts in the mono-cultures and 2-way, 3-way co-cultures in figure 1b. The authors perform good control experiments with an anti-IL1 beta antibody, however IL1 alpha could also play a role in their system and this possibility should be discussed or experimentally addressed.

8) The authors mention that PSCs were only used for 5 passages in the method section. Can the authors please clarify whether this is also true for the Osmr KO PSCs, as it appears difficult to generate Cas9 cells, perform a KO, validate the KO and perform all experiments within 5 passages from the initial isolation of the cell line.

9) The authors should briefly discuss the possibility that the results presented in figure 4b may not be solely due to changes in the fibroblast compartment, but also in cancer cells and immune cells.

10) The authors should briefly discuss the limitations of the PDA cell line-transplantation model they use since, as it is typical for these models, the tumours (see figures 4 and 6) are highly cellular, which is not reflective of human PDA.

11) I find confusing that the levels of PDPN, which is a pan-fibroblast marker in vivo, increase in the triple co-culture (e.g. extended figures 2h and 3b). Can the authors please comment on this? Are the authors suggesting that PDPN is an iCAF marker in their in vitro system?

12) On line 104, the authors state that OSMR, ANTXR1 and IL1R1 expression is restricted to fibroblasts. As above, I would be cautious with these statements, as by sub-clustering on specific populations it is clear that this is not the case, and as single-cell RNA-sequencing does not show lower gene expression levels. I would change this sentence to "largely restricted".

13) Can the authors include OSMR expression levels in iCAF/myCAF in human PDA in addition to figure

1b?

14) Contrary to what stated on lines 273-274, Figure 3C and 3A do not entirely correlate. PCC12 does, but PCC7 (and PCC) less so. The authors should re-phrase these lines and perhaps comment on this discrepancy.

The following experiments will not be required for publication in Nature Communications, but they would strengthen the authors' conclusions if available:

- 1) Can OSM KO macrophage lines be generated (e.g. isolated from Osm KO mice) as a complementary approach to the OSMR KO PSC experiments in vitro?
- 2) Can GM-CSF KO cancer cells be generated as a complementary approach to the neutralizing antibody experiments in vitro?
- 3) Can the authors sort out cancer cells from OSM KO mice and WT mice and evaluate EMT/proliferation signatures/genes?

Reviewer: Giulia Biffi

Reviewer #1 (Remarks to the Author): with expertise in cancer and transcriptomic

The manuscript reveals a novel signaling in pancreatic cancer development that dictates the pro-tumor growth/metastasis and shapes immunosuppression environment in Pancreatic cancer. Overall, the finding is new in PDA. The role of OSM-OSMR signaling in tumor growth and metastasis is mostly convincing except some conflicts of the published results and some confusing result on CAF states that need to be addressed.

We thank the reviewer for the constructive comments and suggestions.

1. First, it is advised the authors should carefully examine and demonstrate the OSM and OSMR signal in the published single-cell RNAseq datasets for real PDAC. (Peng et.al. Cell Research 2019 and Lin et.al. Genome Medicine, 2020). How do these datasets support the authors' conclusions? Does OSMR specifically express in CAF only or any other cell populations? These datasets reflect the real situation of PDA than the ex vivo and animal models.

We thank the reviewer for raising this important point. In the first submission, we analysed publicly available scRNA-seq data from human PDA (Elyada et al. 2019) and from two different murine models of PDA (Elyada et al. 2019 and Dominguez et al. 2020). As kindly suggested by the reviewer, we downloaded and analysed scRNA-seq data from both Peng et al. and Lin et al. (human PDA). However, in contrast to the data from Elyada et al., both of these publications had only used 2 or 3 genes to annotate cell identity. This is much less stringent than for Elyada et al. where a much more extensive gene list was used. Moreover, the data from both Peng et al. and Lin et al. have high dropout. Seeking to re-annotate the scRNA data from Lin et al. and Peng et al. using the more rigorous annotation from Elyada et al. results in drastic loss of data points. For example, at thresholds where 99.9% and 75% of single cells from Elyada et al. pass QC, 80% and 17% of cells from Lin et al. and 61% and 5% from Peng et al. pass QC. Limiting the number of genes used for cell annotation results in poorly defined cell populations and thus annotating cell-specific expression becomes much less stringent.

Due to the higher data quality, we have only included analysis of human PDA scRNA-seq from Elyada et al. in this manuscript (Fig. 1 and Extended Fig. 1). To further confirm *OSMR* expression in human PDA, we have also now included *in situ* hybridisation for *OSMR* (we have not been able to validate any commercially available antibodies). This analysis supports *OSMR* expression in the stroma (Fig. 1d and Extended Fig. 1f). Importantly, we also identify epithelial cells that are positive for *OSMR*, suggesting that at least at mRNA level *OSMR* is also expressed in epithelial cells.

The RT-qPCR data included in Fig. 3a demonstrates cell-specific expression of *Osmr* and *Osm* *in vitro*. To strengthen the observation that paracrine *Osm* signalling reprogram fibroblasts, we have: i) further validated that the PCCs used in this study do not respond to OSM (Extended Fig. 7e) and ii) that regulation of tumour cell signalling in 3-way co-culture is dependent on fibroblast *Osmr* (Fig. 6d). This confirms the presented model (Fig. 7e), at least under the experimental conditions used in this study. We fully acknowledge that mouse *in vitro* and *in vivo* models do not fully replicate the human condition and have also now included a discussion of the study limitations.

2. Fig.1A is unclear. The rows for the specific 3 gene expressions should be marked. The column labels should be marked.

Thank you and we apologies for this omission in the first submission. We have updated the figure and annotated selected genes, including the three receptors highlighted in the analysis (*Osmr*, *Antxr1* and *Il1r1*). All genes included in this analysis has been included in Fig. 1 source data for readers to further interrogate.

3. Fig.1B doesn't show *Osmr* is predominantly expressed to iCAF.

We apologies for the lack of clarity and have edited the text in the manuscript to more accurately reflect the data. It now reads: "Moreover, these receptors [*Osmr*, *Antxr1* and *Il1r1*] were expressed at higher levels in iCAFs compared to myCAFs and apCAFs".

4. As a possible control, why the assays for the 2W co-culture of PSC-MØ were not mentioned at all? I will be still interested to learn what could happen if tumor cell is not available. If GM-CSF from tumor cells drives the secretion of OSM, can we test PSC for inflammatory response in the 2W co-culture of PSC-MØ without PCC but using recombinant GM-CSF medium?

This is a reasonable point that the reviewer raises. As the main objective of the study is to characterise the influence of a tumour co-opted stroma, we had initially omitted this experimental condition. However, we agree with the reviewer that this is an important and interesting experiment and have therefore now included following analyses:

Firstly, we included an analysis of *Osm* and *OsmR* expression in PCCs, PSCs and primary bone marrow-derived macrophages (Fig. 3a). Consistent with the data presented in the initially submitted manuscript we observe *Osm* expression in BMDMs (Fig. 3a). Further, *Osmr* is expressed in PSCs and is increased by conditioned medium from PCC+PSC co-cultures. Three-way co-cultures with PCC+PSC+BMDM as well as GM-CSF+PSC+BMDM further increase PSC *OsmR* expression (Fig. 4d). This data confirms the cell-specific expression of *Osm* and *Osmr* using primary BMDMs and that GM-CSF replicates the effect of PCCs on *Osmr* expression in PSCs.

In addition, we examined the expressed of selected inflammatory genes (*Il6*, *Il4ra* and *Cxcl1*) and the myofibroblastic marker gene *Acta2* in PSCs stimulated with conditioned medium from mono-cultures of bone marrow-derived macrophages (MØ), PSCs and PCCs as well as co-cultures of MØ+PSCs, PSCs+PCCs, MØ+PSCs+PCCs and MØ+PSCs+GM-CSF (Fig. 2f and 4d). Conditioned medium from co-cultured PSCs+MØs does not induce expression of *Il6*, *Il4ra*, *Cxcl1* in PSCs (Fig. 2f). *Acta2* expression also doesn't change, suggesting that interactions between MØs and PSCs does not induce a similar change as observed between PCCs, PSCs and MØs (Fig. 2f). Consistent with the data included in the original submission, MØ+PSCs+PCCs increase *Il6*, *Il4ra* and *Cxcl1* expression and decrease *Acta2* expression in PSCs (Fig. 2f and 4d). Importantly, the effect of PCCs can be substituted by GM-CSFs e.g. MØs+PSCs+GM-CSF conditioned medium result in similar increase in *Il6*, *Il4ra*, *Cxcl1* and *Acta2* expression as MØs+PSCs+PCCs (Fig. 4d).

The new data shows that PSC+BMDM does not change expression of inflammatory or myofibroblastic marker genes in PSCs, suggesting that PCCs play a critical role in co-opting stromal cells to alter PSC marker expression. GM-CSF is able to induce *Osm* in MØs and to substitute for PCCs in co-culture mediated fibroblast reprogramming. This suggests that indeed GM-CSF plays an important functional role in driving deregulated tumour-macrophage-fibroblast interactions.

5. If the authors try to make the same points using Fig. 2B (PSC columns) and Fig. 2D, shouldn't the points of PSC and PSC+PCC be clustered together in the mixed colors, when PCC alone doesn't change PSC? If it does, what has changed?

We thank the reviewer for this question. The segregation of the PSCs under the three *in vitro* conditions is driven by the differences in the experimental conditions e.g. increased expression of inflammatory markers (Fig. 2b) and differential expression of protein markers (Fig. 2g) in 3W co-culture conditions.

The data shown in Fig. 2b is unsupervised hierarchical clustering of cytokine and growth factor expression analysed by RT-qPCR. The separation of PSC and PSC+PCC co-culture is mainly driven by differentially secreted cytokines e.g. *Ccl11*, *Il6*, *Cx3cl1*, *Hgf* and *Fgf2*.

The data shown in Figure 2g is a tSNE plot using of a CyTOF analysis with selected antibody-based markers of mesenchymal cells, with the experimental conditions overlaid.

The specific markers analysed are different between the CyTOF analysis and the RT-qPCR analysis, but both experimental approaches demonstrate a shift in the marker expression consistent with a change in PSC 'state' between mono-cultured PSCs, 2W and 3W co-culture. We suspect the minor difference between PSC mono-culture and PSC+PCC (2W) co-culture is due to stiffness-induced myofibroblastic gene expression in PSCs due to their culture on regular tissue culture dishes (consistent with the previously described myofibroblasts e.g. Ohlund et al. 2017, Biffi et al. 2019 and Elyada et al. 2019).

6. In Fig. 2G and 2H, using one of the PC axes to reflect the inflammatory level of CAF is unusual. The variance other than inflammatory genes could contribute to PC1. Z-score is a statistic defined within the assayed samples. It is also not a great measurement of inflammatory state because there is no clear threshold. The authors should have a more straightforward way to illustrate the inflammatory state of CAFs.

We apologise for the confusion. The PC axes in Figure 3b and 3c are used to show the variation in gene expression from the RT-qPCR analysis. Selected genes from this analysis are displayed in the accompanying bar graphs and in Extended Fig. 3. The iCAF score is a sum of mean z-values of 15 selected inflammatory genes and is overlaid onto the PCA to illustrate the relative expression of inflammatory genes. We have clarified this in the figure legend of the resubmitted version of the manuscript.

7. The color legend of Fig. 4B for WT vs. Osm^{-/-} is different from other Fig. 4 panels. I believe it was mislabeled.

Thank you for picking this up, we have corrected this in the resubmitted version of the manuscript.

8. Other than the comparison of metastasis in wild-type vs. Osm^{-/-} mice, the data to correlate the OSM-OSMR signaling and metastasis through PSC-MØ signaling is inadequate. It would be better if the authors at least check the correlation between the OSM-OSMR signal and the EMT signal (the expression of EMT gene set, including SNAI2, CDH2, ZEB1, TWIST1, VIM etc.) in the published datasets. If the co-upregulation is found in the

same patient, it would be more convincing. Or let's find out if iCAF signal is high in metastatic tumors. Maybe the recent scRNAseq datasets could help.

We thank the reviewer for these comments and suggestions and have now included additional data to support the observations and conclusions made in the manuscript.

As kindly suggested by the reviewer we have compared the level of selected EMT gene expression in PDA patients with high and low *OSMR* expression (Extended Fig 7i). Consistent with the data included in the original submission, this data demonstrates a relationship between *OSMR* expression and individual EMT gene expression in human PDA.

Further, we have used immunoblotting to examine the level of epithelial and mesenchymal markers in tumour cells stimulated with medium from mono- and co-cultures (Fig. 6f and Extended Fig. 7j). Consistent with the data included in the initially submitted manuscript, E-Cadherin is decreased and Vimentin, Slug and Snail are increased in PCCs following stimulation with conditioned medium from PCC+PSC+MØ (Fig. 6f and Extended Fig. 7j). This demonstrates that epithelial marker expression is decreased and mesenchymal markers are increased when tumour cells are exposed to signals from PCC, PSC and MØ co-cultures. This is also confirmed by IHC for SLUG, which is higher in orthotopic tumours from Wt animals compared to *Osm*^{-/-} animals (Fig. 7c and Extended Fig. 8c).

The additional data supports the notion that interaction between tumour, macrophages and fibroblasts engage tumour cell signalling networks leading to a more mesenchymal phenotype.

Overall, I am convinced for the significant role of the OSM-OSMR signaling in pancreatic cancer. However, I am not very sure it works through the regulation of iCAF-myCAF transformation as OSMR is not specific in iCAF subpopulations or even in CAF. Actually, I am neither quite sure if the inflammatory state of CAF can be convincingly defined by the authors in this paper. They used co-culture and FACS sorting and the follow-up qPCR assay, which takes too long after the CAF leaves the original biological context. It may not reflect the real CAF state in the PDA tissue after all. We need a method to measure the inflammatory state in the tissue on-the-fly.

We thank the reviewer for these considerations. The fibroblasts phenotypes (iCAF and myCAF) are based on marker gene expression consistent with previous publications (Ohlund et al. 2017, Biffi et al. 2019 and Elyada et al. 2019). We acknowledge that the *in vitro* model system is reductionist and that further experiments are needed to further elucidate additional signalling regulated by *Osm* in the tumour microenvironment (included in the discussion). Hopefully the observations in the manuscript, together with the growing literature supporting diverse roles of CAFs in the TME, will enable future methods development to determine cell and tissue states on the fly.

As I have checked in previous scRNAseq dataset for PDAC tissues, other than CAF, both EMT tumor cells and Endothelial cells show high expression of OSMR, whose role shouldn't be overlooked in the OSM-OSMR signaling.

We thank the reviewer for this comment and agree that the complexity of the TME *in vivo* can only be partly replicated by reductionist *in vitro* models and have therefore included this in the discussion and highlighted the need for additional experiments. However, while reductionist, *in vitro* models remain valuable in determining the impact of specific cellular interactions on

cellular signalling, which then can be further interrogated *in vivo* and in human clinical samples.

Reviewer #2 (Remarks to the Author): with expertise in PDAC and macrophages

In this paper, Lee et al studies the formation of different subtypes of cancer associated fibroblast (CAF) in pancreatic cancer. They claim that macrophage derived factors, particularly Oncostatin M (OSM) induces the formation of an highly inflammatory CAF phenotype, which promotes tumour formation and metastatic progression. In a pre-clinical mouse model, genetic ablation of OSM expression drives CAF subtypes towards a more myofibroblastic phenotype, reduces tumour burden and metastasis, and promotes the presence of rather immune-stimulatory immune infiltrates.

The paper contains a substantial amount of work. The findings that macrophage-derived OSM promotes iCAF polarisation are novel and interesting and complements previous work from the Tuveson group. The reported experimental data are of high quality and the manuscript is well written.

We thank the reviewer for the constructive comments and suggestions.

Following additional analysis would improve the quality of this manuscript:

Main points:

Figure 1: The identification of OSM expression by macrophages and the expression of OSMR by fibroblasts are based on RNA Seq data analysis, but and their expression hasn't been validated in tissue. Thus, the validation of these findings in human/mouse tissue section of PDAC tumours would strengthen the reported cell type-specific expression of these two factors.

We thank the reviewer for raising this important point. To confirm *OSMR* expression in human PDA, we have now included *in situ* hybridisation for *OSMR* (we have not been able to validate any available antibodies). This analysis support *OSMR* expression in the stroma (Fig. 1d and Extended Fig. 1f). Importantly, we also identify epithelial cells that are positive for *OSMR*, suggesting that at least at mRNA level *OSMR* is also expressed in epithelial cells.

The qPCR included in Fig. 2a and Fig. 3a demonstrates cell-specific expression of *Osmr* and *Osm* *in vitro*. To strengthen the observation that paracrine *Osm* signalling reprogram fibroblasts, we have i) further validated that the PCCs used in this study do not respond to OSM (Extended Fig. 7e) and ii) that regulation of tumour cell signalling in 3-way co-culture is dependent on fibroblast *Osmr* (Fig. 6d). This support that, at least under the experimental conditions used in this study, the presented model (Fig. 7e) holds up. We fully acknowledge that mouse *in vitro* and *in vivo* models cannot fully replicate the human condition and have also now included a discussion of the study limitations.

Figure 2: Comprehensive analysis of 2- and 3-way co-culture systems using macrophages, pancreatic stellate cells and cancer cells. 3W culture systems shows a marked upregulation of inflammatory factors in stellate cells. But what is the effect of macrophages alone on pancreatic stellate cells in the absence of tumour cells? Or in other words, is the presence of tumour cells necessary to induce the highly inflammatory CAF phenotype *in vitro*?

The authors should also comment on the use of podoplanin (PDPN) in their studies: Is

PDPN used a pan-CAF marker as the cell sorting strategy in Figure 4d would suggest and others have previously published (Elyada PMID: 31197017) or is PDPN based on these new studies rather an iCAF marker?

1) This is a reasonable point that the reviewer raises. As the main objective of the study is to characterise the influence of a tumour co-opted stroma, we had initially omitted this experimental condition. However, we agree with the reviewer this is an important and interesting experiment and have therefore now included following analyses:

Firstly, we included an analysis of *Osm* and *OsmR* expression in PCCs, PSCs and primary bone marrow-derived macrophages (Fig. 3a). Consistent with the data presented in the initially submitted manuscript we observe *Osm* expression in BMDMs (Fig. 3a). Further, *OsmR* is expressed in PSCs and is increased by conditioned medium from co-cultures of PCCs+PSCs. Three-way co-cultures with PCC+PSC+BMDM as well as GM-CSF+PSC+BMDM further increase PSC *OsmR* expression (Fig. 4d). This data confirms the cell-specific expression of *Osm* and *Osmr* using primary BMDMs and that GM-CSF replicates the effect of PCCs on *Osmr* expression in PSCs.

In addition, we examined the expressed of selected inflammatory genes (*Il6*, *Il4ra* and *Cxcl1*) and the myofibroblastic marker gene *Acta2* in PSCs stimulated with conditioned medium from mono-cultures of bone marrow-derived macrophages (MØ), PSCs and PCCs as well as co-cultures of MØ+PSCs, PSCs+PCCs, MØ+PSCs+PCCs and MØ+PSCs+GM-CSF (Fig. 2f and 4d). Conditioned medium from co-cultured PSCs+MØs does not induce expression of *Il6*, *Il4ra*, *Cxcl1* in PSCs (Fig. 2f). *Acta2* expression also doesn't change, suggesting that interactions between MØs and PSCs does not induce a similar change as observed between PCCs, PSCs and MØs (Fig. 2f). Consistent with the data included in the original submission, MØ+PSCs+PCCs increase *Il6*, *Il4ra* and *Cxcl1* expression and decrease *Acta2* expression in PSCs (Fig. 2f and 4d). Importantly, the effect of PCCs can be substituted by GM-CSFs e.g. MØs+PSCs+GM-CSF conditioned medium result in similar increase in *Il6*, *Il4ra*, *Cxcl1* and *Acta2* expression as MØs+PSCs+PCCs (Fig. 4d).

The new data shows that PSC+BMDM does not change expression of inflammatory or myofibroblastic marker genes in PSCs suggesting that PCCs play a critical role in co-opting stromal cells to alter PSC marker expression. GM-CSF is able to both induce *Osm* in MØs and to substitute for PCCs in co-culture mediated fibroblasts reprogramming suggesting that indeed GM-CSF plays an important functional role in driving deregulated tumour-macrophage-fibroblast interactions.

2) PDPN is broadly expressed in fibroblasts of the pancreas, and was included as a general marker of fibroblasts. Both myofibroblastic and inflammatory fibroblasts express PDPN, however inflammatory fibroblasts display increased levels of PDPN. In addition to the *in vitro* co-culture data included in this manuscript, mass cytometry analysis of murine KPC tumours (Hutton et al. Cancer Cell 2021) also show that inflammatory CAFs display more abundant PDPN than myofibroblastic CAFs (see Figure insert), however there is clearly a high degree of inter-tumour variability in PDPN levels.

Figure 3: Tumour derived GM-CSF induces OSM secretion in macrophages. Does the tumour cell derived conditioned media also affect macrophage activation/polarisation?

To determine whether tumour cell-derived medium and GM-CSF also change macrophage polarity, we have now included new data where we have analysed the expression of M1 (*Il6*, *Tnfa*) and M2 (*Arg1*, *Ym1*) polarity markers after BMDMs were treated with conditioned medium from tumour cells or GM-CSF. Whereas GM-CSF induced expression of *Osm* comparably to conditioned medium from selected PCCs, expression of *Il6*, *Tnfa*, *Arg1* or *Ym1* are not regulated by GM-CSF. As expected tumour cell lines have different effect on the expression of M1 and M2 markers, with no clear pattern. Since the study focus on interactions between tumour and stromal cells and the effect of macrophage-PSC interactions in driving a distinct tumour cell phenotype, we have included the data in the rebuttal letter (see below) but not in the resubmitted manuscript as we feel this will detract from the main observation without adding much additional insight.

Figure 4: this figure would benefit from additional data to strengthen the statement that OSM depletion induces a type of fibrosis that is tumour growth restrictive.

- i) Tissue section analysis to assess fibrosis/desmoplasia (i.e. Picrosirius Red)
- ii) Assess spatial localisation of selected reported immune cells, particularly CD8⁺ T cells: It has previously been suggested by other groups that high aSMA⁺ CAFs impair T cell infiltration leading to immune cold tumours. Based on the CyTOF data, the authors report that OSM depletion results in an increase in anti-tumorigenic immune cells. However, are these immune cells indeed able to infiltrate aSMA⁺ rich tumours or are the immune cells rather accumulating in the periphery?
- iii) is there a reduction of pSTAT3 in CAFs in the OSM deficient mice which would confirm the in vitro data from Fig 2h?
- iv) Currently, the authors use OSM knockout animals which lack OSM expression in the entire body. Thus, while the RNASeq data analysis suggest that OSM is mainly expressed by macrophages, the reported in vivo mouse study is only in part supporting this statement since all host cells will lack OSM expression in this model. To further strengthen the point that OSM is mainly expressed by macrophages and to show its biological relevance in vivo, the authors could perform a bone marrow transplant experiment by generating chimeric mice

lacking OSM expression in their bone marrow followed by tumour implantation.

The reviewer raises a number of very good points, which we have sought to address experimentally and by clarifying the text where possible. Importantly however, while the tumour cells are labelled by a near-infrared fluorescent proteins which is highly quantitative, no antibodies currently exist allowing us to identify iRFP expressing cells at high confidence in the samples already collected and processed. Unfortunately, due to COVID restrictions we have been unable to undertake additional *in vivo* experiments, such as the experiment suggested in point iv), but aim to further expand on the *in vivo* analysis in future studies.

Specifically:

i) To address whether fibrillar collagen is changed in tumours of Osm WT and OSM KO animals we have now included an analysis using Picrosirius Red (Extended Fig. 5a). Curiously, we do not observe any differences in the staining intensity, suggesting that fibrillar collagen levels are not different under these two conditions. Importantly, this does not demonstrate that the ECM isn't composed differentially between tumours in Wt and *Osm*^{-/-} animals. To address this comprehensively we are aiming to undertake a proteomics experiment (See papers by Naba and colleagues). Unfortunately, due to COVID we have been restricted in our ability to undertake animal experiments and thus these analyses will be included in future studies.

ii) This is a great suggestion. We have included IHC for CD8 and while we do not observe differences in the total number of CD8^{pos} cells, which is consistent with the CyTOF analysis, we do observe difference in the ratio of central vs peripheral CD8^{pos} cells. This data has been included in the resubmitted manuscript (Fig. 5e and Extended Fig. 5c).

iii) To address this point, we co-stained for pSTAT3, Vimentin and CD45 in tumours from Wt and *Osm*^{-/-} animals. To identify stromal, non-immune cells we isolated VIM^{pos}/CD45^{neg} cells and compared pSTAT3 levels. We observe no discernible difference in pSTAT3 levels (see Figure below). However, temporal regulation of signalling and redundancies of signalling pathways will influence this result and therefore complicates conclusions. Consequently, we have included the data here, but have not included it in the resubmitted manuscript.

iv) We appreciate the reviewer's suggestion and agree such an experiment would add great value, however due to COVID restrictions on animal breeding and the development of new experiments models, we have been unable to undertake such an experiment. We thank the reviewer for this very good suggestion and aim to expand our studies to address this in the future.

Figure 5: Reported data and conclusions are based on a large screening experiment. To strengthen their statements, the authors could further validate selected targets using kinase specific antibodies to confirm observed changes in phosphorylation in their screen.

We thank the reviewer for this point. As part of the initial submission we validated several of the key signalling pathways that were identified and regulated in the phosphoproteomics analysis using an antibody phospho-array (Extended Fig. 5d). However, we acknowledge that additional control experiments are warranted and have therefore included the following control experiments:

- i) To ensure OSM signalling specificity, we have included immunoblotting of pSTAT3 in PCCs and PSCs treated with recombinant OSM. This demonstrates that under these conditions, PSCs, but not PCCs, respond to OSM by increased pSTAT3 signalling (Extended Fig. 7e).
- ii) Furthermore, we compared the effect of conditioned medium from co-cultured tumour cells (PCCs), macrophages (MØ) and fibroblasts (PSCs, wild-type or *Osmr* KO) on tumour cell signalling. Immunoblotting of phospho-STAT3 (pSTAT3) levels confirms that pSTAT3 levels are specifically increased in 3W co-cultures with Wt PSCs but not with *Osmr* KO PSCs. This confirms that co-culture mediated rewiring of tumour cell signalling is dependent on *Osmr* in PSCs (Fig. 6d and Extended Fig. 7f).
- iii) Lastly, we compared the effect of conditioned medium from mono, 2W- and 3W- co-cultures on tumour cell signalling (pSTAT3) Fig. 6c. This demonstrates that only conditioned medium from 3W co-cultures induce tumour cell pSTAT3 signalling.

Figure 6: The reported increase in epithelial like tumour cells in response to OSM depletion would benefit from additional data. As shown, in the *OSM*^{-/-} tumours, there are very few CK positive ducts/cells and the ductal structures could also represent in part remaining normal pancreatic ducts, surrounded by tumour cells which lost their CK expression. Could the author strain for another cancer cell marker, or anti-iRFP to rule this out?

Along these lines, some functional analysis of reduced migration/invasion of cancer cells in response to 3W conditioned media would further strengthen the authors claims. Or change in EMT marker expression?

We thank the reviewer for these comments and suggestions and have now included additional data to support the observations and conclusions made in the manuscript.

We have compared the level of selected EMT gene expression in PDA patients with high and low *OSMR* expression (Extended Fig. 7i). Consistent with the data included in the original submission, this data demonstrates a relationship between *OSMR* expression and individual EMT gene expression in human PDA.

Further, we used immunoblotting to examine the level of epithelial and mesenchymal markers in tumour cells stimulated with medium from mono- and co-cultures (Fig. 6f and Extended Fig 7j). Consistent with the data included in the initially submitted manuscript, E-Cadherin is decreased and Vimentin, Slug and Snail are increased in PCCs only following stimulation with conditioned medium from PCC+PSC+MØ (Fig. 6f and Extended Fig 7j). This demonstrates that epithelial marker expression is decreased and mesenchymal markers are increased when tumour cells are exposed to signals from PCC, PSC and MØ co-cultures. This is also confirmed by IHC for SLUG levels, which are higher in orthotopic tumours from Wt animals compared to *Osm*^{-/-} animals (Fig. 7c and Extended Fig. 8c).

The additional data supports the notion that interaction between tumour, macrophages and fibroblasts engage tumour cell signalling networks leading to a more mesenchymal phenotype.

Methods:

Incorrect methods: mouse macrophage cell line. Provided ID is a culture media, not a cell line.

Mouse in vivo: the authors should please refer to the generation of a knockout mouse line rather than the generation of a “genetically engineered mouse model (GEMM)”.

We thank the reviewer for picking this up and have updated the materials and methods

Minor comments:

Figure 1A: Author should provide full list of genes identified for both CAF subtype in their analysis reported in figure 1A.

Thank you and we apologies for this omission in the first submission. We have updated the figure and annotated selected genes, including the three receptors highlighted in the analysis (*Osmr*, *Antxr1* and *Il1r1*). All genes included in this analysis has been included in Fig. 1 source data for readers to further interrogate.

Reviewer #3 (Remarks to the Author): with expertise in PDAC and cancer associated fibroblasts

Lee et al investigate a tumour-promoting cross-talk between epithelial cells, fibroblasts and macrophages in pancreatic cancer. Among their findings, they show that in vitro cancer-secreted GM-CSF induces oncostatin M (OSM) production from macrophages, which in turn induces a more inflammatory phenotype in fibroblasts. By using a whole-body knock out of OSM, the authors also show OSM role in pancreatic tumour growth and immune representation. These are important new findings further highlighting the complexity and heterogeneity of the tumour microenvironment in pancreatic cancer. Due to the known roles of OSM in other malignancies and diseases, these findings could also be relevant in other contexts.

Overall, this is a good study with most conclusions supported by the data and by nice complementary experiments. The text and methods are well presented, though the figure legends would benefit from more details in some instances. A few points should be addressed experimentally and/or some conclusions should be re-phrased prior to publication in Nature Communications.

Thank you, Giulia, for taking the time to review the manuscript, your comments and suggestions are much appreciated.

Main points

1) The authors should clarify for each figure panel how many different lines, and which ones, they have used for the experiments reported. This is not always clear and it is necessary to assess the robustness of their findings. For example, it is unclear whether some transplantation experiments have been only performed with one pancreatic cell line, or

whether only one PCC/fibroblast/macrophage line has been used for some in vitro experiments. Using only one cell line would not be, in my opinion, sufficient for publication.

We thank the reviewer for this point and apologise for not making the information of which cell line(s) were used in each experiment clear. We have now included a table listing the name of all cell lines used in the manuscript (Supplemental Table 1), which should help clarify which cell line(s) have been used for individual experiments.

2) The authors highlight the observation of less metastases in the *Osm* KO mice as striking. However, this is not particularly surprising as it is in line with the fact that primary tumours are smaller in the *Osm* KO mice, and so it is not evidence of the major role of OSM in metastasis formation, but rather of the role of OCM depletion in impairing pancreatic cancer progression. Therefore, this should be rephrased (lines 431-434) or more data should be presented in support of the authors' conclusion. For example, if the metastasis burden was found to be different in the 2 cohorts when mice reached endpoint, that would be a more supportive evidence than the one provided so far as it appears that both cohorts were culled at the same time.

We appreciate the concerns raised by the reviewer and acknowledge that the experiments included in the manuscript cannot with certainty conclude that the absence of metastatic disease is unrelated to the decrease in primary tumour volume. We have therefore changed the text to reflect this and have included this point in the discussion of the manuscript.

3) Related to the point above, the authors should discuss other possibilities of their current cancer cell/macrophage/fibroblast model. In particular, OSM is also expressed by dendritic cells in PDA. Moreover, the authors mention that OSMR is only expressed in fibroblasts. This is not the case. Analysis of the same single-cell RNA-sequencing datasets that the authors look at (from Elyada et al) show that OSMR is also expressed in endothelial cells and pericytes in both murine and human PDA. The violin plots that the authors use to illustrate the data do not show low levels of gene expression. A different type of representation (e.g. normalised/scaled heatmap/dot blot) is particularly important since single-cell RNA-sequencing is already biased towards highly expressed genes due to its poor gene coverage. Can the authors confirm whether OSMR levels were indeed undetectable by qPCR in their pancreatic cancer cell lines? Can the authors confidently state that there is no possible direct OSM-OSMR signalling from macrophages to cancer cells? E.g. can the authors evaluate in published RNA-seq datasets (PDA cancer lines and/or organoids) whether indeed OSMR is not expressed by PDA cells?

We thank the reviewer for the above points and acknowledge that greater clarity is needed in terms of how the data are presented and described in the manuscript.

To improve the interpretation of the scRNA-seq we have now included heatmaps for all violin plots to reflect relative expression levels. In addition, we have now clarified the text emphasising the expression of *Osm* and *Osmr* in other cell populations in the TME. Further analysing *OSMR* expression in human PDA, we have also now included *in situ* hybridisation for *OSMR* (we have not been able to validate any commercially available antibodies). This analysis supports *OSMR* expression in the stroma (Fig. 1d and Extended Fig. 1f). Importantly, this analysis also identifies epithelial cells that are positive for *OSMR*, suggesting that at least at mRNA level *OSMR* is also expressed in epithelial cells.

The RT-qPCR data included in Fig. 3a demonstrates cell-specific expression of *Osmr* and *Osm* *in vitro*. To strengthen the observation that paracrine *Osm* signalling reprogram fibroblasts, we have: i) further validated that the PCCs used in this study do not respond to OSM (Extended Fig. 7e) and ii) that regulation of tumour cell signalling in 3-way co-culture is dependent on fibroblast *Osmr* (Fig. 6d).

The data included supports that interactions between tumour cells, fibroblasts and BMDM indeed are required for fibroblast reprogramming and reciprocal signalling in tumour cells. However, we acknowledge that tumours are more complex than can be accurately modelled *in vitro* and have included a discussion of the limitations of the study and possible implications of other *Osm/Osmr* expressing cell types *in vivo*.

4) In relation to the point above, the fact that endothelial cells express OSMR should be discussed in light of the authors' *in vivo* results and in particular in respect to the authors' claims that OSM plays a role in metastasis formation. Analysis of the vasculature (e.g. CD31 staining) should be performed to evaluate whether any difference is observed between the 2 cohorts, as it could have an effect on the epithelial differentiation status and potential metastatic differences that the authors discuss as mediated by changes in fibroblast composition.

This is a very good point. We have included immunohistochemical analysis for CD31 in the revised version of the manuscript. Here we compared tumours isolated from *Wt* or *OSM^{-/-}* animals. Notably, we don't observe any difference in the CD31 staining. However, this doesn't exclude the possibility that *Osmr* expressing endothelial cells and pericytes play a role. Thus, we have included the data but have not used it to discount a functional role of *Osmr* in these cells *in vivo*.

5) Also related to point 3), in Figure 5, it appears that the 2-way co-culture of cancer cells and macrophages was not included. However, this is a key control to conclude that the macrophage-activated fibroblasts are mediating the changes in signalling observed in the cancer cells, and that this is not a direct effect of macrophages on cancer cells. Moreover, to conclude that these changes are indeed mediated by the OSM-OSMR signalling (lines 405-408), further evidence is needed (e.g. performing the same MS experiments with OSMR KO PSCs). If no additional experiment can be performed, the authors should re-phrase this. To support the authors' conclusions in respect to their *in vivo* findings, more evidence should be provided to show that these changes in signalling pathways have a functional read-out (e.g. is the proliferation of cancer cells affected? Is the migration of cancer cells affected?). Similarly, whereas in extended figure 2B, it appears that 2-way cancer cell/macrophage cultures have been analysed, the results are not reported in extended figure 2c.

We appreciate the comments from the reviewer.

i) For the phospho-proteomics analysis in Fig. 6, relative quantification with stable isotope labelling by amino acid in culture (SILAC) was chosen due to a combination of quantitative accuracy and compatibility with off-line fractionation for improved coverage. However, a limitation of SILAC is that only 3 conditions can be compared. Thus, the included conditioned were selected to best cover the specific questions.

We acknowledge that additional experimental evidence would strengthen the conclusions of this analysis and have therefore now included following experiments:

- i) To ensure OSM signalling specificity, we have included immunoblotting of pSTAT3 in PCCs and PSCs treated with recombinant OSM. This demonstrates that under these conditions' PSCs, but not PCCs, respond to OSM by increased pSTAT3 signalling (Extended Fig. 7e).
- ii) Furthermore, we compared the effect of conditioned medium from co-cultured tumour cells (PCCs), macrophages (MØ) and fibroblasts (PSCs, wild-type or *Osmr* KO) on tumour cell signalling. Immunoblotting of phospho-STAT3 (pSTAT3) levels confirms that pSTAT3 levels are specifically increased in 3W co-cultures with Wt PSCs but not with *Osmr* KO PSCs. This confirms that co-culture mediated rewiring of tumour cell signalling is dependent on *Osmr* in PSCs (Fig. 6d and Extended Fig. 7f).
- iii) Lastly, we compared the effect of conditioned medium from mono, 2W- and 3W- co-cultures on tumour cell signalling (pSTAT3) Fig. 6c. This demonstrates that only conditioned medium from 3W co-cultures induce tumour cell pSTAT3 signalling.

ii) We have also included immunoblotting to examine the level of epithelial and mesenchymal markers in tumour cells stimulated with medium from mono- and co-cultures (Fig. 6f and Extended Fig 7j). Consistent with the data included in the initially submitted manuscript, E-Cadherin is decreased and Vimentin, Slug and Snail are increased in PCCs only following stimulation with conditioned medium from PCC+PSC+MØ (Fig. 6f and Extended Fig 7j). This demonstrates that epithelial marker expression is decreased and mesenchymal markers are increased when tumour cells are exposed to signals from PCC, PSC and MØ co-cultures. This is also confirmed by IHC for SLUG levels, which are higher in orthotopic tumours from Wt animals compared to *Osm^{-/-}* animals (Fig. 7c and Extended Fig. 8c). Moreover, we have compared the level of selected EMT gene expression in PDA patients with high and low *OSMR* expression (Extended Fig. 7i). Consistent with the data included in the original submission, this data demonstrates a relationship between *OSMR* expression and individual EMT gene expression in human PDA.

iii) Lastly, we included an analysis of *Osm* and *OsmR* expression in PCCs, PSCs and primary bone marrow derived macrophages (Fig. 3a). Consistent with the data presented in the initially submitted manuscript we observe *Osm* expression in BMDMs (Fig. 3a). Further, *OsmR* is expressed in PSCs and is increased by conditioned medium from co-cultures of PCCs+PSCs. Three-way co-cultures with PCC+PSC+BMDM as well as GM-CSF+PSC+BMDM further increase PSC *OsmR* expression (Fig. 4d). This data confirms the cell-specific expression of *Osm* and *Osmr* using primary BMDMs and that GM-CSF replicates the effect of PCCs on *Osmr* expression in PSCs.

In addition, we examined the expressed of selected inflammatory genes (*Il6*, *Il4ra* and *Cxcl1*) and the myofibroblastic marker gene *Acta2* in PSCs stimulated with conditioned medium from mono-cultures of bone marrow-derived macrophages (MØ), PSCs and PCCs as well as co-cultures of MØ+PSCs, PSCs+PCCs, MØ+PSCs+PCCs and MØ+PSCs+GM-CSF (Fig. 2f and 4d). Conditioned medium from co-cultured PSCs+MØs does not induce expression of *Il6*, *Il4ra*, *Cxcl1* in PSCs (Fig. 2f). *Acta2* expression also doesn't change, suggesting that interactions between MØs and PSCs does not induce a similar change as observed between PCCs, PSCs and MØs (Fig 2f). Consistent with the data included in the original submission, MØ+PSCs+PCCs increase *Il6*, *Il4ra* and *Cxcl1* expression and decrease *Acta2* expression in PSCs (Fig. 2f and 4d). Importantly, the effect of PCCs can be substituted by GM-CSF, e.g. MØs+PSCs+GM-CSF conditioned medium result in similar increase in *Il6*, *Il4ra*, *Cxcl1* and *Acta2* expression as MØs+PSCs+PCCs (Fig. 4d).

The new data shows that PSC+BMDM does not change expression of inflammatory or myofibroblastic marker genes in PSCs suggesting that PCCs play a critical role in co-opting stromal cells to alter PSC marker expression. GM-CSF is able to both induce *Osm* in MØs and to substitute for PCCs in co-culture mediated fibroblasts reprogramming suggesting that indeed GM-CSF plays an important functional role in driving deregulated tumour-macrophage-fibroblast interactions.

6) Similar to the point above, whereas tumour growth is clearly impaired in OSM KO mice, it is less clear whether this is indeed dependent on a change in differentiation status of the cancer cells, as suggested by the authors. More evidence should be provided in support of a more differentiated histology of tumours in OSM KO mice (e.g. MUC5AC IHC ECAD IHC). Moreover, are Ki67 or pH3 levels different in the 2 cohorts? And CC3 stain? Additionally, the authors should clarify how they performed the epithelial/mesenchymal PanCK quantifications in figure 6.

These are all fair points that the reviewer raises. However, due to the COVID pandemic, we have been restricted in the animal work we could undertake and have therefore not been able to generate additional tumour for the proposed analyses – instead we have analysed available tumours albeit with the caveat that the tumour cells cannot be easily and specifically identified since no antibody exists which recognise the ectopically expressed iRFP used to demark the tumour cells.

Specifically: IHC for CD8, CD31, CC3, Ki67, SLUG, SNAIL and Picrosirius Red have been included in the resubmitted manuscript to confirm the *in vivo* and *in vitro* data.

The quantification of epithelial and mesenchymal cell morphology (Fig. 7b) was performed using the Tissue Classifier machine learning algorithm on HALO and has been included in the materials and methods section.

7) To make conclusions in Figure 3 (panels a, b, c, d), all ELISA results should be complemented by qPCR validation (and ideally also proliferation assays), as cell proliferation differences may lead to misleading results (i.e. more cells could result in more ligand in the media regardless of whether the ligand is actively induced). Similarly, the authors should discuss/show whether the Luminex data in figure 2a may be dependent on differences in cell proliferation. Similarly, the authors mention “despite maintaining a constant total cell number” on lines 145-146. However, it is unclear how this was evaluated, and these data should be shown.

Thank you for this comment. Firstly, we should have been clearer in how we described the cell seeding for the experiment in Fig. 2a, and have now changed the text to hopefully clarify this. In the experiments comparing mono- and co-culture conditioned medium we seeded the same total cell number at the start of the experiment. This is important because comparison between mono-cultures and co-cultures can be set up with either same number of cells seeded for each cell type e.g. total cell number increase between mono-, 2W-, and 3W- co-cultures or with total amount of cells seeded constant (but decreased number of individual cell type seeded, which is what we did). Thus, when observing an increased level of inflammatory signals between 2W and 3W co-cultures this is more likely a reflection of altered cell to cell interactions and not due to an increased number of total cells seeded (initially). We acknowledge that differences may exist between cell proliferation of individual cell populations seeded in mono-, 2W and 3W cultured, which is why we followed up and FACS

isolated individual cell populations for RT-qPCR and RNA-seq analysis of individual signals (Fig. 2b and c).

We also agree that we cannot exclude differences in proliferation of the PCCs seeded for ELISA analysis of GM-CSF (Fig. 4), however the main point of this experiment is to establish a link between the amount of GM-CSF secreted and the level of OSM secretion in MØs treated with the conditioned medium. This link was further established using inhibitors and recombinant GM-CSF (Fig. 4c and 4d). Thus, we have not included RT-qPCR analysis or proliferation assays for all of these conditions since it would not significantly change the conclusion of these experiments.

8) Clear validation of the OSMR KO in PSCs should be included (e.g. western blot or DNA sequencing analysis – qPCR analysis is not sufficient). Can the authors also please clarify whether these are pooled lines or derived from single clones?

We have included DNA sequence analysis of the PSC *Osmr* KO in fibroblasts (Supplementary Data). Unfortunately, there are no available antibodies for murine *Osmr* and therefore we have not been able to confirm KO by western blotting. The *Osmr* KO fibroblasts were generated as pools using nucleoporation of multiguide gRNA/Cas9 complex. The efficiency is very high in fibroblasts and thus there is no need to select single cell clones.

Minor points

1) Some panel figures require more information or need to be fixed, for example:

a. figure 1a requires labels (quiescent PSCs, myCAFs, iCAFs).

We apologise for the omission and have updated figures and legends in the resubmitted version of the manuscript.

b. I believe 1c (right panel, mouse PDA) is not correct. It appears *Osmr* is high in ductal cells and RBCs but the main text states otherwise.

Thank you for picking this up. The reviewer is correct, and we have corrected the figure in the resubmitted version of the manuscript.

c. As mentioned above, I suggest to include a dot blot or normalised heatmap plot instead of/in addition to all violin plots, as it will show more clearly the differences in gene expression between clusters.

We thank the reviewer for making this point. Where possible we have now included individual data points on the violin plots to more clearly illustrate the data distribution and we have included heat maps of relative expression.

d. In the methods, the authors talk about 8513 cells for the generation of transplantation models, however in figure 3a/c they are not shown and it is unclear for which in vivo experiments they have been used.

We apologise for the lack of clarity. All cell lines used in the study has been annotated and referenced in the Supplementary Table 1.

2) I believe figure 6D is not cited in the text.

Thank you for catching this – we have now included a reference for the proposed model in the discussion (now Fig. 7e).

3) The authors perform flow cytometry/FACS on fibroblasts by gating on double positive PDPN and CD90 cells. However, CD90 (=Thy1) has been shown to be more highly expressed in myCAFs in single-cell RNA-sequencing datasets of PDA and therefore it seems confusing to look at inflammatory markers in this population. Although I do not think it is needed for publication to repeat the sorting + qPCR analysis on all PDPN+ cells, the authors should comment on this caveat of their experimental design and clarify their choice. Related to this, as a suggestion, the authors could look at the datasets they already have and calculate the % of CD45-EPCAM-PDPN+CD90+ (myCAFs-enriched) and CD45-EPCAM-PDPN+CD90- (iCAFs-enriched) in control WT mice and Osm KO mice as proxy for detecting changes in iCAF/myCAF populations. Do PDPN+CD90+ increase and PDPN+CD90- decrease as suggested by the other data reported?

The selection of fibroblast markers enabling efficient capture of all CAF subsets is certainly important. While PDPN and CD90 are indeed differentially expressed across CAF subsets, which is a caveat for many markers, both markers are still expressed (Hutton et al Cancer Cell 2021, see below). Thus, when used to isolate all positive cells (Extended Fig. 5b) these markers still work.

4) In Figure 4b, it appears some OSM is detected. Can the authors discuss why this is the case? Is this OSM from the transplanted cancer cells since this is a whole-body KO mouse? How does this affect the model that the authors propose?

Thank you for picking this up. We re-did the ELISA with the same samples and did not detect OSM, indicating that the low levels observed in the initial data set were false positives. We have updated the figure and data.

5) In Figure 4, do the changes in SMA levels correlate with changes in matrix deposition (e.g. Masson's trichrome stain) since myCAFs have been associated with that and this may have an effect on the epithelial differentiation status?

To address whether fibrillar collagen is changed in tumours of Osm WT and OSM KO animals we have now included an analysis using Picrosirius Red (Extended Fig. 5a). Curiously, we do not observe any differences in the staining intensity, suggesting that fibrillar collagen levels are not different under these two conditions. Importantly, this does not demonstrate that the ECM isn't composed differentially between tumours in Wt and *Osm*^{-/-} animals. To address this comprehensively we are aiming to undertake a proteomics experiment (See papers by Naba and colleagues). Unfortunately, due to COVID we have been restricted in our ability to undertake animal experiments and thus these analyses will be included in future studies.

6) The authors should briefly discuss the limitations of the monolayer co-culture system used. For example, Figure 2 shows no induction of inflammatory gene signatures in PSCs upon co-culturing with cancer cells, whereas SMA is induced (Extended figure 3a), which suggest that cancer cells do not induce an inflammatory phenotype in PSCs, which is not correct as shown by other studies and is likely due to the fact that here the PSCs are cultured in monolayer. Whether macrophages would have such a profound effect in inducing the iCAF phenotype in the presence of cancer cells/organoids in less stiff/bi-dimensional conditions (which are rather different from the *in vivo* situation) should also be briefly discussed.

We agree that indeed any *in vitro* culture systems will be restricted in reflecting only some aspects of the more complex scenario observed in human tumours. Addressing this we have now included a discussion of the limitation of the model system and highlighted where additional experiments are needed.

7) As IL1 has been shown to induce the iCAF phenotype, the authors should look at IL-1 (alpha and beta) expression in cancer cells, macrophages and fibroblasts in the mono-cultures and 2-way, 3-way co-cultures in figure 1b. The authors perform good control experiments with an anti-IL1 beta antibody, however IL1 alpha could also play a role in their system and this possibility should be discussed or experimentally addressed.

To address this, we have now included a control comparing blocking antibodies for IL1a and IL1b which lead to comparable results (Extended Fig. 3d).

8) The authors mention that PSCs were only used for 5 passages in the method section. Can the authors please clarify whether this is also true for the *Osmr* KO PSCs, as it appears difficult to generate Cas9 cells, perform a KO, validate the KO and perform all experiments within 5 passages from the initial isolation of the cell line.

The reviewer is correct and we apologise for the lack of clarity. The genetically manipulated PSCs were used below passage 9 and all other PSCs were used below passage 5. We have updated the materials and methods to reflect this.

9) The authors should briefly discuss the possibility that the results presented in figure 4b may not be solely due to changes in the fibroblast compartment, but also in cancer cells and immune cells.

This is a reasonable comment and we have included this point in discussing the limitations of the model system and future experiments needed.

10) The authors should briefly discuss the limitations of the PDA cell line-transplantation model they use since, as it is typical for these models, the tumours (see figures 4 and 6) are highly cellular, which is not reflective of human PDA.

This is a reasonable comment and we have included this point in discussing the limitations of the model system and future experiments needed.

11) I find confusing that the levels of PDPN, which is a pan-fibroblast marker in vivo, increase in the triple co-culture (e.g. extended figures 2h and 3b). Can the authors please comment on this? Are the authors suggesting that PDPN is an iCAF marker in their in vitro system?

PDPN is broadly expressed in fibroblasts of the pancreas, and was included as a general marker of fibroblasts. Both myofibroblastic and inflammatory fibroblasts express PDPN, however inflammatory fibroblasts display increased levels of PDPN. In addition to the *in vitro* co-culture data included in this manuscript, mass cytometry analysis of murine KPC tumours (Hutton et al. Cancer Cell 2021) also show that inflammatory CAFs display more abundant PDPN than myofibroblastic CAFs (see Figure insert), however there is clearly a high degree of inter-tumour variability in PDPN levels.

12) On line 104, the authors state that OSMR, ANTXR1 and IL1R1 expression is restricted to fibroblasts. As above, I would be cautious with these statements, as by sub-clustering on specific populations it is clear that this is not the case, and as single-cell RNA-sequencing does not show lower gene expression levels. I would change this sentence to “largely restricted”.

This is a fair point and we have amended the text

13) Can the authors include OSMR expression levels in iCAF/myCAF in human PDA in addition to figure 1b?

This is a good point and we would be very interested in exploring this further. Unfortunately, we have been unable to address this, possibly due to the read depth of available human CAFs scRNA-seq data.

14) Contrary to what stated on lines 273-274, Figure 3C and 3A do not entirely correlate. PCC12 does, but PCC7 (and PCC) less so. The authors should re-phrase these lines and perhaps comment on this discrepancy.

This is a fair point and we have amended the text.

The following experiments will not be required for publication in Nature Communications, but they would strengthen the authors' conclusions if available:

- 1) Can OSM KO macrophage lines be generated (e.g. isolated from Osm KO mice) as a complementary approach to the OSMR KO PSC experiments in vitro?
- 2) Can GM-CSF KO cancer cells be generated as a complementary approach to the neutralizing antibody experiments in vitro?
- 3) Can the authors sort out cancer cells from OSM KO mice and WT mice and evaluate EMT/proliferation signatures/genes?

The reviewer raises a series of interesting questions, which are all well worth exploring. However, given the limited possibility of additional murine experiments we will have to defer these experiments for future studies. We thank the reviewer for the very good suggestions and hope to soon be able to continue with these experiments.

Reviewer: Giulia Biffi

REVIEWERS' COMMENTS

Reviewer #1 (Remarks to the Author):

The author indicated that the two published datasets not supporting their conclusion have poor quality, which sounds like cherry-picking attitude. The generalizability of their findings across different datasets is critical. At least, the author should clearly show what distinction has been found due to the 'poor' quality of the other dataset. Was the distinction caused by the 'poor quality' or by the heterogeneity of the sampling.

When I raised this question, my concern was the author claimed the new signaling pathway was specific to the CAF. If that is the case, the gene expression should be specific to the fibroblast or there could be other source of production. I am not sure if this is the case in the two 'poor quality' datasets. Unless the author show how the key gene expressions look like in all the datasets mentioned in the information exchange, I am not 100% positive to the conclusion of this manuscript.

Reviewer #2 (Remarks to the Author):

After going through the point-by-point response and evaluating the revised manuscript, I am pleased to say that the authors have addressed well all my initial concerns.

Congratulations to this very interesting study.

Reviewer #3 (Remarks to the Author):

I believe that the authors have addressed well, with either additional experimental data or by expanding the discussion, both my main points and those of the other reviewers. I do not have additional concerns and I welcome publication of this study in Nature Communications.

We would like to thank the reviewers for taking time to read and comment on the revised version of our manuscript NCOMMS-20-46941A.

Below, please find our response to the concerns and critiques raised:

Reviewer #1 (Remarks to the Author): with expertise in cancer and transcriptomic

The author indicated that the two published datasets not supporting their conclusion have poor quality, which sounds like cherry-picking attitude. The generalizability of their findings across different datasets is critical. At least, the author should clearly show what distinction has been found due to the 'poor' quality of the other dataset. Was the distinction caused by the 'poor quality' or by the heterogeneity of the sampling.

When I raised this question, my concern was the author claimed the new signaling pathway was specific to the CAF. If that is the case, the gene expression should be specific to the fibroblast or there could be other source of production. I am not sure if this is the case in the two 'poor quality' datasets. Unless the author show how the key gene expressions look like in all the datasets mentioned in the information exchange, I am not 100% positive to the conclusion of this manuscript.

This is a reasonable point and we acknowledge we should have included the comparison of the data sets in the first response to the reviewer:

To avoiding miss-classification of gene expression, accurate and consistent annotation of cell identity is important. To ensure similar cell identities are assigned across available human scRNA-seq data included in this analysis we had to re-assign cell identification, using same parameters, across all three data sets. While using a limited number of genes to annotate cell identity carries a risk of miss-assigning individual cell population, the use of a large fixed lists of specific genes becomes slightly trickier as differences in drop-out between data sets will limit the number of genes available to include in the analysis.

To circumvent this challenge, we used a more flexible approach. Here a list with genes for cell type annotation (between 20-100 genes pr cell type) from Elyada et al, was used, but rather than requiring that all genes must be identified in order to annotate a cell type, we assign cell types based on a fraction of the genes belonging to a distinct cell type e.g. for a given threshold a specific fraction of the genes on the list should be met in order to assign cell identify whereas the specific genes used on the list was not pre-set.

Below have included the re-analysed scRNA-seq data from Ping et al, Ling et al and Elyada et al with thresholds of 0.1 (only for Ling and Ping), 0.3, 0.5 and 0.7. As the threshold increase e.g. the fraction of the genes from the list required to be detected before 'assigning' cell types, the number of cells that can be interrogated in a meaningful manner decrease. Below each data set we have included violin plots of OSM and OSMR across individual cell populations.

What's notable is that the number of cells that 'drop out' as threshold increase is much higher for Ling et al and Ping et al data set. This can be visualised both by the number of black dots on the tSNE but also on the bar-graph at the top of each tSNE plot (which shows the number of cells binned across the threshold).

To ensure optimal annotation of cell identity we used a stringent cut-off for and focussed on the human data from Elyada et al, in this manuscript. Non the less, the points raised by this reviewer, as well as the other reviewers, related to the cell-specificity remains important. We therefore included ISH analysis of OSMR mRNA expression in the previously submitted

version of the manuscript (Fig 1d and Supplemental Fig 1c). Moreover, while the data in this manuscript demonstrate a heterocellular role of OSM-OSMR signalling between macrophages and fibroblasts, we acknowledge and discuss the limitations of this study and the need to further confirm whether OSMR also play a functional role in a subset of epithelial cells.

Figure legend: Threshold evaluation of Elyada et al (top), Lin et al (middle) and Peng et al (bottom) with cut-off of 0.3,0.5 and 0.7 (Elyada et al) or 0.1, 0.3, 0.5 and 0.7, (Lin et al and Peng et al). OSM and OSMR expression included below each threshold setting. The top bar graph of each panel annotates the number of cells available for analysis e.g. for 0.5 and 0.7 cut-off Elyada et al: 92% and 70%; Lin et al: 80% and 17% and Peng et al: 39% and 5%

We are very excited to resubmit the manuscript and trust that the reviewers find that the additional data and textual edits have addressed the key concerns.

Dr Claus Jørgensen